# Reduced phosphorus loads from the Loire and Vilaine Rivers were accompanied by increasing eutrophication in Vilaine Bay (South Brittany, France)

Widya Ratmaya[1]*, Dominique Soudant[2], Jordy Salmon-Monviola[3], Nathalie Cochennec-Laureau[1], Evelyne Goubert[4], Françoise Andrieux-Loyer[5], Laurent Barillé[6], Philippe Souchu[1]

[1] Ifremer – LER MPL, Rue de l'Ile d'Yeu, BP 21105, 44311 Nantes Cedex 03, France

[2] Ifremer, VIGIES, Rue de l'Ile d'Yeu, BP 21105, 44311 Nantes Cedex 03, France

[3] INRA/Agrocampus Ouest – UMR1069 Sol Agro et Hydrosystème Spatialisation, 35000 Rennes, France

[4] Université de Bretagne Sud – GMGL, UMR CNRS 6538 DO, Campus Tohannic, 56000 Vannes, France

[5] Ifremer – DYNECO PELAGOS, ZI Pointe du Diable, 29280 Plouzané, France

[6] Université de Nantes, Mer Molécules Santé EA 2160, Faculté des Sciences et des Techniques, BP 92208, 44322 Nantes cedex 3, France

*Correspondence to*: Widya Ratmaya (widya.ratmaya@ifremer.fr)

**Abstract.** The evolution of eutrophication parameters (i.e., nutrients and phytoplankton biomass) during recent decades was examined in coastal waters of the Vilaine Bay (VB, France) in relation to those in the Loire and Vilaine Rivers. Dynamic Linear Models were used to study long-term trends and seasonality of dissolved inorganic nutrient and chlorophyll *a* concentrations (Chl *a*) in rivers and coastal waters. For the period 1997-2013, the reduction in dissolved riverine inorganic phosphorus concentrations (DIP) led to the decrease in their Chl *a* levels. However, while dissolved inorganic nitrogen concentrations (DIN) decreased only slightly in the Vilaine, they increased in the Loire, specifically in summer. Simultaneously, phytoplankton in the VB underwent profound changes with increase in biomass and change in the timing of the annual peak from spring to summer. The increase in phytoplankton biomass in the VB, manifested particularly by

increased summer diatom abundances, was due to enhanced summer DIN loads from the Loire, sustained by internal regeneration of DIP and dissolved silicate (DSi) from sediments. The long-term trajectories of this case study provide a more evidence that significant reduction of P inputs without simultaneous N abatement was not yet sufficient to control eutrophication all along the Loire/Vilaine – VB continuum. Upstream rivers reveal indices of recoveries following the

significant diminution of P, while eutrophication continues to increase downstream, especially during the period of N limitation. More N input reduction, paying particular attention to diffuse N-sources, is required to control eutrophication in receiving VB coastal waters. Internal benthic DIP and DSi recycling appears to have contributed to the worsening of summer VB water quality, augmenting the effects of anthropogenic DIN inputs. For this coastal ecosystem, nutrient management strategies should consider the internal nutrient loads in counteracting decreased external inputs.

**Keywords**: eutrophication, phytoplankton, internal nutrient loads, dual nutrient reductions, Vilaine Bay, Dynamic Linear Models

## 1 Introduction

Anthropogenic eutrophication is widely regarded as one of the major problems affecting both inland and coastal aquatic ecosystems (Downing, 2014). The increase in phytoplankton biomass is the most common symptom of eutrophication

among the myriad responses of aquatic ecosystems to anthropogenic inputs of nitrogen (N) and phosphorus (P) (Cloern, 2001; Glibert et al., 2011). Since the beginning of the 1990s, measures to reduce nutrient inputs in European rivers were more effective for P, originating largely from point sources, than for N, coming mainly from diffuse sources (Grizzetti et al., 2012). However, this strong imbalance between N and P input reduction still led to substantial decrease in phytoplankton biomass in many European rivers (Istvánovics and Honti, 2012; Romero et al., 2013). This result is consistent with the idea

that P universally limits primary productivity in many freshwater ecosystems (Correll, 1999). Thus, reducing P inputs, and not N, can mitigate eutrophication of freshwater ecosystems (Schindler et al., 2008; Schindler et al., 2016).

Despite significant P input reduction, eutrophication persists in some rivers (Neal et al., 2010; Bowes et al., 2012; Jarvie et al., 2013), and particularly in downstream coastal ecosystems, where the primary productivity is often limited by N (Ryther and Dunstan, 1971; Howarth and Marino, 2006; Paerl, 2018). As freshwater systems drain into coastal waters (Vannote et al., 1980; Bouwman et al., 2013), the efficient P reduction without simultaneous N abatement may result in more N being transported downstream, where it can exacerbate eutrophication problems in coastal ecosystems, delaying recovery (Paerl et al., 2004), for example the Neuse River Estuaries (Paerl et al., 2004), the Chesapeake Bay (Harding et al., 2016), Belgian coastal waters (Lancelot et al., 2007), and the Seine Bay (Romero et al., 2013). Despite more than 20 years of nutrient reduction implementation in European freshwater ecosystems, including rivers (e.g., Nitrates Directive, 91/676/EEC; Urban Waste Water Treatment Directive, 91/271/EEC), little measurable progress has been observed in many European coastal waters (EEA, 2017; OSPAR, 2017).

The Loire River, alongside the Vilaine River, are among these major European rivers whose phytoplankton biomass and P concentrations have decreased since the early 1990s, but with minor, if any, simultaneous diminution in N concentrations (Romero et al., 2013; Minaudo et al., 2015). Affected by the Loire and Vilaine river runoff (Guillaud et al., 2008; Gohin, 2012; Ménesguen et al., 2018b), the Vilaine Bay (VB) is one of the European Atlantic coastal ecosystems most sensitive to eutrophication (Chapelle et al., 1994; Ménesguen et al., 2014, 2019). The VB coastal waters are classified as a problem area due to elevated phytoplankton biomass, according to the criteria established within OSPAR (OSPAR, 2017) and the European Water Framework Directive (Ménesguen et al., 2018b). However, there is little information on how eutrophication

parameters have evolved in the VB over the past 20 years in the light of eutrophication mitigation in the Loire and Vilaine Rivers. An approach taking into account seasonal variations is required as phytoplankton in many coastal ecosystems, such as the coastal waters off the Loire and Vilaine Rivers, is often limited by P in spring and by N in summer (Lunven et al., 2005; Loyer et al., 2006).

In temperate coastal waters, diatoms and dinoflagellates constitute the two dominant phytoplankton classes (Sournia, 1982; Sournia et al., 1991). In term of nutrient requirements, the balance between these classes is controlled by silica (Si) availability. Increased inputs in N and P (and not Si) in aquatic ecosystems can lead to limitation in diatom biomass due to lack of dissolved silicate (Conley et al., 1993). Therefore, increasing eutrophication may favor the development of non-siliceous algae, such as dinoflagellates and harmful species (Billen and Garnier, 2007; Lancelot et al., 2007; Howarth et al.,
2011).

The present study investigated the long-term evolution (trend and seasonality) of eutrophication parameters (dissolved inorganic nutrient concentrations and phytoplankton biomass) in the VB coastal waters, in relation to those in the Loire and the Vilaine between 1980 and 2013, using Dynamic Linear Models. This long-term ecosystem-scale analysis provided an opportunity to test the hypothesis that eutrophication trajectories in the downstream VB coastal waters during recent decades
have been influenced by those in the Loire and Vilaine Rivers. We aim to establish the link between fresh and marine water trajectories and highlight the impact of nutrient reduction strategies in rivers on coastal water quality.

## 2 Material and Methods

### 2.1 Sites

The Loire is the longest and widest river in France (1,012 km) with a watershed of 117,000 km$^2$, while the Vilaine watershed is only 10[th] the size, with an area of 10,800 km$^2$ (Fig. 1). Their catchment areas are dominated by agricultural activity, together sustaining two-thirds of the national livestock and half the cereal production (Bouraoui and Grizzetti, 2008; Aquilina et al., 2012). The Arzal dam, 8 km from the mouth of the Vilaine, was constructed in 1970 to regulate freshwater discharge and prevent saltwater intrusion (Traini et al., 2015). The two studied rivers, especially the Loire, are the main nutrient sources in the northern Bay of Biscay, including VB (Guillaud et al., 2008; Ménesguen et al., 2018a).

The VB, average depth 10 m, is located under direct influence of these two rivers (Fig. 1). The Loire river plume tends to spread north-westward with a dilution of 20 to 100-fold by the time it reaches the VB (Ménesguen and Dussauze, 2015; Ménesguen et al., 2018b). The ECO-MARS3D model estimates that the Loire constitutes >60% of VB DIN concentrations during flood regimes and from 20 to 40% during low discharge periods (Gohin, 2012; M. Plus, Ifremer Brest, pers. comm.). The Vilaine river plume tends to spread throughout the bay before moving westward (Chapelle et al., 1994).

The water residence time in the VB varies between 10 and 20 days depending on the season and tends to be longer during calm periods (Clément, 1986; Chapelle, 1991), with tidal ranges varying between 4 and 6 m (Merceron, 1985). The water circulation is characterized by low tidal and residual currents, driven mainly by tides, winds and river flows (Lazure and Salomon, 1991; Lazure and Jegou, 1998). During periods of prevailing winds, particularly from south-west and west, the water column of the VB is subjected to vertical mixing, which can lead sometimes to sediment resuspension and high

turbidity (Goubert et al., 2010). Except during winter and period of high hydrodynamic activity, phytoplankton production in the VB is not limited by light (Guillaud et al., 2008).

## 2.2 Long-term monitoring dataset: Rivers and VB

The Loire-Brittany River Basin Authority (http://osur.eau-loire-bretagne.fr/exportosur/Accueil) furnished dissolved

inorganic nutrients and phytoplankton biomass data (dissolved inorganic phosphorus concentrations, DIP; dissolved inorganic nitrogen concentrations, DIN, dissolved silicate concentrations, DSi and chlorophyll *a* concentrations, Chl *a*) in rivers, at pre-estuarine stations located closest to the river mouth upstream of the haline intrusion (Fig. 1). DIN was defined as the sum of nitrate, nitrite and ammonium, with nitrate as the major component (>90%). Sainte-Luce-sur-Loire on the Loire and Rieux on the Vilaine provided DIP, DIN and Chl *a*, measured monthly since the 1980s. For Sainte-Luce-sur-Loire,

the influence of tidal dynamics was avoided by discarding data collected during high tide. Monthly DSi data were available from 2002 at Montjean-sur-Loire on the Loire and at Férel on the Vilaine (Fig. 1).

In order to calculate riverine nutrient loads, gauging stations located close to the river mouth were selected. River discharge data were extracted from the French hydrologic "Banque Hydro" database (http://www.hydro.eaufrance.fr/). For the Loire, river discharge measurements at Montjean-sur-Loire were used due to the absence of data at Sainte-Luce-sur-Loire. For the

Vilaine, daily discharge data were available at Rieux from the 1980s. DIN and DIP loads from rivers were calculated using averaged monthly discharge and individual monthly nutrient concentrations (Romero et al., 2013).

Nutrient and Chl *a* concentrations, plus phytoplankton count data in the VB, provided by the French National Observation Network for Phytoplankton and Hydrology in coastal waters (REPHY, 2017), were collected from Ouest Loscolo station

(Fig. 1). This station is representative of the VB coastal waters (Gohin, 2011; Bizzozero et al., 2018; Ménesguen et al., 2019) and displayed the longest dataset (from 1983 for phytoplankton counts and 1997 for nutrient and Chl *a* concentrations).

Acquisition periods, sampling frequencies and methods of analysis are detailed in Table S1. Briefly, nutrient concentrations were measured manually or automatically in flow analysis using standard colorimetric methods with fluorimetry or photometry detection. Chlorophyll *a* concentrations (Chl *a*) were measured with either spectrophotometry or fluorimetry. Microscopic quantitative micro-phytoplankton analyses in coastal waters were conducted on Lugol-fixed samples and counted according to Utermöhl (1958). Phytoplankton identification and counts were carried out for organisms whose size is >20 µm (i.e., micro-phytoplankton) and smaller species with chain structure. Further details about sampling and processing of phytoplankton species are available in Hernández-Fariñas et al. (2014) and Belin and Neaud-Masson (2017). In order to account for the role of DSi, of all the micro-phytoplankton classes, genera and species identified in the VB, only total counts of diatoms (Bacillariophyceae) and dinoflagellates (Dinophyceae) were used in this work. Other micro–phytoplankton classes (Dictyophyceae, Prasinophyceae, Cyanophyceae, Chrysophyceae and Raphidophyceae) together represented only 10 to 15 % of the VB total counts (Belin and Soudant, 2018).

### 2.3 Time-series analyses

### 2.3.1 Data pre-processing

Prior to analysis, all datasets were examined using time scaled scatter plots. For DIP in rivers, these showed periods during which a limited set of values appeared repeatedly (Fig. S1), which resulted from analytical problems (Loire-Brittany River Basin Authority, S. Jolly, pers. comm.). Consequently, these suspect data were discarded to avoid misinterpretation. The removed DIP datasets represented 29% and 31% of the total number of data, corresponding respectively to the period 1980-

1989 in the Loire, and 1980-1989 and 2009-2011 in the Vilaine. DSi in rivers was not analyzed for trends because of the short data period.

Prior to time series decomposition, a variance-stabilizing base $e$ log transformation was applied to all variables, except for phytoplankton counts for which the base was 10, to ensure compliance with the constant variance assumption (i.e. homoscedasticity).

### 2.3.2 Time-series decomposition

The time-series were modeled using Dynamic Linear Models (DLM, West and Harrison, 1997) with the *dlm* package (Petris, 2010) in R software (R core team 2016). This tool belongs to the family of methods which encompass, for example, State-Space models, Structural Time Series Model, Unobserved Component Model (Harvey et al., 1998) and Dynamic Harmonic Regression (Taylor et al., 2007). The model decomposes an observed time-series into component parts, typically trend, seasonal component (i.e., seasonality) and residual. The DLM approach is particularly suitable for environmental data series characterized by outliers, irregular sampling frequency and missing data. The latter are taken into account by the Kalman filter (Kalman, 1960), using a prior which replaces the missing value, i.e., no information leads to no change in distributions for model parameters (West and Harrison, 1997). For other examples of DLM applications, readers are referred to Soudant et al. (1997), Scheuerell et al. (2002), and Hernández-Fariñas et al. (2014, 2017).

The model used was a second order polynomial trend, which allows modelling up to quadratic trend. This was chosen because linear trend (i.e., first order polynomial) was too restrictive and cubic trend (i.e., third order polynomial) might lead to an over fitted model. For the seasonal component, the model used was trigonometric with two harmonics, which allows

modelling up to bimodal pattern. This bimodal pattern is characterized by two peaks per year, such as spring and autumn or summer and winter blooms. This model specification was used for all parameters.

The time unit was defined as the smallest time interval between sampling dates within a period of analysis (i.e., one year). The time unit was weekly, fortnightly or monthly according to sampling frequencies of variables (see Table S1). Normality of standardized residuals was checked using QQ-plot and their independence using estimates of autocorrelation function. If deviations were suspected, outliers were identified as 2.5 % higher and lower than standardized residuals and treated appropriately, i.e., specific observational variances were estimated for each outlier. The DLM time-series analysis provides figures allowing the visual identification of trends and variations in seasonality.

### 2.3.3 Trend

The DLM trend plot displayed observed values with a shade of color for each time unit segments: weekly, fortnightly or monthly. The trend was represented by a dark grey line with the shaded area indicating the 90% confidence interval. For the longest common record of all variables, 1997-2013 called the "common period", a monotonic linear trend significance test was performed on DLM trend components using a modified non-parametric Mann-Kendall (MK) test (Yue and Wang, 2004). When monotonic linear trends were significant ($p<0.05$), changes were calculated from differences between the beginning and the end of the common period of the Sen's robust line (Helsel and Hirsch, 2002).

### 2.3.4 Seasonality

The seasonality plot displayed the DLM seasonal component values. The figure gave a visual access to the inter-annual evolution of the amplitude, corresponding to the difference between the minimum and maximum values of each year. As dependent variables have been log-transformed, the model was multiplicative. Therefore, when seasonal component values

equaled to 1 (i.e., horizontal line), fitted values equaled to the trend. The seasonality plot also allowed a visualization of how the values have evolved over the years according to their seasonal position. The significance of changes in the seasonality (monotonic linear increase or decrease in the value for a given season) was assessed for the common period using the modified MK test performed on DLM seasonal components for each season. The seasons were defined as winter (January,

February, March), spring (April, May, June), summer (July, August, September), and autumn (October, November, December). The interpretation of the seasonal components per se was not meaningful, therefore changes were not calculated, but when monotonic linear trends were significant ($p<0.05$), the sign and the percentage of the changes were provided.

### 2.4 Correlation analysis

Spearman Correlations were computed for annual median values of the common period in order to analyze relationships

between variables, and tested using STATGRAPHIC CENTURION software (Statgraphics Technologies Inc., Version XVII, Released 2014).

## 3 Results

### 3.1 Long term trends in eutrophication parameters in river basin outlet

The daily discharge of the Loire varied between 111 and 4,760 $m^3$ $s^{-1}$ for the period 1980-2013, with DLM trend displaying

oscillations with periodicities of 6-7 years (Fig. 2a). A significant negative trend was detected for the common period (1997-2013), with a decrease of 94 $m^3$ $s^{-1}$ (Table 1). The seasonality plot displayed no marked change, with maximum values always observed in winter (blue) and minimum in summer (orange/red, Fig. 2b) and no significant linear change whatever the season (Table 2). The Vilaine discharge, median of 32 $m^3$ $s^{-1}$ for the period 1980-2013), corresponded to 6 % of the Loire

discharge and displayed similar trend and seasonality to those of the Loire (Fig. S2, Table 1, 2), as highlighted by the significant correlation between their annual medians (Table 3).

DIP in the Loire varied between 0.1 and 9.4 µmol $L^{-1}$ for the period 1990-2013 (Fig. 3a). A significant decrease of 0.85 µmol $L^{-1}$ was detected for the common period (Table 1). Also during this period, the seasonality plot indicated a noteworthy shift in timing of annual DIP minima from summer to spring, as indicated by its change in color from yellow/orange (summer) in 2000 to green (spring) from 2006 onwards (Fig. 3b). This change was accompanied by a significant negative trend for winter-spring seasonal components and a significant positive trend for summer-autumn ones (Table 2). DIP loads from the Loire ranged between <0.1 and 15 mol $s^{-1}$ for the period 1990 2013, with trend displaying oscillations reflecting the influence of river discharge (Fig 3c). For the common period, the Loire DIP loads decreased significantly by 52% (Table 1). The seasonality plot of DIP loads from the Loire reflected that of discharge with annual minimum and maximum values always observed in summer and winter respectively (Fig. 3d). Trends of DIP and DIP loads for the Vilaine were similar to those for the Loire (Fig. S3, Table 1, 2), as indicated by a significant correlation between annual medians of DIP in the two rivers (Table 3).

DIN in the Loire ranged between 11 and 489 µmol $L^{-1}$ for the period 1980-2013, with trend displaying a decrease between the 1980s and the early 1990s, followed by an increase (Fig. 4a). However, the increase was not significant for the common period (Table 1). The DLM Loire DIN seasonality plot indicated a decrease in the seasonal amplitude starting in 1990 (Fig. 4b). For the common period, this decreasing amplitude resulted from a significant decrease in winter DIN maxima on the one hand and significant increase in summer minima on the other hand (Table 2) by around 60 µmol $L^{-1}$ (Fig.4a). The DIN loads from the Loire varied from <1.0 to 1,142 mol $s^{-1}$ and displayed similar trend and seasonality to those of DIN

(Figs. 4c, d), with an increase in summer minima from around 5 to 50 mol s$^{-1}$ for the common period (Fig. 4c, Table 2). The trend of DIN in the Vilaine displayed an oscillation (Fig. S4), with a slight significant decrease over the common period (Table 1) and no marked variation in the seasonality (Fig. S4b, Table 2). As for the Loire, the trend and seasonality of DIN loads from the Vilaine were similar to those of DIN (Figs. S4c, d, Table 1, 2).

DIN:DIP ratios in both rivers ranged between 1.0 and 1,000 with >80% of value being higher than 30 and displayed an increasing trend between 1990 and 2013 (Fig. S5). A significant increase of 85% and 303%, respectively for the Loire and the Vilaine, was detected for the common period (Table S3). DSi in rivers ranged between 46 and 261 µmol L$^{-1}$ in the Loire and from 5.0 to 201 µmol L$^{-1}$ in the Vilaine for period of available data (2002-2013). More than 80% of DIN:DSi ratios in rivers were higher than the theoretical molar N:Si ratio of 1 for potential requirement of diatoms (data not shown).

Chl $a$ in the Loire ranged between >200 µg L$^{-1}$ during the 1980s and <1.0 µg L$^{-1}$ in the 2010s. The Chl $a$ trend remained stable between 1980 and 2000 before decreasing subsequently (Fig. 5a). For the common period, the Loire Chl $a$ decreased by 93% (54 µg L$^{-1}$, Table 1). The DLM Loire Chl $a$ seasonality plot displayed a shift in timing of the annual Chl $a$ maximum, as indicated by its change in color from orange/red (summer) during 1980-1990 to green (spring) during 2005-2013 (Fig. 5b). For the common period, this change in timing was accompanied by a significant negative trend for

autumn seasonal components and significant positive trend for winter and spring (Table 2). Results for Chl $a$ in the Vilaine revealed similar trend and seasonality to those in the Loire (Fig. S6, Table 1, 2), as indicated by a significant correlation between Chl $a$ annual medians in the two rivers (Table 3).

## 3.2 Long term trends in eutrophication parameters in the VB

DIP in the VB varied between <0.1 and >1.0 µmol L$^{-1}$ with no noticeable trend (Fig.6a). A significant decrease of 0.05 µmol L$^{-1}$ was detected over the common period (Table 1). The seasonality plot of the VB DIP revealed a change in timing of the minimum values, as indicated by its change in color from yellow/orange (summer) before 2006 to green (spring) afterwards (Fig. 6b). This shift was accompanied by a significant negative linear trend for spring seasonal components and a significant positive trend for summer (Table 2).

DIN in the VB varied between <1.0 and >200 µmol L$^{-1}$ with trend displaying an oscillation (Fig. 6c). A significant increase of 3.2 µmol L$^{-1}$ was detected for the common period (Table 1). The DLM seasonality indicated that this increase was focused on winter (Fig. 6d, Table 2). Annual DIN medians in the VB were positively correlated with those of discharge from the two rivers (Table 3).

DSi in the VB varied between <1.0 and 100 µmol L$^{-1}$ without noticeable trend (Fig. 6e). For the common period, a significant increase of 3.6 µmol L$^{-1}$ was detected, which was comparable to that of DIN (Table 1). The seasonality did not indicate any particular change (Fig. 6f, Table 2). Annual DSi medians in the VB were positively correlated with those of the Loire discharge and with the VB DIN (Table 3).

DIN:DIP and DIN:DSi ratios in the VB ranged between <1.0 and 650, and from <0.1 to 44 respectively (Fig. S7). Summer values of DIN:DIP and DIN:DSi ratios were often below theoretical values respectively of 16 and 1 for potential requirements of diatoms (Fig. S7). DSi:DIP ratios in the VB ranged between <5.0 and >100, with >80% of values being above the theoretical value of 16 (Fig. S7). The trends for dissolved inorganic nutrient ratios in the VB displayed a significant increase for the common period (Fig. S7, Table S3).

Chl *a* in the VB ranged between 0.1 and 116 µg L$^{-1}$, with trend displaying an increase (Fig. 7a). For the common period, the VB Chl *a* increased significantly by 126% (2.1 µg L$^{-1}$, Table 1). The seasonality plot of Chl *a* in the VB displayed a shift in the timing of the annual maximum, indicated by its change in color from green (spring) before 2006 to orange/red (late summer) afterwards (Fig. 7b). This change was accompanied by a significant negative linear trend for spring seasonal components (Table 2). Annual Chl *a* medians in the VB were negatively correlated with those of Chl *a* from both rivers and with DIP in the Vilaine (Table 3).

Diatom abundances varied between 200 and 1.3 10$^7$ cells L$^{-1}$ for the period 1983-2013, with the DLM trend showing an increase (Fig. 7c). For the common period, diatom abundances increased significantly by 227% (90 10$^3$ cells L$^{-1}$, Table 1). Although diatom abundances continued to peak in spring (Fig. 7d), their seasonality plot indicated a significant increase in summer seasonal components over the common period (Table 2). Dinoflagellate abundances were about ten-fold less than those of diatoms, with values ranging between 40 and 3.4 10$^6$ cells L$^{-1}$ over the period 1983-2013. Like diatoms, the DLM trend for dinoflagellate abundances in the VB displayed an increase (Fig. 7d). For the common period, dinoflagellates abundances increased by 8 10$^3$ cells L$^{-1}$ (108%, Table 1). However, the DLM seasonality plot indicated that summer seasonal components of dinoflagellate abundances, corresponding to dinoflagellate annual peak, displayed a significant decreasing trend over the common period (Fig. 7f, Table 2).

## 4 Discussion

The sequence of causes and effects between eutrophication in continental aquatic ecosystems and in those located downstream can be studied by observing trends of eutrophication indicators using the same tool and during the same periods. In the present study, eutrophication trajectories in the downstream VB coastal waters during recent decades were examined,

through long-term trends of phytoplankton biomass and nutrient concentrations, in relation to the restoration of the eutrophic Loire and Vilaine Rivers. The DLM analysis provided the opportunity to explore trends and changes in seasonality in a visual manner with figures displaying individual data. The modified non-parametric Mann-Kendall test applied to DLM trend and seasonal components of all variables over common period has permitted corroboration of DLM observations.

Overall results demonstrate that upstream recoveries from eutrophication were accompanied by increased eutrophication downstream. The significant reduction in P input relative to N was not enough to mitigate eutrophication all along this river – coastal marine continuum. More reduction of N input, paying particular attention to diffuse N-sources, is necessary to mitigate eutrophication effectively in the VB coastal waters.

### 4.1 Eutrophication trajectories at the river basin outlet

The decrease in Chl $a$ in pre-estuarine stations on the Loire and Vilaine Rivers over the past decades reflects the global diminution in eutrophication in north American and European rivers (Glibert et al., 2011; Romero et al., 2013). This decrease in Chl $a$ was also observed in the Upper and Middle Loire (Larroudé et al., 2013; Minaudo et al., 2015). However, the Loire did not retrieve its oligotrophic state of the 1930s (Crouzet, 1983). At the studied stations, the annual Chl $a$ peak decreased and shifted from late summer to spring (Figs. 8a, 8b). The parallel decrease of DIP and Chl $a$ in the Loire and

Vilaine Rivers underlines the role of decreasing P in reducing phytoplankton biomass (Descy et al., 2012; Minaudo et al., 2015), as also found in other river systems, such as the Danube (Istvánovics and Honti, 2012), the Seine (Romero et al., 2013), and some Scandinavian rivers (Grimvall et al., 2014). This decreasing trend of DIP is a result of improved sewage treatment, decreased use of P fertilizers and the removal of P from detergents (Glibert, 2010; Bouraoui and Grizzetti, 2011). However, the decline of Chl $a$ in both studied rivers began several years after that of DIP when the latter reached limiting

concentrations for phytoplankton, as deduced at Montjean on the Loire by Garnier et al. (2018). The change in timing of the

annual DIP minima from summer to spring in the Loire and Vilaine Rivers during last decades of the studied period, concomitant with that of the annual peak of Chl *a*, can be explained by the increasingly early depletion of DIP by phytoplankton (see Floury et al., 2012 for the Loire).

The trend of DIN in studied rivers reveals the general trends observed in other large European rivers, showing a slight
decrease, a steady trend or even an increase, depending on the degree of fertilizer application in catchment areas (Bouraoui and Grizzetti, 2011; Romero et al., 2013). The increase in summer Loire DIN since the early 1990s was offset by the decrease in winter values, which is related to the reduction in N point source emissions and N fertilizer application (Poisvert et al., 2016; data from French Ministry of Agriculture, S. Lesaint, pers. comm.). An increase summer DIN of several tens of $\mu$mol L$^{-1}$ was also reported in the Middle Loire (Minaudo et al., 2015). This increase in summer DIN is the result of a
delayed response due to the long transit time of DIN through soils and aquifers in the Loire catchment (up to 14 years; Bouraoui and Grizzetti, 2011). The decreasing DIN uptake by phytoplankton in the Loire, may have also contributed to the increase in summer DIN (Lair, 2001; Floury et al., 2012). Concerning the Vilaine, the slight decrease in DIN from the early 1990s reflects the decrease in N fertilizer application in the Vilaine catchment (Bouraoui and Grizzetti, 2011; Aquilina et al., 2012), which is facilitated by a relatively short transit time of DIN in the Vilaine watershed (~5-6 yr, Molenat and Gascuel-
Odoux, 2002; Aquilina et al., 2012).

DSi data series in both rivers were too short to investigate long-term trends and seasonality, but provided values in order to examine nutrient stoichiometry. Larroudé et al. (2013) observed no significant trend in DSi between 1985 and 2008 in the Middle Loire, as also confirmed at Montjean station by Garnier et al. (2018). The decrease in DIP led to the increasing trend of DIN:DIP ratios, and probably DSi:DIP, in both rivers, as was observed in numerous rivers (Beusen et al., 2016). Based on

these trends, the DIP limitation has been thus reinforced in studied rivers during the last decades, and potentially in receiving coastal waters, regardless of the season.

## 4.2 Eutrophication trajectories in the VB

In contrast to what happened in rivers, eutrophication in the downstream VB coastal waters has worsened during recent decades, as indicated by significant increase in Chl $a$, also confirmed by the significant augmentation of both diatom and dinoflagellate abundances. The increase in Chl $a$ in the VB was accompanied by a shift in its annual peak from spring to summer (Figs. 8c, 8d). This modification in the seasonal course of phytoplankton biomass coincides with the increase in diatom abundances, occurring mainly in summer. The dynamics of phytoplankton in the VB during the last decade of the studied period thus underwent important changes: 1) an increase in biomass, 2) a change in timing of the annual peak from spring to summer, 3) a modification in seasonal course of diatoms and dinoflagellates.

### 4.2.1 Increased Chl $a$

The increase in phytoplankton biomass could result from several causes, namely overfishing, decrease in commercially grown suspension-feeders, increase in temperature, and increase in nutrient inputs. Increased predation on planktonic herbivores could reduce grazing on phytoplankton (Caddy, 2000). In the VB, commercial fishing is banned in order to protect its ecological function as nursery for demersal fish (Désaunay et al., 2006). The decline in fisheries in the Bay of Biscay since the 1990s (Rochet et al., 2005; Lassalle et al., 2012) was unlikely to have caused increased Chl $a$ in the VB, since phytoplankton biomass in these oceanic waters has always been lower than that in the VB (Table S2). Grazing activity by bivalve suspension-feeders can modify phytoplankton biomass (Cloern, 1982; Souchu et al., 2001). In the VB, there was an increase in commercial mussel production (*Mytilus edulis*) between 2001 and 2012 (Le Bihan et al., 2013). This should

have led to depletion in phytoplankton biomass, in fact the opposite trend was observed. In regions where the phytoplankton productivity is limited by light availability, an increase in sea surface temperature can promote phytoplankton growth due to water column stabilization (Doney, 2006; Boyce et al., 2010) and decreased turbidity (Cloern et al., 2014). In the VB, except during winter and high hydrodynamic activity periods, phytoplankton production is limited by nutrients (Guillaud et al., 2008). Therefore, the increase in Chl *a* in the VB was particularly due to enhanced nutrient availability, as also reported in China Sea coastal waters by Wang et al. (2018).

### 4.2.2 Changes in timing of annual Chl *a* peak

Seasonal changes in phytoplankton biomass peaks have been reported in other aquatic ecosystems and mostly attributed to climate change-induced temperature (Edwards and Richardson, 2004; Racault et al., 2017). Variations in nutrient availability can also induce a change in the seasonal pattern of phytoplankton biomass (Thackeray et al., 2008; Feuchtmayr et al., 2012). These authors observed that the advancement in the timing of the spring diatom bloom in some English lakes was related to the increase in winter DIP. In the VB, the shift in annual Chl *a* peak from spring to summer, coupled with the change in position of the annual DIP minima from summer to spring, suggests that DIP depletion by phytoplankton bloom occurred progressively earlier during the last two decades. Based on nutrient concentrations and stoichiometry (Justić et al., 1995), the first nutrient limiting phytoplankton biomass in the VB shifts seasonally from DIP in spring to DIN in summer, as verified by bioassays (Retho et al. Ifremer, unpublished data). The conjunction of the decrease in DIP and an increase in DIN in the VB has probably also contributed to the shift in annual Chl *a*.

### 4.2.3 Role of DSi on seasonal course of diatoms and dinoflagellates

In terms of nutrients, the balance between diatoms and dinoflagellates is predominantly regulated by the DSi availability (Egge and Aksnes, 1992). In the VB, based on nutrient concentrations and stoichiometry, diatoms were rarely limited by the DSi availability, thanks probably to internal DSi regeneration, as suggested by Lunven et al. (2005) and Loyer et al. (2006) in the northern Bay of Biscay continental shelf. The fact that diatoms have increased more than dinoflagellates in the VB, contradicts the idea that excessive DIN and DIP inputs favor phytoplankton species, which do not require DSi (Conley et al., 1993; European Communities, 2009; Howarth et al., 2011). An increase in diatom abundances during the eutrophication process was also observed in Tolo Harbor (Yung et al., 1997; Lie et al., 2011) and the coastal waters of the Gulf of Finland (Weckström et al., 2007). Conversely, decreasing eutrophication in the Seto Inland Sea (Yamamoto, 2003), in Thau (Collos et al., 2009) and other Mediterranean Lagoons (Leruste et al., 2016) was accompanied by the increase in dinoflagellate abundances to the detriment of diatoms. These observations and our results provide evidence that eutrophication can be manifested by an increase in diatom abundances.

### 4.3 Loire/Vilaine - VB continuum

In theory, several external nutrient sources could have contributed to nutrient availability in the VB: atmospheric, oceanic and riverine inputs. DIN inputs from rainwater estimated by Collos et al. (1989) represent only 1% of river inputs, while levels of nutrients and Chl $a$ in the Bay of Biscay always remained low during the studied period (Table S2). The proximity of the VB to the Loire and Vilaine Rivers designates riverine inputs as main external nutrient sources in these coastal waters (Ménesguen et al., 2018a, b).

### 4.3.1 Rivers as the main external nutrient source to the VB

Watersheds, rivers and coastal waters located at their outlet, constitute a continuum in which anthropogenic pollution, generated in watersheds, are transported to coastal zones (Vannote et al., 1980; Bouwman et al., 2013). The transfer of nutrients from continents to coastal waters is largely determined by freshwater inputs, the dynamics of which depend largely on precipitation in watersheds. Trends in the Loire and the Vilaine discharges displayed similar oscillations to those of rivers flowing to the North Sea as reported by Radach and Pätsch (2007), suggesting a common hydro-climatic pattern in Western Europe linked to the North Atlantic Oscillation. The decrease in the Loire discharge observed between 1997 and 2013 was also found in the middle section of the river for the period 1977-2008 (Floury et al., 2012) and attributed essentially to abstraction for irrigation and drinking water by these authors. The strong correlation between Loire and Vilaine discharges underlines the similarities between the two rivers concerning the precipitation regime. However, with a tenfold higher discharge than the Vilaine, the Loire remains the main source of freshwater for the northern Bay of Biscay, with a major role in the eutrophication of coastal waters in south Brittany, including the VB (Guillaud et al., 2008; Ménesguen et al., 2018a, 2019). Aside from flood periods, the closure of the Arzal dam during the low-water periods (Traini et al., 2015), makes nutrient inputs into the VB by the Vilaine negligible in summer, compared to those from the Loire.

### 4.3.2 Role of estuaries and the Vilaine dam

Biogeochemical processes within estuaries may alter the nutrient transfer from rivers to coastal waters (Statham, 2012; Jickells et al., 2014). Coupled nitrification-denitrification and ammonification-anammox can be a sink of N in estuaries (Howarth et al., 1996; Abril et al., 2000). Inorganic nutrients in estuaries can also be removed by phytoplankton uptake, which is nonetheless limited by turbidity (Middelburg and Nieuwenhuize, 2000; Guillaud et al., 2008). Estuaries can also act as a source of nutrients, resulting from mineralization of riverine phytoplankton organic matter (Meybeck et al., 1988;

Middelburg et al., 1996; Etcheber et al., 2007). However, for the studied rivers, this process may have diminished with the decreasing trend in riverine Chl *a*. The desorption of loosely bound P from suspended mineral particles on arrival in saline waters can also provide a source of DIP (Deborde et al., 2007; van der Zee et al., 2007). Except during flood periods, the suspended particle fluxes from the Loire are generally low (Moatar and Dupont, 2016). In addition to these biogeochemical

processes, the increase in population around the Loire estuary (ca. 1% per year, INSEE, 2009) during the last decades may have contributed to the increase in N and P inputs. However, inputs of DIN and DIP from wastewater treatment plants in the Loire and Vilaine estuaries have not increased due to improved treatment techniques (Loire-Brittany River Basin Authority, P. Fera, pers. comm.). The presence of a dam at the river outlet may increase water residence time, thus favoring nutrient uptake by phytoplankton and loss of N via denitrification (Howarth et al., 1996; Seitzinger et al., 2006). Unfortunately, for

these two studied rivers, processes in estuaries and dam are poorly investigated and quantified, which makes it difficult to estimate their influence on nutrient transfer to coastal zone.

Despite influences of estuaries and dam, the increase in DIN:DIP and DSi:DIP ratios in rivers during last two decades, with values already largely above the theoretical value of 16 in the 1990s, has been reflected in the VB coastal waters (Figs. S5, S7). Although biogeochemical processes in estuaries and the Vilaine dam may introduce bias in nutrient transfer

from rivers to the VB, they are probably not intense enough to decouple the observed trends between rivers and the VB, as suggested by Romero et al. (2016) for the Seine River – Seine Bay continuum. Moreover, significant negative correlations between annual Chl *a* medians in the VB and in rivers, as well as significant positive correlations between annual medians of DIN and DSi in the VB with those of river discharge suggest that changes in eutrophication parameters in the VB (i.e., phytoplankton biomass) were related to changes in rivers (Ménesguen et al., 2018a, b).

### 4.3.3 Link between eutrophication trajectories in rivers and in the VB

During the last two decades, the downstream VB coastal waters have received decreasing DIP inputs, increasing DIN inputs especially from the Loire during summer, and no change in DSi inputs (Fig. 8). The decrease in riverine DIP loads was the cause of the simultaneously decreasing trend in the VB DIP and may have reinforced spring DIP limitation as also reported

by Billen et al. (2007) in the Seine Bay. The worsening eutrophication in the VB was the consequence of increasing DIN inputs from the Loire. A similar observation was reported in other coastal ecosystems, such as the Neuse River estuary (Paerl et al., 2004), Belgian coastal waters (Lancelot et al., 2007), and the Seine Bay (Romero et al., 2013), where decreasing upstream Chl $a$, due to DIP input reduction, was accompanied by the increase in downstream Chl $a$, as a result of increasing DIN input. The seasonal change in annual Chl $a$ peak in the VB resulted also from the conjunction of decreasing DIP loads

and increasing summer DIN loads from the Loire. The summer limitation of phytoplankton production by DIN in the VB cannot be explained by the stoichiometry of nutrients in rivers. Internal sources of nutrients, especially sediments (see below), were also likely to support a significant portion of nutrient availability for phytoplankton production during the period of low river discharge (Cowan and Boynton, 1996; Pitkänen et al., 2001).

### 4.3.4 Role of internal nutrient loads

In shallow ecosystems, internal nutrient recycling can regulate phytoplankton production and potentially exacerbate eutrophication (Paerl et al., 2016), as observed both in lakes (Jeppesen et al., 2005) and coastal ecosystems (Pitkänen et al., 2001). Compared to freshwater, the fragility of marine ecosystems is related to salinity (Blomqvist et al., 2004). The presence of sulfate a major element of salinity) decreases the efficiency of sediments to retain DIP (Caraco et al., 1990; Lehtoranta et al., 2009) and favors the recycling of DIP over DIN, the latter being potentially eliminated through

denitrification (Conley, 2000; Conley et al., 2009). In the VB, measurements of benthic nutrient fluxes confirm that

sediments represent a substantial DIP and DSi source compared to riverine inputs (Ratmaya, 2018), allowing summer phytoplankton production to benefit from surplus DIN inputs from the Loire. Sediments were then able to support phytoplankton production by providing DIP and DSi, as found in other coastal ecosystems (Cowan and Boynton, 1996; Boynton et al., 2008), and probably to switch the first limiting nutrient from DIP in spring to DIN in summer, as observed in

the Baltic Sea (Conley, 2000; Pitkänen et al., 2001). Consequently, the increase in summer diatom abundances in the VB was mainly due to increased summer DIN loads from the Loire, sustained by internal sources of DIP and DSi coming from sediments.

## 4.4 Implications for nutrient management

### 4.4.1 Impact of nutrient management strategies

The need to control both N and P inputs to mitigate eutrophication along the freshwater-marine continuum is still debated within the scientific community (see Schindler et al., 2008; Conley et al., 2009; Schindler, 2012; Paerl et al., 2016; Schindler et al., 2016). Despite the imbalance between P and N input reduction, eutrophication in the river section of the Loire/Vilaine – VB continuum has diminished but the increase in phytoplankton biomass in the VB provides evidence that significant reduction of P inputs, without concomitant N abatement, was not yet sufficient to improve water quality along the entire

continuum. Targeting N and P pollution from point sources has successfully reduced eutrophication in marine ecosystems, as evidenced in Tampa Bay (Greening and Janicki, 2006) and in several French Mediterranean lagoons (Collos et al., 2009; Leruste et al., 2016). However, N pollution in coastal waters from rivers with watersheds largely occupied by intensive agriculture remain problematic in many European countries (Bouraoui and Grizzetti, 2011; Romero et al., 2013). Reducing diffuse N inputs through improved agricultural practices and structural changes in the agro-food system (Desmit et al., 2018;

Garnier et al., 2018) would probably help to lessen eutrophication (Conley et al., 2009; Paerl, 2009). Assuming that rapid

and radical change in farming practices is implemented, the delayed responses due to variations in transit time of $NO_3^-$ in aquifers should be taken into account for restauration strategy (Bouraoui and Grizzetti, 2011).

In the VB, a reduction in DIN inputs especially during the summer would probably have prevented eutrophication from worsening in this ecosystem. Given that in many other coastal ecosystems the first nutrient limiting phytoplankton
production tends to switch from DIP in spring to DIN in summer (Fisher et al., 1992; Del Amo et al., 1997; Conley, 2000; Tamminen and Andersen, 2007), it would be relevant to take into account seasonal aspects for nutrient reduction strategy.

### 4.4.2 Influence of internal nutrient regeneration

In the VB, the internal nutrient recycling from sediments appears to have contributed to the worsening of summer water quality during the last two decades and augmented the effects of anthropogenic nutrient inputs. Internal nutrient loads can
delay ecosystem recovery from eutrophication following external nutrient input reduction (Duarte et al., 2009). In lakes, this delay induced by internal loads of P on the oligotrophication process varies from 10 to 20 years (Jeppesen et al., 2005; Søndergaard et al., 2007). In coastal ecosystems, the delay resulting from internal nutrient loads was less studied. However, Soetaert and Middelburg (2009), using a model in a shallow coastal ecosystem, estimated a delay of more than 20 years following the reduction of external N input. Therefore, for the Loire/Vilaine – VB continuum, nutrient management
strategies should consider the internal nutrient loads in order to anticipate the delay in recovery of the VB coastal waters from eutrophication.

## 5. Conclusions and perspectives

Parallel investigation of eutrophication parameters in the Loire and Vilaine Rivers, and coastal waters under their influence revealed several striking patterns and relationships, of which the most apparent was upstream recoveries from eutrophication accompanied by increased eutrophication downstream (Fig. 8). During the last two decades, Loire-Vilaine coastal waters have experienced a diminution in DIP inputs, whereas DIN continued to increase in the Loire during summer. While the decreasing trends in DIP were accompanied by declining phytoplankton biomass in rivers, the seasonal cycle of phytoplankton has been changed in downstream VB, with an increase in biomass, a shift in its annual peak from spring to summer, and a modification in the seasonal course of diatoms and dinoflagellates. Moreover, the concept of diatom replacement by dinoflagellates during the eutrophication process does not seem to be applicable to all shallow coastal ecosystems.

These results open up a whole field of investigation into the effects of changes in the phytoplankton dynamics on food webs, which is of major importance to this flatfish nursery and commercial shellfish area (Désaunay et al., 2006; Chaalali et al., 2017). Further studies are necessary to investigate the modifications in the phytoplankton community, especially the phenology of the different species, as well as the possible consequence on food webs. Finally, the internal loads of nutrients from sediments are suspected of counteracting the reduction of external nutrients, thus delaying the restoration progress. During the eutrophication process, sediments may also play an important role in the balance between diatoms and others classes of phytoplankton. Taking into account these internal processes in modelling studies (i.e., ECO-MARS3D, Ménesguen et al., 2018a, b; Ménesguen and Lacroix, 2018), will better simulate nutrient load scenarios in shallow coastal bays (work in progress).

**Data availability**

All data used in this study are available in the following online data bases: French National Observation Network for Phytoplankton and Hydrology in coastal waters (https://doi.org/10.17882/47248), French Oceanographic Cruises PELGAS surveys (http://campagnes.flotteoceanographique.fr/series/18/), Loire-Brittany River Basin Authority (http://osur.eau-loire-bretagne.fr/exportosur/Accueil), French hydrologic database (http://www.hydro.eaufrance.fr/), ICES Oceanographic database (http://ocean.ices.dk/HydChem/HydChem.aspx?plot=yes).

**Author contribution**

PS and WR designed the study. WR compiled and prepared the datasets. DS performed statistical and time series analyses. WR wrote the manuscript with contributions from all co-authors (PS, DS, JSM, NCL, EG, FAL, LB). Author abbreviations: WR = Widya Ratmaya, DS = Dominique Soudant, JSM = Jordy Salmon-Monviola, NCL = Nathalie Conchennec-Laureau, EG = Evelyne Goubert, FAL = Françoise Andrieux-Loyer, LB = Laurent Barillé, PS = Philippe Souchu.

**Competing interests**

The authors declare that they have no conflict of interest.

**Acknowledgements**

This study was funded by The Loire – Brittany Water Agency (AELB). The authors are grateful to Sylvain Jolly from AELB for providing datasets of the Vilaine and the Loire. The authors thank IFREMER-LER/MPL staff for their technical contributions, especially Karine Collin, Yoann Le Merrer, Mireille Fortune, Michël Retho, Raoul Gabellec, Jacky Chauvin,

Isabelle Truquet and Anne Schmitt. We thank Alice Mellor for the proof reading. The authors are grateful to Drs. F. Gerald Plumley and Anniet M. Laverman for kindly reviewing before submission. Authors acknowledge IFREMER and the Regional Council of the Région des Pays de la Loire for the funding of W. Ratmaya' PhD.

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

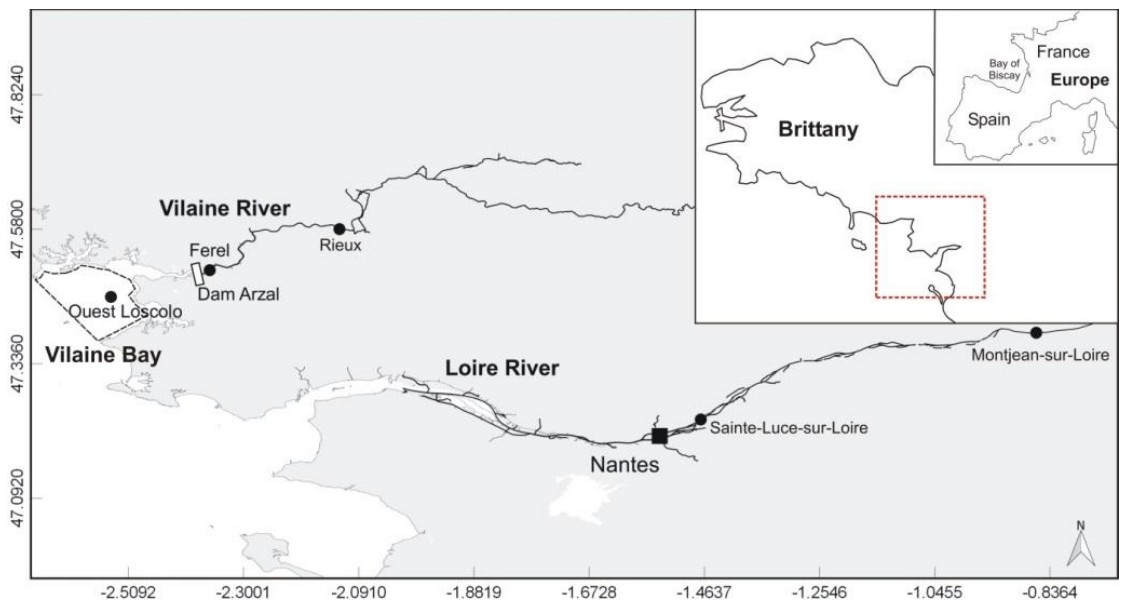

**Figure 1: Map of the area studied showing Loire and Vilaine rivers and delimitation of Vilaine Bay (inset red dotted line). Black dots mark the sampling and gauging stations cited**

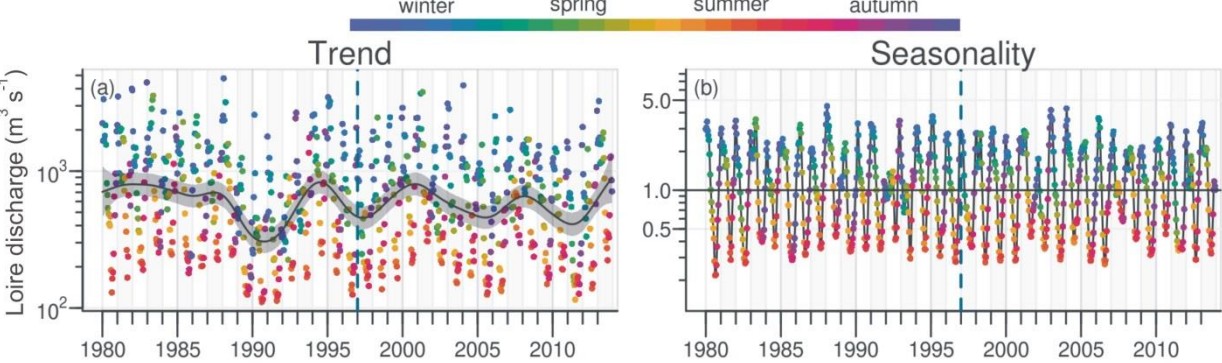

Figure 2: Long-term trend and seasonality of the Loire discharges (a, b). Dark grey lines represent DLM trends. Shaded areas indicate the 90 % confidence interval. Each dot in the trend plot (left) represents an observed value, those in the seasonality plot (right) represent values estimated by the model. On the seasonality plot, the horizontal line (y = 1.0) indicates seasonal components for which fitted values equal to the trend. Dashed vertical blue line corresponds to the longest common period for all studied variables in rivers and in the VB

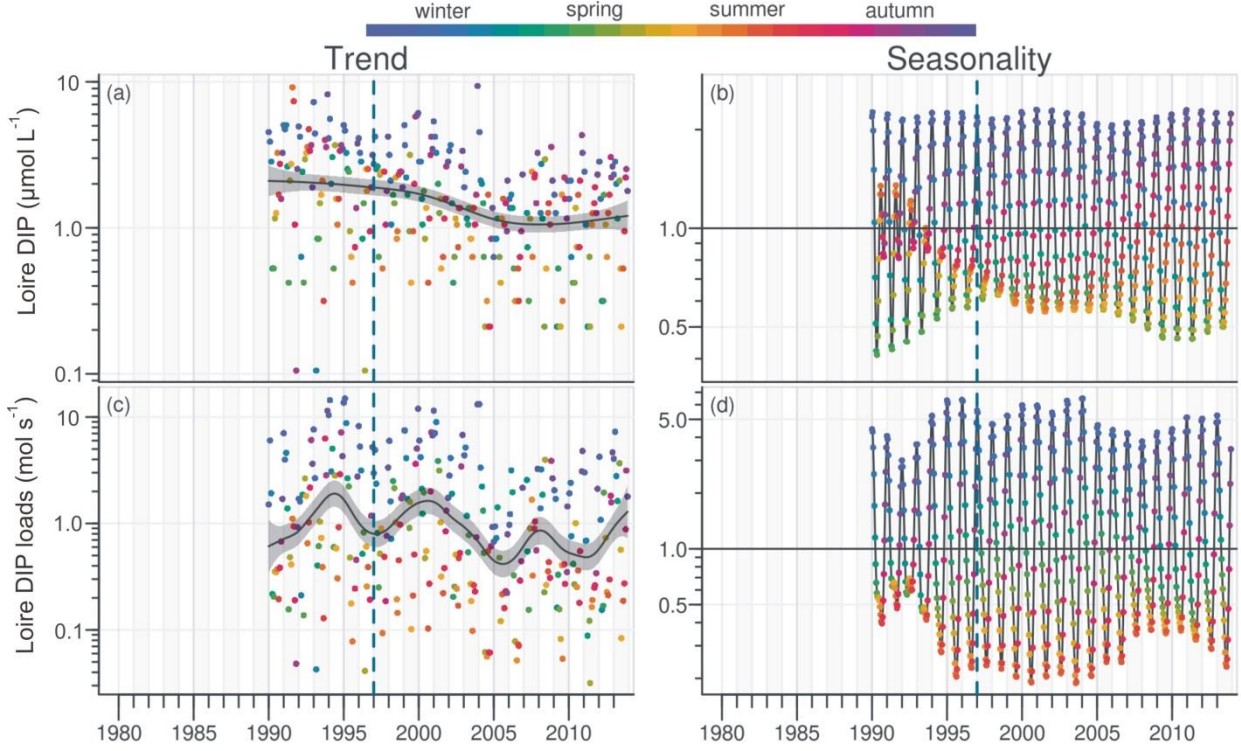

**Figure 3: Long-term trend and seasonality of DIP in the Loire (a, b) and DIP loads from the Loire (c, d). See Fig. 2 for details**

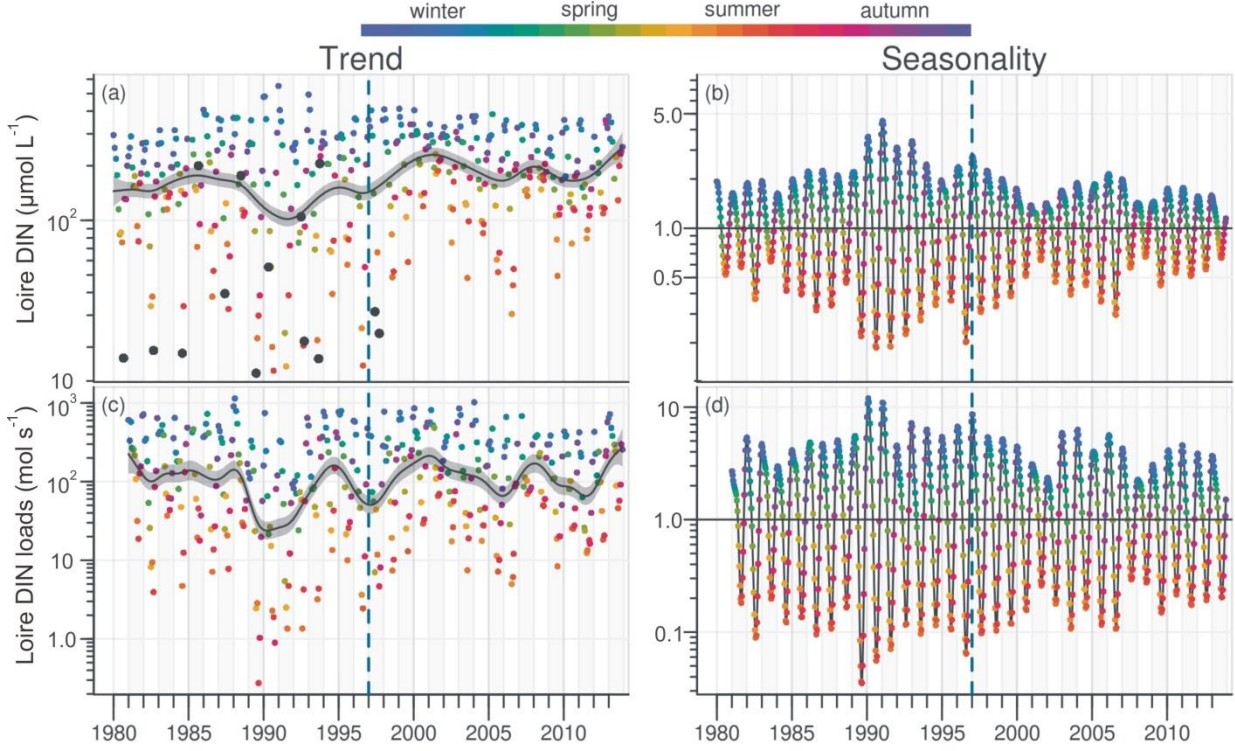

**Figure 4: Long-term trend and seasonality of DIN in the Loire (a, b) and DIN loads from the Loire (c, d). Black dots represent data considered as outliers (see Section 2.4.). See Fig. 2 for details**

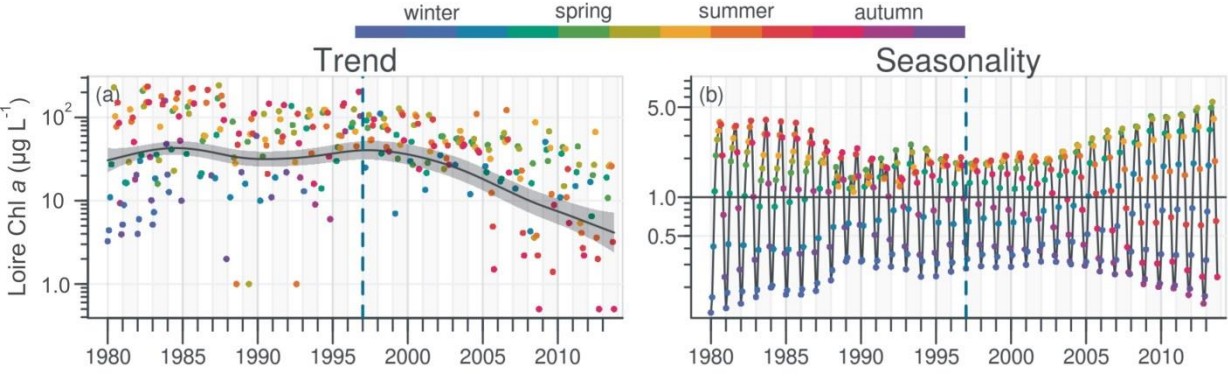

**Figure 5: Long-term trend and seasonality of Chl *a* in the Loire (a, b). See Fig. 2 for details**

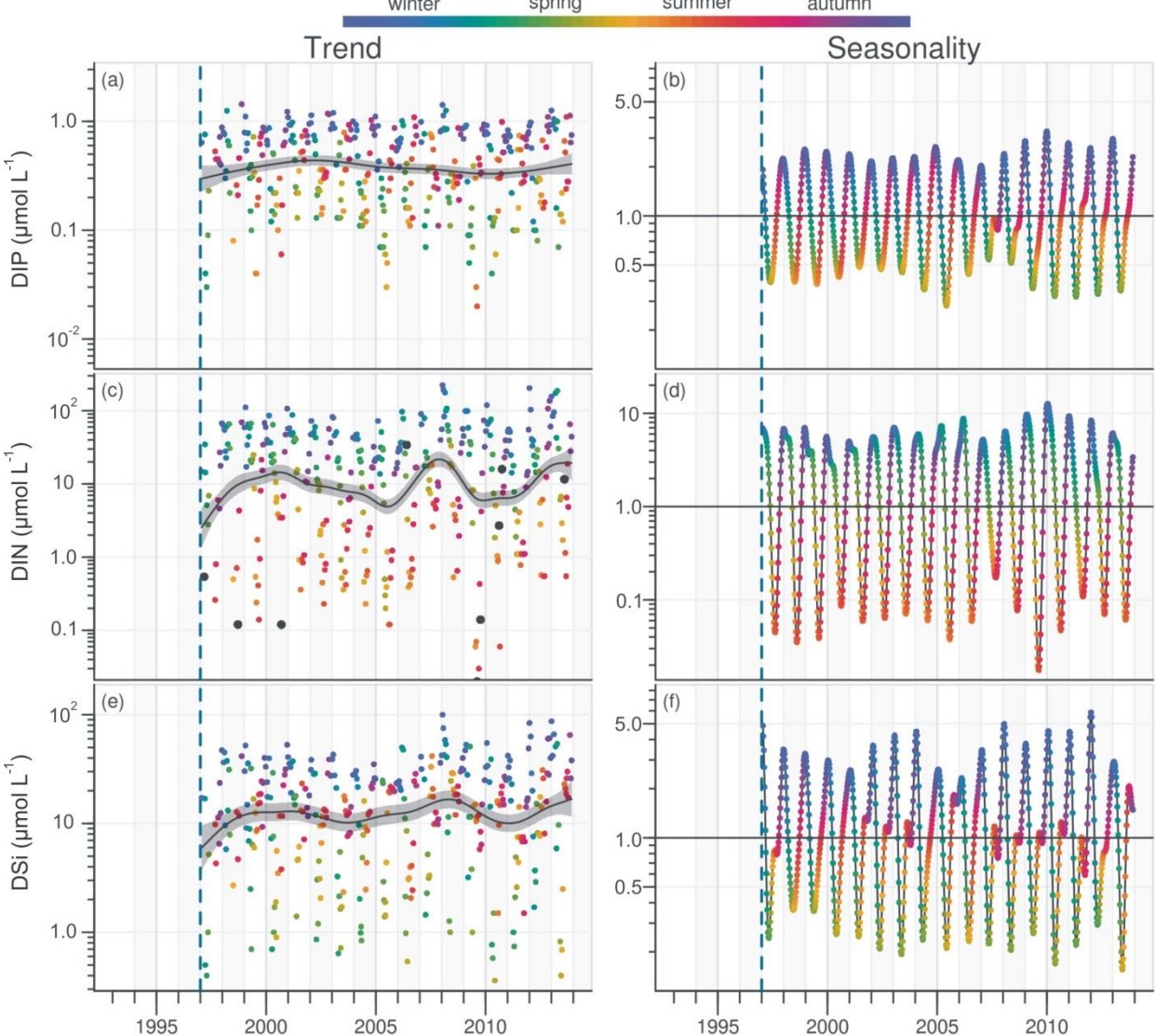

**Figure 6: Long-term trend and seasonality of DIP (a, b), DIN (c, d) and DSi (e, f) in the VB. Black dots represent data considered as outliers (see Section 2.4.). See Fig. 2 for details**

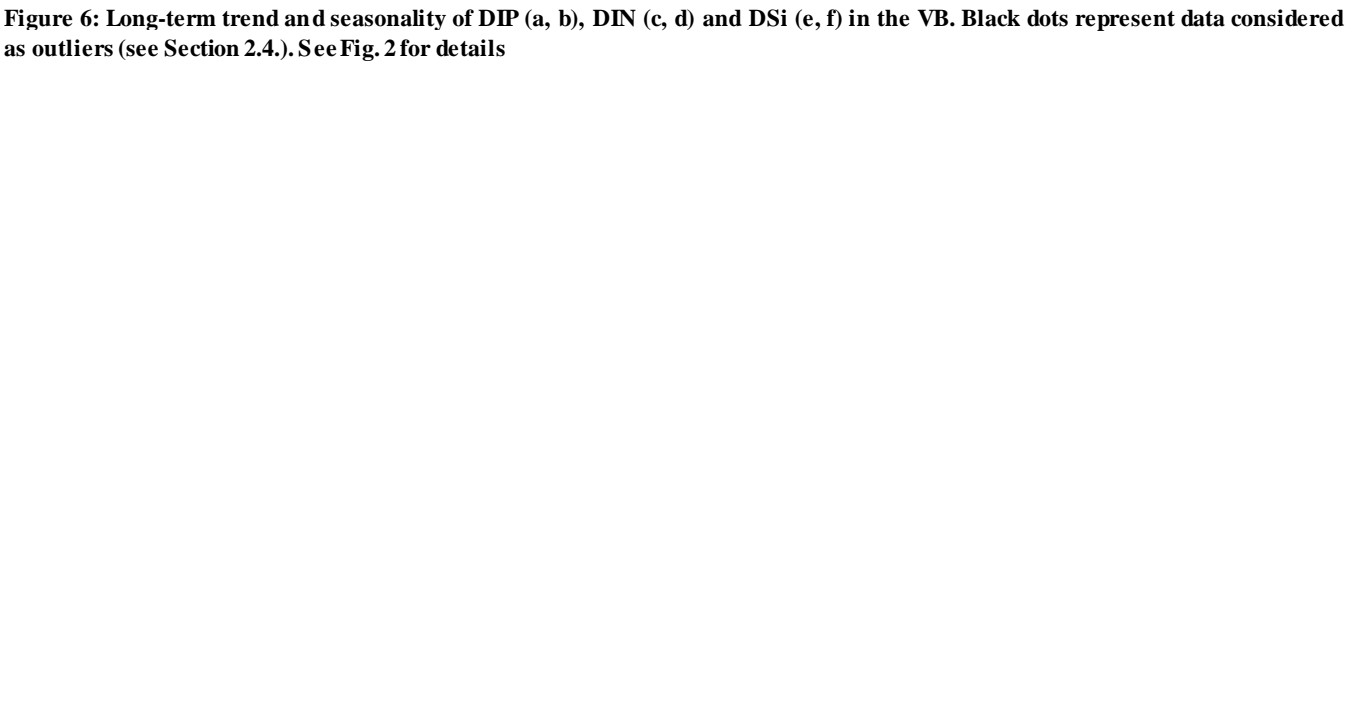

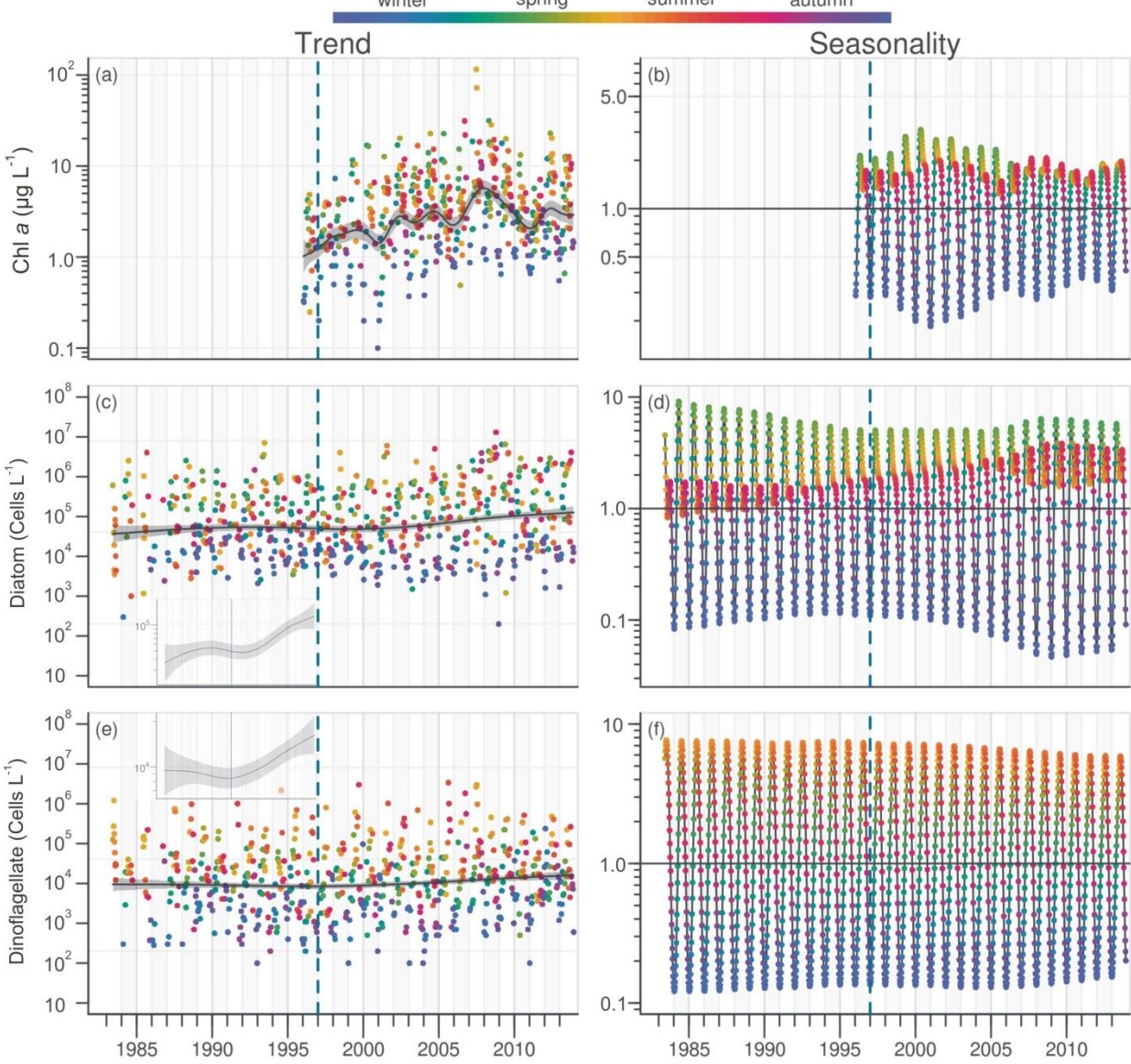

**Figure 7: Long-term trend and seasonality of Chl *a* (a, b), diatom (c, d) and dinoflagellate (e, f) in the VB. Insets display trends of diatom and dinoflagellate abundances with optimal scale. See Fig. 2 for details**

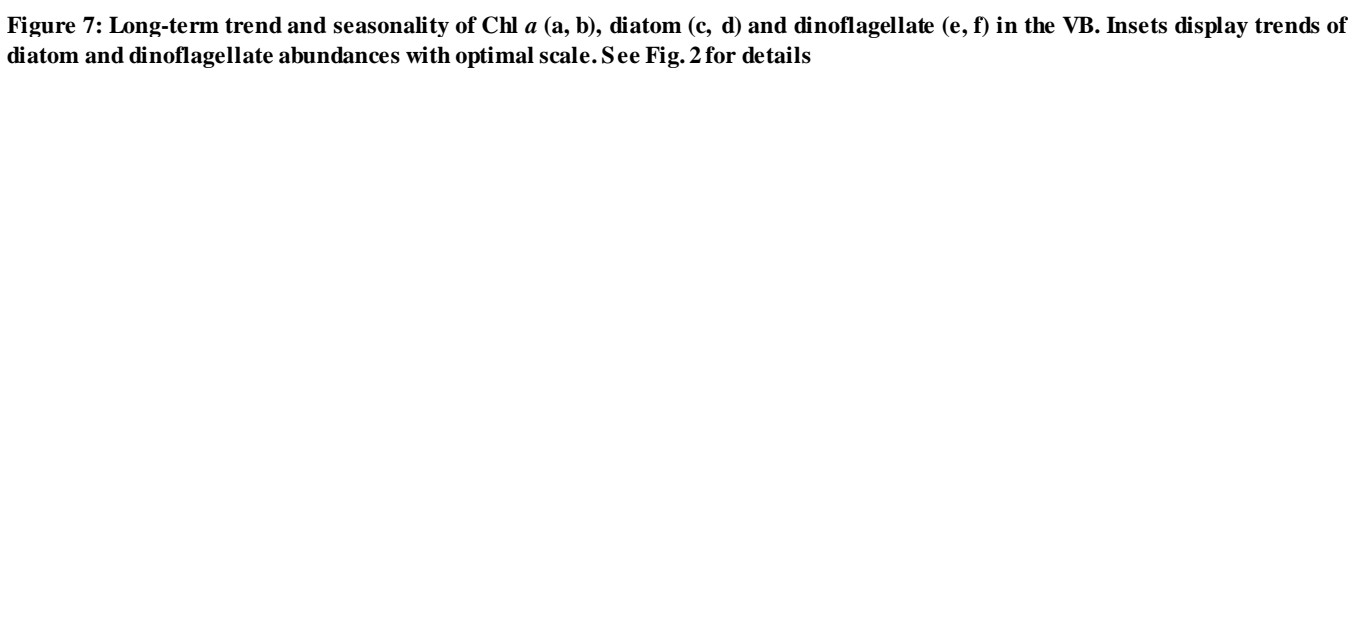

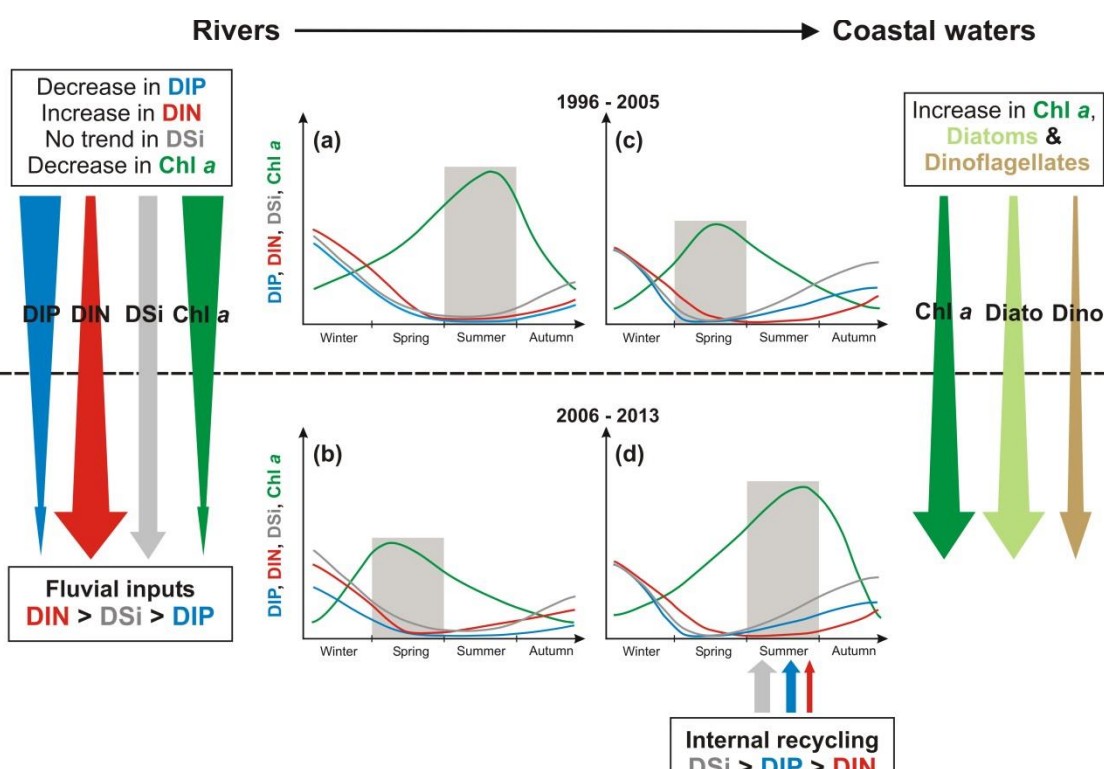

**Figure 8: Graphical representation of the major changes in nutrient concentrations and phytoplankton in river (a, b) and the VB coastal waters (c, d) for the period 1996-2005 (top) and 2006-2013 (bottom). Downward arrows and curves, representing respectively long-term trends and seasonal courses of eutrophication parameters in rivers and in the VB, were fitted according to results. Shaded areas underline the season of maximum Chl *a*. Internal benthic nutrient inputs (upward arrows) were fitted according to the measurement of benthic fluxes in summer 2015 (Ratmaya, 2018)**

**Table 1: Statistical results from Mann-Kendall test performed DLM trend components of eutrophication parameters in rivers and in the VB coastal waters for the common period 1997-2013. If the test was significant at $p<0.05$, differences of the Sen's robust line between the beginning and the end of the period (17 years) were calculated. Values in parentheses are percentages of changes relative to the initial values of the Sen's robust line. Increasing or decreasing trends are indicated by + and − signs respectively. Cells were left blank when tests were not applicable. NS = non-significant**

| Site | Discharge ($m^3 s^{-1}$) | | DIP ($\mu mol\ L^{-1}$) | | DIP loads ($mol\ s^{-1}$) | | DIN ($\mu mol\ L^{-1}$) | | DIN loads ($mol\ s^{-1}$) | | DSi ($\mu mol\ L^{-1}$) | | Chl $a$ ($\mu g\ L^{-1}$) | | Diatoms (Cells $L^{-1}$) | | Dinoflagellates (Cells $L^{-1}$) | |
|---|---|---|---|---|---|---|---|---|---|---|---|---|---|---|---|---|---|---|
| | $p$ | Change (%) | $p$ | Change (%) | $p$ | Change (%) | $p$ | Change (%) | $p$ | Change (%) | $p$ | Change (%) | $p$ | Change (%) | $p$ | Change (%) | $p$ | Change (%) |
| Loire | 0.01 | − 94 (16%) | <0.001 | − 0.85 (47%) | <0.001 | − 0.60 (52%) | 0.63 | NS | 0.42 | NS | | | <0.001 | − 54 (93%) | | | | |
| Vilaine | 0.02 | − 8.7 (23%) | <0.001 | − 1.9 (75%) | <0.001 | − 0.09 (88%) | <0.001 | − 71 (21%) | <0.001 | − 4.6 (38%) | | | <0.001 | − 12 (76%) | | | | |
| VB | | | <0.001 | − 0.05 (13%) | | | 0.01 | + 3.2 (40%) | | | <0.001 | + 3.6 (34%) | <0.001 | + 2.1 (126%) | <0.001 | $+ 90*10^3$ (227%) | <0.001 | $+ 8*10^3$ (108%) |

**Table 2: Statistical results of modified Mann-Kendal test performed on DLM seasonal components of eutrophication parameters in rivers and in the VB for the common period 1997–2013. If the test was significant at *p*<0.05, percentages of changes relative to the initial values of the Sen's robust line were calculated. Increasing or decreasing trends are indicated by + and − signs respectively. Cells were left blank when tests were not applicable. NS = non-significant**

| Site/ Season | Discharge ($m^3 s^{-1}$) | | DIP ($\mu mol\ L^{-1}$) | | DIP loads ($mol\ s^{-1}$) | | DIN ($\mu mol\ L^{-1}$) | | DIN loads ($mol\ s^{-1}$) | | DSi ($\mu mol\ L^{-1}$) | | Chl *a* ($\mu g\ L^{-1}$) | | Diatoms ($Cells\ L^{-1}$) | | Dinoflagellates ($Cells\ L^{-1}$) | |
|---|---|---|---|---|---|---|---|---|---|---|---|---|---|---|---|---|---|---|
| | *p* | % | *p* | % | *p* | % | *p* | % | *p* | % | *p* | % | *p* | % | *p* | % | *p* | % |
| **Loire** | | | | | | | | | | | | | | | | | | |
| Winter | 0.63 | NS | 0.04 | − 23% | <0.01 | − 41% | 0.02 | − 24% | <0.01 | − 40% | | | <0.001 | + 190% | | | | |
| Spring | 0.50 | NS | <0.001 | − 28% | 0.02 | − 33% | 0.21 | NS | 0.49 | NS | | | <0.001 | + 283% | | | | |
| Summer | 0.60 | NS | <0.001 | + 33% | <0.001 | + 59% | <0.01 | + 55% | 0.01 | + 69% | | | 0.09 | NS | | | | |
| Autumn | 0.98 | NS | <0.01 | + 35% | 0.26 | NS | 0.29 | NS | 0.92 | NS | | | <0.001 | − 82% | | | | |
| **Vilaine** | | | | | | | | | | | | | | | | | | |
| Winter | 0.23 | NS | 0.02 | − 17% | 0.07 | NS | 0.90 | NS | 0.11 | NS | | | <0.01 | + 97% | | | | |
| Spring | 0.93 | NS | 0.06 | NS | 0.07 | NS | 0.99 | NS | 0.56 | NS | | | <0.001 | + 63% | | | | |
| Summer | 0.26 | NS | <0.001 | + 9.4% | 0.09 | NS | 0.29 | NS | 0.28 | NS | | | <0.001 | − 41% | | | | |
| Autumn | 0.97 | NS | 0.51 | NS | 0.40 | NS | 0.66 | NS | 0.69 | NS | | | 0.01 | − 44% | | | | |
| **VB** | | | | | | | | | | | | | | | | | | |
| Winter | | | 0.73 | NS | | | 0.03 | + 32% | | | 0.329 | NS | 0.11 | NS | 0.85 | NS | 0.05 | NS |
| Spring | | | <0.001 | − 30% | | | 0.10 | NS | | | 0.086 | NS | <0.001 | − 36% | 0.93 | NS | 0.83 | NS |
| Summer | | | <0.001 | + 80% | | | 0.17 | NS | | | 0.085 | NS | 0.19 | NS | <0.001 | + 43% | <0.001 | − 23% |
| Autumn | | | 0.94 | NS | | | 0.76 | NS | | | 0.647 | NS | 0.37 | NS | 0.27 | NS | 0.87 | NS |

Table 3: Spearman's rank correlations between annual median values of river discharge, nutrient concentrations and phytoplankton biomass in the Loire, Vilaine and the VB for the common period ($n = 17$). Asterisks designate significant correlations (\*\*\*$p<0.001$, \*\*$p<0.01$, \*$p<0.05$)

| | Loire discharge | Vilaine discharge | DIN Loire | DIP Loire | Chl a Loire | DIN Vilaine | DIP Vilaine | Chl a Vilaine | DIN VB | DIP VB | DSi VB | Chl a VB |
|---|---|---|---|---|---|---|---|---|---|---|---|---|
| Loire discharge | 1.00 | | | | | | | | | | | |
| Vilaine discharge | 0.88\*\*\* | 1.00 | | | | | | | | | | |
| DIN Loire | 0.52\* | 0.39 | 1.00 | | | | | | | | | |
| DIP Loire | 0.51\* | 0.43 | 0.44 | 1.00 | | | | | | | | |
| Chl a Loire | -0.08 | -0.06 | 0.25 | 0.35 | 1.00 | | | | | | | |
| DIN Vilaine | 0.33 | 0.47 | 0.02 | 0.55\* | 0.59\* | 1.00 | | | | | | |
| DIP Vilaine | 0.16 | 0.24 | 0.23 | 0.77\*\* | 0.65\* | 0.54 | 1.00 | | | | | |
| Chl a Vilaine | -0.21 | -0.28 | 0.31 | 0.20 | 0.64\*\* | 0.04 | 0.35 | 1.00 | | | | |
| DIN VB | 0.78\*\* | 0.74\*\* | 0.36 | 0.35 | -0.10 | 0.29 | -0.01 | -0.20 | 1.00 | | | |
| DIP VB | 0.13 | -0.09 | 0.07 | 0.38 | 0.05 | 0.11 | 0.29 | 0.19 | -0.12 | 1.00 | | |
| DSi VB | 0.55\* | 0.41 | 0.35 | 0.08 | -0.48 | -0.17 | -0.51 | -0.31 | 0.63\* | -0.02 | 1.00 | |
| Chl a VB | 0.11 | 0.17 | -0.14 | -0.48 | -0.61\* | -0.34 | -0.58\* | -0.50\* | 0.25 | -0.45 | 0.33 | 1.00 |