# Peer review of "Reduced phosphorus loads from the Loire and Vilaine Rivers were accompanied by increasing eutrophication in Vilaine Bay (South Brittany, France)"

_Biogeosciences, 2018_

## Referee Comment (RC1) · C. Minaudo (Referee) · 5 Oct 2018

——————— General comments ——————— This study by Ratmaya et al., focuses on eutrophication trajectories over three decades in a large bay located in the French Atlantic coastal zone. It tries to link long term trends and seasonal evolutions in the main bay tributaries with the ones observed in the bay itself, based on a trend+seasonalilty time series decomposition algorithm. This study could be of interest for Biogeosciences readers, but suffers from too many issues such as lack of a clear research question, lack of structure within and between sections, and several

technical issues that need to be addressed before it can be considered for publication. The main issues to me are the following: - Concentration time series in the Loire River (the main tributary) originate from a station located in a river section under estuarine influence but was considered as representative of the freshwater part. - Methodology is not clear, especially for the seasonal analysis using the DLM approach. Authors need to define clearly the metrics that were used in this work (e.g. little is said on MK slopes p-values although they appear in Tables) - Nothing is presented on the impact of estuarine zones on DIN and NIP, disabling the credibility of the interpretations made to explain eutrophication trajectory in the coastal zone. - If the presence of a dam at the outlet of one of the two tributaries is mentioned, nothing is explained on the potential impacts this should have on the nutrient dynamics discharged into the bay Additionally, this manuscript needs language editing. Many sentences need to be either removed or modified for the sake of clarity. I decided to focus on specific comments on Method and Results sections, because I think interpretation in the Discussion section might change once everything has been addressed properly.

——————————— Specific comments ——————————— Page 2; Lines 28-29 (2;28-29): this hypothesis has been proven wrong in many studies. I don't think you should present your problematic this way. Page 3, section 2.2: explain that you extracted the longest records available. The reader doesn't know at this point that multi-decadal data is available. Page 3, section 2.2: If Montjean is considered as the last freshwater station on the Loire, why would you use concentrations originating from Ste Luce located in the zone influenced by estuarine salinity? This is a choice that could mislead your interpretations. Also, when computing loads, which site served as the reference? That means did you calculate loads at Montjean or Ste Luce and how did you proceed (e.g. catchment areas ratio?)? Page 4, Line 4-5 (4;4-5): you should make sure this assumption on NO3 being >90% TN is correct. For the riverine part, AELB also provides TN concentrations. 4;6: this method for load calculations is subject to large errors, especially on DIP. You should use a discharge weighted method, commonly used by our community, and recommended within OSPAR convention. 4;24: residuals

as white noise is an hypothesis that is not always met by these algorithms. Please, remove "white noise" in this sentence. We need a metric to assess if your algorithm performs well or note, especially when working at the seasonal scale with variables that don't have stable seasonality patterns (e.g. phytoplankton biomass). 5;1: why was this log-transformation necessary? It needs justification. 5;7: reading log-transformed units is not convenient for the reader. You can log-transform the axis but still present actual values. Why did you log-transform the data in the first place? It makes the trend observation less clear to the reader. The authors decided to use units that are consistent in the manuscript, but not commonly used by researchers on lotic environment. Please, convert all mol/L into mg/L or $\mu$g/L 5;8-9: the explanation on trends significativity test is not clear nor properly justified. You need a metric for this. Why not use Sen's Slope significativity test? 5;11: the authors should define clearly which metrics were extracted for the time-series analysis and used for further analysis. 5;15: have you conducted MK test on de-seasonalized = observations – seasonal component, or on de-seasonalized = trend component? The latter discards residuals from the analysis and this choice should be justified. Also, I think residuals from your DLM algorithm should be plotted along. 5;17: it is not clear how you proceeded to identify seasonal trends. Did you use a seasonal MK test? This needs more details since it is the core of your analysis. Besides, how would you justify analyzing loads evolutions and not only concentrations since you show that Q was stable over time? Removing all the load trajectory description would save space for other elements in your paper, and benefit to the clarity of your messages. 5;21: What is STATGRAPHIC CENTURION and what are the metrics/analysis conducted with this? Please, add a reference for this. 5;26: how significant is this trend in Q data? A large slope in MK tests doesn't mean that it is statistically significant. 6;5: this seasonal shift is not observable in Figure 3. Consider adding a Figure to show seasonal variations and evolutions. In the Result section, it is good to refer to Tables and Figures, but the reader also needs actual values included in the text, otherwise he always has to go back and forth from text to Table/Figure. 6;12: please, be more specific, and always use similar ways of describing the data: first,

trends. Second, seasonal variations. It helps increasing the clarity of the manuscript and makes things easier for the reader. You have too many additional figures. Please, make a selection of the ones that are really useful to support your ideas. 7;19: add a section for this correlation analysis 7;24-29: Do you believe your DLM analysis is suitable for phytoplankton biomass description at the seasonal scale? You need to validate this first, and plots in Figure 7 don't help answer this question if you don't show residuals (you'll see that they don't look like white noise). Insets in Figure 7 are not explained. It has to be. Section 4: this section could be reorganized as follows 1) Nutrients and Chl-a trends at river basin outlets 2) Nutrients and Chl-a trends in the bay 3) River to bay continuum 4) Implications for management How does the estuarine zone could interfere in your interpretations? Same question with the presence of a dam at the outlet of the Vilaine river? This needs to be addressed, at list by listing the different processes that occur. Many has been done on the subject. 8;21: This can't be said like this. At the outlet of large and intensively managed catchments, nutrients variations are co-controlled by upstream hydrological variations, delivery to stream modalities (point or diffuse sources?), and by instream retention processes through physical and biogeochemical processes. 8;22: You should mention the North Atlantic Oscillation to explain the 7 years cycles. See also Dupas et al., 2018 (WRR) Figure 8 could be a great final figure, but needs to be explained once the processes explaining the different patterns in eutrophication metrics are completely described.

——————————— Technical comments ——————————— Page 1, Line 14 (1;14): remove Âń(i.e., phytoplankton biomass) Âż, as eutrophication expression is not only phytoplankton excessive biomass. 2;5-7: this has to do with different source types and it should be explained. Environmental measures to tackle P were successful because P largely originated from point sources with limited legacy effects in the streams. For N, diffuse sources dominate and there is large legacy effect. 2;7-11: you should also mention freshwater ponds and lakes were eutrophication is still severe despite large P reductions. 2;16-17: I'd remove the codes for what you called "water masses". Do the authors mean "water body"? 2;18: an actual scientific reference would be better. 3;3:

"widest" is not correct. You may refer to "largest river basin". 3;4: sentence is not clear, please, rephrase it. 3;22: sentence is not clear, please, rephrase it. 5;1: the use of ":" separates the sentence in a way that makes it hard to understand. Please, modify. 5;8-9: check for use of different tenses throughout the manuscript. 5;23: Change this section title to "Discharge and nutrients long term trends in freshwater basin outlets"

---

## Referee Comment (RC2) · Anonymous Referee #2 · 9 Nov 2018

This paper studies the long term trends in nutrient and phytoplankton dynamics in the Loire and Vilaine rivers, and in the Vilaine Bay (VB). The authors discuss changes in eutrophication of these systems, and relate changes in the VB to those in nutrient inputs from the two rivers. They show that, even though phytoplankton blooms decreased in the riverine systems following reduction in dissolved inorganic P, phytoplankton biomass in the VB has continued to increase. This could be fueled by nitrogen delivery from the rivers (slightly increasing trend for the Loire), together with phosphorus and silica recycling from bottom sediments in the coastal area. This is an

interesting discussion point, that totally fits Biogeosciences' scope. This is however only superficially discussed, and the layout of the paper makes it difficult to identify the main conclusions. I also noted important gaps in the methods' description.

The presentation of the river trajectories is extensive, but was already thoroughly discussed in a previous study (Minaudo et al., 2015). Very complete time series of Chla concentrations and abundances of different phytoplankton species in the VB are presented, and could be extremely valuable to examine changes in community structure. However, these are not discussed in depth. Moreover, more elements should be provided to the reader to justify that the data presented here is enough to support the conclusions of the study. In fact, interpretations of the dynamics in the VB are derived from observations at a single point, at which the influence of the Loire river is not obvious and not discussed.

I believe these major shortcomings should be addressed before this work can be published.

—— General comments ——

1. More information on the influence of the Loire river on the VB dynamics is needed. In fact, nutrients need to travel more than 120km from the Loire river monitoring station (Saint Luce sur Loire) to the Bay, through the Loire estuary and along the coast. Do coastal currents carry most of the Loire river's exports to the VB? How can processing in the estuary and along the coast impact loads reaching the VB?

2. Methods on the Dynamic Linear Models (DLM) and Mann-Kendall (MK) test analysis are not detailed enough. I am also not convinced that the MK test provides any more information than the DLM analysis. To my understanding, numerical estimates on trends and seasonal variations can also be extracted from the latter. Using these two methods to come up with the same interpretations waters down important messages in the results and discussion sections.

3. Authors refer several times throughout the manuscript to "management scenarios focused solely on P reduction" or on "P alone". However, this is not totally accurate for the study area, and should be moderated. Even though ecosystems responded quicker to P reduction strategies (e.g. for point sources) than to policies on agricultural fertilization, those already exist (e.g. EU Nitrates Directive).

4. In general, statements are sometimes vague or not totally accurate. The structure of the results and discussion sections makes it difficult for the reader to identify the main conclusions of the study.

These points, together with more minor concerns, are more detailed hereafter, in the specific comments.

—— Specific comments/scientific questions ——

1. L7-8, P2. "This result is consistent with the idea that reducing P alone, and not N, can mitigate eutrophication of freshwater systems (Schindler et al., 2008)": This paper from Schindler et al. does not show this; they study the effect of reducing N only. Moreover, this is not a scientific consensus (e.g. Pearl et al., 2016, Environ. Sci. Technol. 50, pp 10805–10813). This sentence should be moderated.

2. L14-15, P2. "Nutrient inputs. . . control phytoplankton production in coastal waters of the northern Bay of Biscay": Riverine inputs constitute the major nutrient source, but don't necessarily control phytoplankton dynamics. Guillaud et al. (2018) show that sediments have a high influence on Chla levels as well (light limitation in high flow periods/winter).

3. L22-24, P2. Consider adding references to support this.

4. L9, P3. "The VB. . . is located under direct influence of these two rivers": This is not really clear from Fig. 1. See general comment 1.

5. L4-5, P4. The link between the first two sentences of this paragraph is not clear.

[Figure]

6. L25, P4 – L6, P5. This paragraph would benefit from more explanations on the DLM method. When you say "look like interpolation", do you mean it is equivalent to interpolation? If yes, which kind of interpolation? Why do you choose to fit second order polynomial functions for the trends, and bimodal trigonometric functions for the seasonality? Is it based on any preliminary analysis of the data? What does "time units" refer to? Is it the frequency at which the trends/seasonal variations are estimated? Why are those plotted with (two different types of) log scales? It makes it more difficult to link the figures with the values provided in text.

7. L14-19, P5. What extra information does the MK test provide? Trend values can already be extracted from the DLM analysis. Is the method applied to the trend/seasonality functions from the DLM analysis, or to the raw data? Are uncertainties accounted for?

8. L19, P7-L7, P8. Results on Chla concentrations and phytoplankton species in the VB are not thoroughly presented here. It seems from the seasonality plot that, in the timeframe of the study, Chla has always peaked in spring and summer, and that since 2006 the summer peak has reached similar concentrations to the spring one. It's also interesting to note that there seems to be a succession of 3 algae blooms: a diatom bloom in spring, a dinoflagellate one in early summer, when DSi is depleted, and another diatom one in late summer.

9. L21-22, P8. Why would trends in discharge in the studied rivers depend on variations in the precipitation in river basins flowing to the North Sea?

10. L30, P8-L4, P9. This paragraph would be more convincing if estimates of the loads from the different sources were provided. Is the Loire "probably" the major nutrient source, or has it been shown that it actually is? How much water/nutrients are retained in the Arzal dam, and how does it influence the loads reaching the VB? Are the discharge and loads from the Vilaine really negligible in summer, even though it flows directly to the Bay, while the Loire river plume has to travel 120km?

11. L9, P9-L20, P9. The phytoplankton succession is not thoroughly discussed here. See Specific comment 8. Even though they are decreasing, spring diatom abundances are still superior to summer ones. It is mentioned that temperature changes can induce shifts in species' succession. Is it the case here? It would also be interesting to discuss the relationship between phytoplankton successions and variations in DSi, for example.

12. L20, P10. Does Table S2 show values for the Bay of Biscay or for the Ouest Loscolo station only?

13. L21-23, P10. Precise that these correlations are at the annual scale. Seasonal variations of DIN and DSi do not seem correlated.

14. L5-16, P11. Please provide some numbers to support your conclusions.

15. L9-12, P12. An opening on eutrophication and its mitigation would fit better, regarding the introduction.

16. Table S1. When different measurement methods were used for a same variable, consider indicating which time period corresponds to which method.

—— Wording ——

- Throughout the text: "Vilaine Bay/VB" -> "the Vilaine Bay/VB"

- L15, P1. "in relation to those in their…" -> "in relation to changes in its"?

- L4, P2. "myriad responses" -> "myriad of responses"

- L15-18, P4. "The removed… general trend observed": Please reformulate.

- L10&12, P5. "position of" -> "timing of"

- L14, P9 & L8, P12. "course" -> "succession"

---

## Author Comment (AC1) · 30 Nov 2018

Dear editor,

We thank the referees for having agreed to evaluate this manuscript. In a previous submission to Estuaries and Coast, referees urged us to draw a short paper focused on the consequences of a decrease in nutrient concentrations along a land-sea continuum resting almost exclusively on P. One of the Estuaries and Coast referees suggested the present title.

Referees' comments on this second submission indicate that our results may be of great interest. The referee #1 even indicates that figure 8 can be "great final figure" and referee #2 summarizes well the core ideas in our paper, writing that our message is clear. However, through their remarks we understand that the manuscript suffers from shortcuts and that we must review three aspects in particular:

- The introduction should not simply start with N versus P abatement strategies, but also include the concept of continuum,
- The long-term series analysis method needs to be better explained,
- The continuum from the Loire and Vilaine Rivers to the Vilaine Bay must be more clearly established, especially the link between the Loire and the Vilaine Bay.

We expect to be given the opportunity to revise the manuscript paying particular attention to the following points. The **introduction** will start with the hypothesis that eutrophication trajectories in the Vilaine Bay in recent decades have been influenced by those in the Loire and Vilaine Rivers. The objective becomes an investigation into long-term evolution in eutrophication parameters in rivers and coastal waters. We aim to establish the link between fresh and marine water trajectories and highlight the impact of N versus P reduction strategies in rivers on coastal water quality.

In the **materials and methods**, we will add elements showing the representativeness of the marine bay monitoring station and the links between the two rivers and the bay. The description of tools used to assess the long-term trends and changes in seasonality will be extended to answer referees' questions about how we used the Mann-Kendall test to support the DLM results.

The **result** section will be rewritten and the figures will be modified so that the reader can access the actual values. We decided to remove the result of diatom/dinoflagellate ratios.

The **discussion** section will be reorganized and extended following the suggestion of referee #1:

1) Eutrophication trajectories at river basin outlet,

2) Eutrophication trajectories in the Vilaine Bay, including a section on the respective contributions of diatoms and dinoflagellates to eutrophication processes in the Vilaine Bay,

3) River to Bay continuum, including a section dedicated to the potential influence of the processes within estuaries and dam on nutrient loads,

4) Implications for nutrient management.

We sincerely thank the editor and look forward to your positive feedback.

On behalf of the co-authors,
Widya Ratmaya

**Referee 1 – C. Minaudo**

**General comments**

**Referee's comments (RC)** - This study by Ratmaya et al., focuses on eutrophication trajectories over three decades in a large bay located in the French Atlantic coastal zone. It tries to link long term trends and seasonal evolutions in the main bay tributaries with the ones observed in the bay itself, based on a trend+seasonalilty time series decomposition algorithm. This study could be of interest for Biogeosciences readers, but suffers from too many issues such as lack of a clear research question, lack of structure within and between sections, and several technical issues that need to be addressed before it can be considered for publication.

The main issues to me are the following:

- Concentration time series in the Loire River (the main tributary) originate from a station located in a river section under estuarine influence but was considered as representative of the freshwater part.

- Methodology is not clear, especially for the seasonal analysis using the DLM approach. Authors need to define clearly the metrics that were used in this work (e.g. little is said on MK slopes p-values although they appear in Tables)

- Nothing is presented on the impact of estuarine zones on DIN and NIP, disabling the credibility of the interpretations made to explain eutrophication trajectory in the coastal zone.

- If the presence of a dam at the outlet of one of the two tributaries is mentioned, nothing is explained on the potential impacts this should have on the nutrient dynamics discharged into the bay Additionally, this manuscript needs language editing. Many sentences need to be either removed or modified for the sake of clarity. I decided to focus on specific comments on Method and Results sections, because I think interpretation in the Discussion section might change once everything has been addressed properly.

**Author's comments (AC)** - We thank the referee for the detailed and constructive comments. All issues raised are listed and carefully answered point by point below. The previous manuscript was carefully reviewed by an English native. However, the sentences that referee pointed out will be reviewed and modified if necessary.

**Specific comments**

1. **RC** - Page 2; Lines 28-29 (2;28-29): this hypothesis has been proven wrong in many studies. I don't think you should present your problematic this way.

   **AC** - The hypothesis tested in the present study deals with coastal waters, based on Schindler et al. (2008) and Schindler (2012), who stated that the reduction of P inputs is enough to mitigate eutrophication in lakes and other freshwater ecosystems. Although authors carried out their experiments on lakes, they wondered whether the P-only reduction paradigm could be applied to coastal waters. Schindler (2012) also stated that he was unable to find long-term, ecosystem-scale evidence that controlling N input, either alone or in addition to P resulted in oligotrophication of estuaries. We believe that our dataset provides the opportunity to demonstrate that, conversely,

without N input reduction in rivers, the coastal waters under their influence are unlikely to recover from eutrophication.

Author's changes in manuscript:

We plan to review the introduction by integrating recent articles by Schindler et al. (2016), Paerl et al. (2016) and others (see letter to editor).

2.  **RC** - Page 3, section 2.2: explain that you extracted the longest records available. The reader doesn't know at this point that multi-decadal data is available.

Author's changes in manuscript:

The information on the availability of dataset will be added to the main text of the manuscript.

3.  **RC** - Page 3, section 2.2: If Montjean is considered as the last freshwater station on the Loire, why would you use concentrations originating from Ste Luce located in the zone influenced by estuarine salinity? This is a choice that could mislead your interpretations. Also, when computing loads, which site served as the reference? That means did you calculate loads at Montjean or Ste Luce and how did you proceed (e.g. catchment areas ratio?)?

**AC** - Sainte-Luce is the last station for water quality monitoring on the Loire, located upstream of the haline intrusion (Guillaud et al., 2008), therefore it is a freshwater station closer to the river mouth than Montjean. The influence of tidal dynamics at Sainte-Luce was avoided by discarding data collected during high tide. In the database of Loire-Brittany River Basin Authority, Sainte-Luce displays a longer dataset (since 1980s) than Montjean (from 1995). Nutrient concentrations measured at Montjean showed parallel long-term evolutions to those observed at Sainte-Luce (Figure R1).

[Figure]

Figure R1. DIN concentrations in the Loire River at Sainte-Luce (blue) and Montjean (red)

Riverine nutrient load calculations were based on nutrient concentrations at Sainte-Luce and river discharge at Montjean, since there is no measurement of discharge at Sainte-Luce. Guillaud et al. (2008) calculated riverine nutrients loads based on the same stations, as a forcing parameter for the ecological ECO-MARS3D model simulating phytoplankton production in the Bay of Biscay (see Huret et al., 2013; Ménesguen et al., 2014; Ménesguen and Dussauze, 2015; Ménesguen et al., 2018).

Riverine nutrient loads were calculated as a function of river discharge and nutrient concentrations, not in relation to the catchment area (see below).

For this section, we decide to keep Saint-Luce as reference station for water quality. Exchanging Sainte-Luce dataset for that of Montjean will not affect the overall results.

Author's changes in manuscript:

"DIP, DIN and Chl a data came from stations located upstream of the haline intrusion: Sainte-Luce-sur-Loire on the Loire and Rieux on the Vilaine, and DSi from Montjean-sur-Loire on the Loire and Férel on the Vilaine (Fig. 1)."

The following lines will be added to the manuscript in order to clarify the choices of stations:

"For Sainte-Luce-sur-Loire, the influence of tidal dynamics was avoided by discarding data collected during high tide."

"In order to calculate riverine nutrient loads, gauging stations located close to the river mouth were selected and specifically those where nutrient concentrations were regularly measured. When information on nutrient concentrations was not available for the same location, data from the nearest station representative of the river outlet was used."

4. **RC** - Page 4, Line 4-5 (4;4-5): you should make sure this assumption on NO3 being >90% TN is correct. For the riverine part, AELB also provides TN concentrations.

    **AC** - The referee seems to confuse dissolved inorganic nitrogen (DIN) with TN. In the present study, we considered dissolved inorganic nutrients (as bioavailable forms of nutrient for phytoplankton). Concerning DIN, nitrate was the most dominant form (>90%) of DIN (see Bouraoui and Grizzetti, 2011; Garnier et al., 2018; Ménesguen et al., 2018). Thus, our sentence was correct. However, we can improve the clarity of the sentence.

    Author's changes in manuscript:

    "DIN was defined as the sum of nitrate, nitrite and ammonium, with nitrate as the major component (>90%)."

5. **RC** - 4;6: this method for load calculations is subject to large errors, especially on DIP. You should use a discharge weighted method, commonly used by our community, and recommended within OSPAR convention.

    **AC** - The method recommended by OSPAR, discharge weighted concentration (DWC) is commonly used when calculating annual loads, as also stated in Dupas et al. (2018) and in RID document (OSPAR Commission, 2017). This method is not relevant when long-term trend study includes seasonal variation. Moreover, this method has disadvantages. One of which is the application of limit of quantification when data is missing or unavailable. This can overestimate load estimation. Therefore, we prefer to retain our method of load calculation.

6. **RC** - 4;24: residuals as white noise is an hypothesis that is not always met by these algorithms. Please, remove "white noise" in this sentence.

    Author's changes in manuscript:

    "white noise" will be removed.

**7.** RC - We need a metric to assess if your algorithm performs well or note, especially when working at the seasonal scale with variables that don't have stable seasonality patterns (e.g. phytoplankton biomass).

AC - In the present study, we use DLM to analyze time series data of water quality parameters linked to eutrophication. The DLM time-series analysis provides figures allowing the visual identification of changes in trends and in seasonality.

The trend plot displayed observed values with colored dots corresponding to the season. The trend was represented by a dark grey line with its 90% confidence interval (shaded area). For period 1997-2013, the longest common record for all variables, a linear trend significance test was performed on trend components from DLM using a modified non-parametric Mann-Kendall (MK) test (Yue and Wang, 2004). When linear trends were significant ($p < 0.05$), their magnitude was calculated using non-parametric Sen's slope estimator (Helsel and Hirsch, 2002).

The DLM seasonality plot displays the seasonal component estimated by the DLM, which indicates the seasonal position of maximum and minimum values, and the amplitude of seasonal cycles. Changes in the timing of annual maximum or minimum values were highlighted in the seasonality plots by colored dots, which change over time. Changes in the seasonal amplitude (increase or decrease of the value for a given season) were assessed using the modified MK test performed on DLM seasonal components from each season. The seasons were defined as: winter (January, February, March), spring (April, May, June), summer (July, August, September), and autumn (October, November, December).

Thus, the significance test of linear changes in DLM trends and seasonality components were provided by the modified MK test associated with Sen's slope estimator.

Table R1 below shows the coefficient of determination for each model parameter, which gives an indication of the goodness of fit of a model. It is estimated by calculating the square of the sample correlation coefficient between the observed outcomes and the observed predictor values. It can also be viewed as the ratio of the explained variance to the total variance. We also add the estimated significance of trends and seasonality based on the squared correlation coefficient between the calculated trend and deseasonalized data and on the squared correlation coefficient between the calculated seasonal component and detrended data respectively (Minaudo et al., 2015).

Table R1. Coefficient of determination, significance of trends and seasonality estimated for the period of 1997-2013

| Parameters | Overall (%) | Trend (%) | Seasonality (%) |
|---|---|---|---|
| Loire discharge | 94.6 | 52.8 | 94.1 |
| Loire DIP | 51.5 | 16.9 | 45.6 |
| Loire DIP Loads | 71.3 | 29.4 | 67.1 |
| Loire DIN | 87.9 | 38.9 | 87.1 |
| Loire DIN Loads | 92.6 | 60.7 | 91.8 |
| Loire DIN/DIP | 48.1 | 18.6 | 41.6 |
| Loire Chl a | 68.8 | 55.6 | 53.3 |
| Vilaine discharge | 96.1 | 65.9 | 95.7 |
| Vilaine DIP | 52.0 | 40.0 | 28.4 |
| Vilaine DIP loads | 71.5 | 48.4 | 60.1 |
| Vilaine DIN | 95.7 | 60.1 | 95.4 |
| Vilaine DIN loads | 95.2 | 62.4 | 94.7 |
| Vilaine Chl a | 59.0 | 44.4 | 43.3 |
| Vilaine DIN/DIP | 66.3 | 42.3 | 55.5 |
| VB DIP | 61.0 | 4.9 | 60.5 |
| VB DIN | 85.7 | 33.4 | 85.0 |
| VB DSi | 64.3 | 9.9 | 62.6 |
| VB DIN/DIP | 83.2 | 23.5 | 82.7 |
| VB DIN/DSi | 78.9 | 18.0 | 78.1 |
| VB DSi/DIP | 28.1 | 16.9 | 15.6 |
| VB Chl a | 58.8 | 26.7 | 51.0 |
| VB Diatoms | 48.5 | 6.9 | 46.7 |
| VB Dinoflagellates | 43.7 | 2.7 | 43.0 |

8.   **RC** - 5;1: why was this log-transformation necessary? It needs justification.

**AC** - The log transformation was necessary because:

"many measurements show a more or less skewed distribution. Skewed distributions are particularly common when mean values are low, variances large, and values cannot be negative, as is the case, for example, with species abundance, lengths of

latent periods of infectious diseases, and distribution of mineral resources in the Earth's crust. Such skewed distributions often closely fit the log-normal distribution (Aitchison and Brown 1957, Crow and Shimizu 1988, Lee 1992, Johnson et al. 1994, Sachs 1997)."

This subject has been deeply discussed in Limpert et al. (2001).

Log-normal distribution induces a variance to mean relationships, that is, as in our case the mean and the variance vary with time, and thus the homoscedastic hypotheses, i.e., specifically for us, the equality of error terms variance through time, may not be fulfilled. This is why "A variance stabilizing log transformation…" is applied in the first place.

In addition, since all of our variables are positive, treating them without log-transformation may lead confidence intervals to include negative values, which consequently leads to inadequate models.

Author's changes in manuscript:

Justification of this log-transformation will be added in the revised manuscript.

9.  **RC** - 5;7: reading log-transformed units is not convenient for the reader. You can log-transform the axis but still present actual values. Why did you log-transform the data in the first place? It makes the trend observation less clear to the reader.

Author's changes in manuscript:

We suggest log-transforming the axis to present the actual values. Please see an example Figure R2.

[Figure]

Figure R2. Trends of Chl a concentrations in the Loire

10. **RC** - The authors decided to use units that are consistent in the manuscript, but not commonly used by researchers on lotic environment. Please, convert all mol/L into mg/L or µg/L

    **AC** - The SI unit of concentration (quantity of substance) is the mole per cubic meter (mol m$^{-3}$), which is commonly used in marine environment research. The use of "mol" is also consistent with the calculation of nutrient molar ratios, which permit us to assess potential nutrient limitation. Therefore, we prefer to keep as it is.

11. **RC** - 5;8-9: the explanation on trends significativity test is not clear nor properly justified. You need a metric for this. Why not use Sen's Slope significativity test?

    **AC** - Please see the detail for trend significance test in point #7.

Author's changes in manuscript:

The use of the modified Mann-Kendall test as a trend significance test will be more detailed in the section 2.4.

12. **RC** - 5;11: the authors should define clearly which metrics were extracted for the time-series analysis and used for further analysis.

   **AC** - Please see point #7 for trend significance test.

13. **RC** - 5;15: have you conducted MK test on de-seasonalized = observations – seasonal component, or on de-seasonalized = trend component? The latter discards residuals from the analysis and this choice should be justified. Also, I think residuals from your DLM algorithm should be plotted along.

   **AC** - The modified Mann-Kendall test was conducted on de-seasonalized = trend component from DLM (please see point #7 for details).

   We provide residual plots for all parameters treated in this study in Appendix 1 for the consideration and verification of the referee. We did not include residual plots in results, as the number of figures was already high, as noted by the referee in point #21. Furthermore, papers including such figures are very uncommon.

14. **RC** - 5;17: it is not clear how you proceeded to identify seasonal trends. Did you use a seasonal MK test? This needs more details since it is the core of your analysis.

   **AC** - Please see point #7 for detail in seasonal trends.

15. **RC** - Besides, how would you justify analyzing loads evolutions and not only concentrations since you show that Q was stable over time? Removing all the load trajectory description would save space for other elements in your paper, and benefit to the clarity of your messages.

   **AC** - River discharge (Q) appeared stable in spite of oscillations. The modified MK test applied to river discharge trend component from DLM showed a significant decrease between 1997 and 2013 ($p<0.05$, Table 1). Therefore, it was necessary to calculate riverine nutrient loads in order to show that these loads displayed similar trends to those of nutrient concentrations.

16. **RC** - 5;21: What is STATGRAPHIC CENTURION and what are the metrics/analysis conducted with this? Please, add a reference for this.

   **AC** - In the manuscript section 2.4, we mentioned the use of STATGRAPHIC CENTURION software for Spearman Correlation analysis.

Author's changes in manuscript:

We suggest a modification if needed as follows:

"Spearman Correlations were computed for annual median values in order to analyze relationships among variables, and tested using STATGRAPHIC CENTURION software (Statgraphics Technologies Inc, Version XVII, Released 2014)."

17. **RC** - 5;26: how significant is this trend in Q data? A large slope in MK tests doesn't mean that it is statistically significant.

   **AC** - Table 1 shows a $p$-value of 0.014 (for significance level of 0.05) for river discharge, indicating a significant trend. The negative slope indicates a decreasing

trend. This decrease in Loire discharge was also observed in previous studies (please see point #15)

18. **RC** - 6;5: this seasonal shift is not observable in Figure 3. Consider adding a Figure to show seasonal variations and evolutions.

**AC** - The seasonal shift in position of annual DIP minimum from summer to spring is clearly visible by the change in color (Fig. 3b). It started with yellow (summer) around 1999 and changed to green (spring) from 2007 to the end of studied period.

Author's changes in manuscript:

We modified sentences concerning the seasonality as follows.

"Changes in position of annual maximum or minimum were highlighted in the seasonality plots by colored dots, which change over time. Changes in the seasonal amplitude (increase or decrease of the value for a given season) were assessed using the modified MK test performed on DLM seasonal components from each season. The seasons were defined as: winter (January, February, March), spring (April, May, June), summer (July, August, September), and autumn (October, November, December)."

The interpretation of the seasonality of Fig. 3b in the result section will be more detailed and used as an example of seasonality change for other figures.

19. **RC** - In the Result section, it is good to refer to Tables and Figures, but the reader also needs actual values included in the text, otherwise he always has to go back and forth from text to Table/Figure.

**AC** - The values were already in the text accompanying the trend interpretation, except for loads

Author's changes in manuscript:

Actual values will be added for those that were missing.

20. **RC** - 6;12: please, be more specific, and always use similar ways of describing the data: first, trends. Second, seasonal variations. It helps increasing the clarity of the manuscript and makes things easier for the reader.

**AC** - We described results as follows: 1) trends accompanied by actual data; 2) seasonality; 3) correlation.

Author's changes in manuscript:

This plan will be more systematically applied in the revised version

21. **RC** - You have too many additional figures. Please, make a selection of the ones that are really useful to support your ideas.

**AC** - Our manuscript present results of two rivers and a coastal ecosystem. We decided to place the results of one of the two rivers in supplementary materials because they must be accessible for the reader.

22. **RC** - 7;19: add a section for this correlation analysis

Author's changes in manuscript:

We will modify the description of results, rather than adding a new section (please see point #20).

23. **RC** - 7;24-29: Do you believe your DLM analysis is suitable for phytoplankton biomass description at the seasonal scale? You need to validate this first, and plots in Figure 7 don't help answer this question if you don't show residuals (you'll see that they don't look like white noise).

**AC** - We presume that these comments deal with phytoplankton abundances, not phytoplankton biomass. We do not understand the second sentence of the referee's comment. It is a time series and for different reasons (e.g.., missing data, irregular sampling frequencies, exceptional abundances), we believe that DLM, which has been used previously in published studies (see Soudant et al., 1997; Scheuerell et al., 2002; Hernández-Fariñas et al., 2014; Hernández-Fariñas et al., 2017) is an appropriate tool for such data. Let us define white noise as values that are mutually uncorrelated with zero mean and have the same Gaussian probability distribution.

The residuals of diatom abundances are available in Appendix 1. The residuals QQ-plot of diatom abundances is presented below.

[Figure]

Figure R3. The QQ-plot of diatoms abundances:

The Kolmogorov-Smirnov $p$-value is equal to 0.2244. The Stoffer-Toloi (i.e., autocorrelation test) p-value is 0.1723. These results suggest that this is actually white noise.

24. **RC** - Insets in Figure 7 are not explained. It has to be.

**AC** - The explanation is already in the legend.

"Figure 7: Long-term trends and seasonality of Chl a (a, b), diatom (c, d), dinoflagellate (e, f) and diatom:dinoflagellate ratios (g, h) in VB. Insets show trends with optimal scale. See Fig. 2 for details"

Author's changes in manuscript:

More information on graphical representation will be added in the method section of the revised version.

25. **RC** - Section 4: this section could be reorganized as follows 1) Nutrients and Chl-a trends at river basin outlets 2) Nutrients and Chl-a trends in the bay 3) River to bay continuum 4) Implications for management

**AC** - The discussion section of the previous manuscript takes into account an unequivocal link between the bay and rivers, as shown in previous studies (e.g.,

Guillaud et al., 2008; Ménesguen and Dussauze, 2015; Ménesguen et al., 2018b). Therefore, the paragraphs described the variables (nutrients and phytoplankton) by grouping results from rivers with those from the Vilaine Bay. We understand that it was too hasty (i.e., continuum).

Author's changes in manuscript:

We propose to reorganize the discussion section as follows: 1) Eutrophication trajectories at river basin outlet 2) Eutrophication trajectories in the Vilaine Bay 3) River to Bay continuum 4) Implication for nutrient management. Please see letter to editor for detail.

26. **RC** - How does the estuarine zone could interfere in your interpretations? Same question with the presence of a dam at the outlet of the Vilaine river? This needs to be addressed, at list by listing the; different processes that occur. Many has been done on the subject.

Author's changes in manuscript:

We will add text explaining the potential influence of processes within estuaries and dam to section 4.3 of discussion (River to Bay continuum) of the revised manuscript (please see letter to editor). However, we conclude that they may not have modified trends in riverine loads.

27. **RC** - 8;21: This can't be said like this. At the outlet of large and intensively managed catchments, nutrients variations are co-controlled by upstream hydrological variations, delivery to stream modalities (point or diffuse sources?), and by instream retention processes through physical and biogeochemical processes.

**AC** - The lines that the referee has pointed out dealt with the variation in nutrient transfer from watershed to coastal waters, not the variations in nutrient concentrations in river waters: "The transfer of nutrients from continents to coastal waters is largely driven by freshwater inputs, the dynamics of which depend largely on precipitation in watersheds". Thus, in our opinion, our sentence was correct.

28. **RC** - 8;22: You should mention the North Atlantic Oscillation to explain the 7 years cycles. See also Dupas et al., 2018 (WRR)

**AC** - The inter-annual variability (i.e., oscillation of 6-7 years) of river discharges are related to precipitation regimes, which are modulated by climate (i.e., NAO for the North Atlantic region). The relationship between flow regimes, precipitation and NAO was explicitly detailed in Radach and Pätsch (2007). We supposed that we did not need to mention this point, which is not essential to support the main subject of the paper. However, for BGS we can include reference to NAO.

Author's changes in manuscript:

The sentences corresponding to this subject will be modified as follows:

"Trends in the Loire and the Vilaine discharges displayed similar oscillations to those of rivers flowing to the North Sea as reported by Radach and Pätsch (2007), suggesting a common hydro-climatic pattern in Western Europe linked to NAO."

29. **RC** - Figure 8 could be a great final figure, but needs to be explained once the processes explaining the different patterns in eutrophication metrics are completely described.

Author's changes in manuscript:

We will explain in more detail the construction of figure 8 in the text and the caption.

**Technical comments**

30. **RC** - Page 1, Line 14 (1;14): remove "(i.e., phytoplankton biomass)", as eutrophication expression is not only phytoplankton excessive biomass.

Author's changes in manuscript:

The sentence will be modified as follows:

"The evolution of eutrophication variables (i.e., nutrients and phytoplankton biomass) during recent decades was examined in the coastal waters of the Vilaine Bay (VB, France) in relation to those in their main external nutrient sources, the Loire and Vilaine Rivers."

31. **RC** - 2;5-7: this has to do with different source types and it should be explained. Environmental measures to tackle P were successful because P largely originated from point sources with limited legacy effects in the streams. For N, diffuse sources dominate and there is large legacy effect.

**AC** - The information suggested by the referee has been extensively explained in the cited articles and we found it unnecessary to add information that was detailed elsewhere. We mentioned the question of different source types and legacy effects in the discussion section 4.3.

32. **RC** - 2;7-11: you should also mention freshwater ponds and lakes were eutrophication is still severe despite large P reductions.

**AC** - We can mention it, but this is not the main subject of the study. We would like to focus on eutrophication in coastal ecosystem, by highlighting the needs to reduce both P and N loads to mitigate eutrophication along the land-sea continuum.

33. **RC** - 2;16-17: I'd remove the codes for what you called "water masses". Do the authors mean "water body"?

Author's changes in manuscript:

The codes will be removed and the sentences will be modified (see letter to editor)

34. **RC** - 2;18: an actual scientific reference would be better.

Author's changes in manuscript:

The sentences will be modified and scientific references will be added in the revised version.

35. **RC** - 3;3: "widest" is not correct. You may refer to "largest river basin".

**AC** - The use of the word "widest" refers to the river Loire and not to the river basin. This usage has been confirmed by our English native.

36. **RC** - 3;4: sentence is not clear, please, rephrase it.

Author's changes in manuscript:

The sentence will be rephrased as follows:

"Their catchment areas are dominated by agricultural activity, together sustaining two-thirds of the national livestock and half the cereal production."

37. **RC** - 3;22: sentence is not clear, please, rephrase it.

Author's changes in manuscript:

The sentence will be rephrased in the next version as follows.

"Nutrient and Chl a concentrations, plus phytoplankton counts in VB (Ouest Loscolo, Fig. 1) were provided by the French National Observation Network for Phytoplankton and Hydrology in coastal waters (REPHY, 2017)."

38. **RC** - 5;1: the use of ":" separates the sentence in a way that makes it hard to understand. Please, modify.

   **AC** - Corrected

39. **RC** - 5;8-9: check for use of different tenses throughout the manuscript.

Author's changes in manuscript:

The sentence will be removed. The method section 2.4 will be detailed

40. **RC** - 5;23: Change this section title to "Discharge and nutrients long term trends in freshwater basin outlets"

Author's changes in manuscript:

The title suggested does not take into account the Chl *a*. We propose a modification to the section title as follows:

"Long term trends in eutrophication parameters in river basin outlets"

**Referee 2 – Anonymous referee**

**Referee's comments (RC)** - This paper studies the long term trends in nutrient and phytoplankton dynamics in the Loire and Vilaine rivers, and in the Vilaine Bay (VB). The authors discuss changes in eutrophication of these systems, and relate changes in the VB to those in nutrient inputs from the two rivers. They show that, even though phytoplankton blooms decreased in the riverine systems following reduction in dissolved inorganic P, phytoplankton biomass in the VB has continued to increase. This could be fueled by nitrogen delivery from the rivers (slightly increasing trend for the Loire), together with phosphorus and silica recycling from bottom sediments in the coastal area. This is an interesting discussion point, that totally fits Biogeosciences' scope. This is however only superficially discussed, and the layout of the paper makes it difficult to identify the main conclusions. I also noted important gaps in the methods' description.

The presentation of the river trajectories is extensive, but was already thoroughly discussed in a previous study (Minaudo et al., 2015). Very complete time series of Chla concentrations and abundances of different phytoplankton species in the VB are presented, and could be extremely valuable to examine changes in community structure. However, these are not discussed in depth. Moreover, more elements should be provided to the reader to justify that the data presented here is enough to support the conclusions of the study. In fact, interpretations of the dynamics in the VB are derived from observations at a single point, at which the influence of the Loire river is not obvious and not discussed.

I believe these major shortcomings should be addressed before this work can be published.

**General comments**

1.  More information on the influence of the Loire river on the VB dynamics is needed. In fact, nutrients need to travel more than 120km from the Loire river monitoring station (Saint Luce sur Loire) to the Bay, through the Loire estuary and along the coast. Do coastal currents carry most of the Loire river's exports to the VB? How can processing in the estuary and along the coast impact loads reaching the VB?

2.  Methods on the Dynamic Linear Models (DLM) and Mann-Kendall (MK) test analysis are not detailed enough. I am also not convinced that the MK test provides any more information than the DLM analysis. To my understanding, numerical estimates on trends and seasonal variations can also be extracted from the latter. Using these two methods to come up with the same interpretations waters down important messages in the results and discussion sections.

3.  Authors refer several times throughout the manuscript to "management scenarios focused solely on P reduction" or on "P alone". However, this is not totally accurate for the study area, and should be moderated. Even though ecosystems responded quicker to P reduction strategies (e.g. for point sources) than to policies on agricultural fertilization, those already exist (e.g. EU Nitrates Directive).

4.  In general, statements are sometimes vague or not totally accurate. The structure of the results and discussion sections makes it difficult for the reader to identify the main conclusions of the study.

These points, together with more minor concerns, are more detailed hereafter, in the specific comments.

**Author's comments (AC)** - We thank the referee for the detailed analysis and constructive comments. For this referee, the link between the Loire River and the VB was not well enough established in spite of numerous works cited in the previous version. We will add the information on the continuum to the revised version. In the case of the Loire/Vilaine Rivers – the VB continuum, we recognize that there was no formal decision to reduce P-only. However, the small decrease of DIN concentrations in the Vilaine and their increase in the Loire especially in summer during recent decades provide a scenario that allows testing the P-only paradigm.

We also agree to add more detail on the method used and to restructure and extend the discussion.

Concerning the influence of estuarine processes, also mentioned by referee #1, we will add texts explaining the biogeochemical processes within estuaries and dam (please see letter to editor).

All issues raised by the referee are carefully answered point by point below.

**Specific comments/scientific questions**

1.  **RC** - L7-8, P2. "This result is consistent with the idea that reducing P alone, and not N, can mitigate eutrophication of freshwater systems (Schindler et al., 2008)": This paper from Schindler et al. does not show this; they study the effect of reducing N only. Moreover, this is not a scientific consensus (e.g. Pearl et al., 2016, Environ. Sci. Technol. 50, pp 10805–10813). This sentence should be moderated.

    **AC** - It is true that these authors also studied the reduction of N in their lake, but it was to support the hypothesis that the reduction of P can be enough for lake restoration. At the end of their summary, the authors also stated that to reduce eutrophication, the focus of management must be on decreasing inputs of P. In a more recent article, Schindler et al. (2016) clearly argued for a reduction of P alone to control eutrophication in lakes and other freshwater ecosystems, even though they recognize that anthropogenic nitrogen emissions can also affect human health and ecosystems (i.e., Box 2).

    Author's changes in manuscript:

    We plan to review the introduction by integrating recent articles by Schindler et al. (2016), Paerl et al. (2016) and others (see letter to editor).

2.  **RC** - L14-15, P2. "Nutrient inputs ...control phytoplankton production in coastal waters of the northern Bay of Biscay": Riverine inputs constitute the major nutrient source, but don't necessarily control phytoplankton dynamics. Guillaud et al. (2018) show that sediments have a high influence on Chla levels as well (light limitation in high flow periods/winter).

    **AC** - It is true that there are also environmental conditions allowing nutrients to be consumed by primary producers, such as water residence time, light availability, etc. The primary production in coastal waters off the Loire and Vilaine River is limited by light availability due to insufficient irradiance during winter and suspended sediment flux from rivers and resuspension during the period of high hydrodynamic activity (Guillaud et al., 2008). However, these authors also showed that, except during periods of light limitation (November - February), phytoplankton blooms in this area respond to the variation in river discharge.

Author's changes in manuscript:

A section explaining the link between the Loire and Vilaine Rivers and the VB coastal waters will be added to the method section (please see below).

3. **RC** - L22-24, P2. Consider adding references to support this.

Author's changes in manuscript:

The following reference will be added (Lunven et al., 2005; Loyer et al., 2006).

4. **RC** - L9, P3. "The VB...is located under direct influence of these two rivers": This is not really clear from Fig. 1. See general comment 1.

**AC** - The link between the Loire inputs and dynamics of the coastal waters of the Northern Bay of Biscay, including the Vilaine Bay, has been established using ecological model ECO-MARS3D (see Ménesguen et al., 2018b). These authors showed in their figure 6 the influence area of several large French Atlantic river plumes during three different flow regimes. It also showed that the VB coastal waters are always affected by the Loire river plume whatever the regime scenario. This can justify the link between the Loire River and the VB (i.e., continuum).

Author's changes in manuscript:

The following reference will be added to support the sentence mentioned by referee (Guillaud et al., 2008; Ménesguen and Dussauze, 2015; Ménesguen et al., 2018b).

The section 2.1 will be completed with the following text explaining the contribution of the Loire and Vilaine Rivers to the VB fertilization.

"The Loire river plume tends to spread north-westward with a dilution of 20 to 100-fold by the time it reaches the VB (Ménesguen and Dussauze, 2015; Ménesguen et al., 2018b). The Vilaine river plume tends to spread throughout the bay and then move westward (Chapelle et al., 1994). The ECO-MARS3D model estimates that the Loire constitutes >60% of VB DIN concentrations during flood regimes and from 20 to 40% during low discharge periods (Gohin, 2012; M. Plus, comm. pers.). Exceptional floods from the Loire and Vilaine can lead to high surface water turbidity in the VB (Guillaud et al., 2008)."

5. **RC** - L4-5, P4. The link between the first two sentences of this paragraph is not clear.

Author's changes in manuscript:

These sentences will be modified as follows.

"DIN was defined as the sum of nitrate, nitrite and ammonium, with nitrate as major component (>90%)."

6. **RC** - L25, P4 – L6, P5. This paragraph would benefit from more explanations on the DLM method. When you say "look like interpolation", do you mean it is equivalent to interpolation? If yes, which kind of interpolation?

**AC** - The sequential DLM approach is provided by the Kalman filter, by identifying the missing values and replacing them with normal random variables. This approach may be viewed as one that uses a prior for the parameter which replaces the missing values. This is another way to say "absence of data leads to no change in distributions for model parameters".

Author's changes in manuscript:

This method section 2.4 will be reorganized and the sentence corresponding to this subject will be replaced with the following:

The DLM approach is particularly suitable for environmental data series characterized by outliers, irregular sampling frequency and missing data. The latter are taken into account by the Kalman filter (Kalman, 1960), a component of the DLM, i.e., no information leads to no changes in distributions for model parameters (West and Harrison, 1997). This approach may be viewed as one which uses a prior for the parameter which replaces the missing value.

**RC** - Why do you choose to fit second order polynomial functions for the trends, and bimodal trigonometric functions for the seasonality? Is it based on any preliminary analysis of the data?

**AC** - We choose a second order polynomial model because looking at the log-transformed time series it appeared to us that a first order (i.e., adapting trend up to linear) was too restrictive and a third order (i.e., adapting trend up to cubic) was not necessary, leading to an over fitted model.

In our geographical area, the annual patterns of phytoplankton variability have a six months periodicity as described by Winder and Cloern (2010). This bimodal pattern is characterized by two peaks per year, such as spring and autumn or summer and winter blooms. In order to allow our model to adapt to such periodicity we have to include a two harmonics seasonal component. Thus, yes it is based on preliminary analysis of the data.

Author's changes in manuscript:

A justification of this subject will be added in the section 2.4 of revised manuscript.

**RC** - What does "time units" refer to? Is it the frequency at which the trends/seasonal variations are estimated?

**AC** - Time unit is the smallest time interval between sampling dates within a period of analysis. In our case, the period is one year. Time units are weekly, fortnightly or monthly depending on the data. Seasonal variations are estimated for each time unit.

**RC** - Why are those plotted with (two different types of) log scales? It makes it more difficult to link the figures with the values provided in text.

**AC** - Y-axis of all graphics has been modified to show original units using a log-scaled y-axis.

7. **RC** - L14-19, P5. What extra information does the MK test provide? Trend values can already be extracted from the DLM analysis. Is the method applied to the trend/seasonality functions from the DLM analysis, or to the raw data? Are uncertainties accounted for?

**AC** - The modified MK test was used as a formal trend significance test. The test was applied respectively to trend and seasonality components from DLM, not to the raw data (please see comment from referee #1point #7 for further detail).

Yes, uncertainties are taken into account by the DLM.

Author's changes in manuscript:

The information on the modified Mann-Kendall test will be more detailed and clearly explained in the revised version.

8. **RC** - L19, P7-L7, P8. Results on Chla concentrations and phytoplankton species in the VB are not thoroughly presented here. It seems from the seasonality plot that, in the timeframe of the study, Chla has always peaked in spring and summer, and that since 2006 the summer peak has reached similar concentrations to the spring one. It's also interesting to note that there seems to be a succession of 3 algae blooms: a diatom bloom in spring, a dinoflagellate one in early summer, when DSi is depleted, and another diatom one in late summer.

**AC** - In this study, we used only the total counts of diatoms and dinoflagellates, to account for the role DSi. These two groups represent >85% of total micro-phytoplankton counts and thus the biomass (section 2.3, line 9-12, page 4).

This change in Chl *a* seasonality was mainly due to the increase in summer diatom abundances and the decrease in spring ones, as suggested by their seasonality.

The seasonal pattern of phytoplankton blooms was characterized by a diatom bloom in spring corresponding to high river flows and another one in late summer. Dinoflagellates tend to increase in summer but their abundances remain lower than those of diatoms, except during discolored water events (Souchu et al., 2013; Sourisseau et al., 2016). The collapse of spring diatom bloom is due more to DIP depletion than DSi (please see point #11).

9. **RC** - L21-22. P8. Why would trends in discharge in the studied rivers depend on variations in the precipitation in river basins flowing to the North Sea?

Author's changes in manuscript:

It will be rephrased as follows:

"Trends in the Loire and the Vilaine discharges displayed similar oscillations to those of rivers flowing to the North Sea as reported by Radach and Pätsch (2007), suggesting a common hydro-climatic pattern in Western Europe linked to NAO."

10. **RC** - L30. P8-L4. P9. This paragraph would be more convincing if estimates of the loads from the different sources were provided. Is the Loire "probably" the major nutrient source, or has it been shown that it actually is? How much water/nutrients are retained in the Arzal dam, and how does it influence the loads reaching the VB? Are the discharge and loads from the Vilaine really negligible in summer, even though it flows directly to the Bay, while the Loire river plume has to travel 120km?

**AC** - According to the modelling study, the Loire is actually the major nutrient source (please see point #4).

Concerning the Vilaine, during the period of low water discharge ($10 - 100$ m$^3$ s$^{-1}$), the dam is closed at high tide. The small releases due to the lock functions of the dam (shipping, fish-way and salt-water pump) represent half of the "natural" discharge (Traini et al., 2015). The dam is closed below 10 m$^3$ s$^{-1}$. The summer Vilaine discharge measured at Rieux displayed strong variations, ranging between <1 to 100 m$^3$ s$^{-1}$, with >95% of values below 60 m$^3$ s$^{-1}$. Therefore, we consider that half of water discharge during the low water period was retained by the dam.

Unfortunately, we do not have any measurement of nutrient concentrations at the dam outlet nor inside the dam, which could be used to estimate the nutrient retention. The presence of a dam at the river outlet may increase water residence time, thus favor nutrient uptake by phytoplankton and loss of N via denitrification (Howarth et al., 1996; Seitzinger et al., 2006). The presence of Arzal dam may thus attenuate nutrient transport from the Vilaine River to the VB. The use of loads calculated from Rieux station has likely overestimated nutrient loads from the Vilaine River. However, taking into account dam retention for the Vilaine, our estimate of nutrient loads from the Loire remains higher than those from the Vilaine (see Table 4).

Author's changes in manuscript:

The estimates of loads from the Loire and Vilaine Rivers will be added to the text.

Concerning the influence of estuarine processes, we provided text that will be added to the revised manuscript (please see letter to editor).

11. **RC** - L9. P9-L20. P9. The phytoplankton succession is not thoroughly discussed here. See Specific comment 8. Even though they are decreasing, spring diatom abundances are still superior to summer ones. It is mentioned that temperature changes can induce shifts in species' succession. Is it the case here? It would also be interesting to discuss the relationship between phytoplankton successions and variations in DSi, for example.

**AC** - Concerning phytoplankton succession, please see point #8. This is true that the spring diatom abundances remain higher than summer ones. The point that we would like to highlight here is the increase in summer diatom abundances, accompanying the increase in summer Chl a as the indicator of the VB degradation. The increase in sea surface temperature (SST) has been reported in the Bay of Biscay by Huret et al. (2013). Désaunay et al. (2007) have reported an increase in winter temperatures on the continental shelf. Thus, the change in the timing of annual maxima observed for phytoplankton biomass could not be attributed to the increase in temperature and was better explained by changes in nutrient loads. Moreover, changes in phytoplankton community structure at species level, in relation to changes in environmental parameters (e.g., temperature), will be examined in other study.

In the coastal waters off the Loire and Vilaine Rivers, including the VB, the phytoplankton bloom is generally limited by DIP in spring and by DIN in summer (Loyer et al., 2006; Guillaud et al., 2008). This pattern has been verified by bioassay (M. Retho, Ifremer 2015, unpublished data). On the basis of DIP and DSi concentrations, and DSi:DIP ratios, the diatoms are rarely limited by DSi. The decrease in DIP riverine loads has increased the DSi:DIP ratios in the VB during the past decades (Fig. S7) and reinforced therefore the DIP limitation in the VB, as suggested by Billen et al. (2007) for the Seine River – the Seine Bay continuum.

Author's changes in manuscript:

The discussion section will be reorganized (please see letter to editor). We will discuss in more detail the relationship between diatom/dinoflagellate dynamics and variations in DSi concentrations.

12. **RC** - L20. P10. Does Table S2 show values for the Bay of Biscay or for the Ouest Loscolo station only?

**AC** - No, it shows global annual median values for the Bay of Biscay, as detailed in the table legend.

13. **RC** - L21-23. P10. Precise that these correlations are at the annual scale. Seasonal variations of DIN and DSi do not seem correlated.

    **AC** - The precision of correlation analysis has been given in the end of the section 2.4. We explained that we used annual median values to compute Spearman's Rank correlation analysis.

    As mentioned in Table 3, annual medians of DIN and DSi in the VB were correlated.

    Author's changes in manuscript:

    The sentence will be modified as follows.

    "Significant negative correlations between annual medians of Chl a in the VB and Chl a and DIP in rivers, as well as significant positive correlations between annual medians of DIN and DSi in the VB with those of river discharge suggest that changes in eutrophication parameters in the VB (i.e., phytoplankton biomass) were directly related to changes in rivers."

14. **RC** - L5-16. P11. Please provide some numbers to support your conclusions.

    Author's changes in manuscript:

    Numbers will be added.

15. **RC** - L9-12. P12. An opening on eutrophication and its mitigation would fit better. regarding the introduction.

    **AC** - The conclusion section will be modified following modification made in discussion section.

    Author's changes in manuscript:

    The text below will be added concerning perspective on eutrophication mitigation

    The internal loads of nutrients from sediments may counteract the reduction of external nutrient loads and may delay the restauration progress. Taking into account these internal processes in modelling studies (i.e., ECO-MARS3D model, Ménesguen et al., 2018a, b; Menesguen and Lacroix, 2018) will better simulate nutrient load reduction scenarios.

16. Table S1. When different measurement methods were used for a same variable. consider indicating which time period corresponds to which method.

    Author's changes in manuscript:

    This information will be added to the revised manuscript, particularly for dataset in the VB.

**Wording**

**RC** - Throughout the text: "Vilaine Bay/VB" -> "the Vilaine Bay/VB"

**AC** - Corrected

**RC** - L15. P1. "in relation to those in their..." -> "in relation to changes in its"?

**AC** - Corrected

**RC** - L4. P2. "myriad responses" -> "myriad of responses"

**AC** - The use of "myriad responses" was based on Cloern (2001) and it has been validated by our English native.

**RC** - L15-18. P4. "The removed...general trend observed": Please reformulate.

**AC** - Corrected

**RC** - L10&12. P5. "position of" -> "timing of"

**AC** - Corrected

**RC** - L14. P9 & L8. P12. "course" -> "succession

**AC** - Corrected

**Appendix 1.** Residual plots for all parameters used in the present study

[Figure]

| Figures | Parameters | Residual plots |
|---------|-----------|----------------|
| **2** | Loire discharge | LL=, 210.54, AIC=, -413.08, Séquence(s)=, 7-1 |
| **3** | Loire DIP | LL=, -102.24, AIC=, 212.48, Séquence(s)=, 9-1, 7-3 |
| **3** | Loire DIP Loads | LL=, -196.59, AIC=, 401.18, Séquence(s)=, 7-1 |
| **4** | Loire DIN | LL=, 166.09, AIC=, -294.18, Séquence(s)=, 8-1, 7-2 |
| **4** | Loire DIN Loads | LL=, -138.95, AIC=, 285.9, Séquence(s)=, 7-2 |

| Figures | Parameters | Residual plots |
|---|---|---|
| 5 | Loire Chl *a* |
[Figure]
 |
| 6 | Vilaine Bay DIP | |
| 6 | Vilaine Bay DIN | |
| 6 | Vilaine Bay DSi | |
| 7 | Vilaine Bay Chl *a* | |

| Figures | Parameters | Residual plots |
|---|---|---|
| **7** | Vilaine Bay Diatom | |
| **7** | Vilaine Bay Dinoflagellate | |
| **Supp 2** | Vilaine discharge | |
| **Supp 3** | Vilaine DIP | |
| **Supp 3** | Vilaine DIP loads | |

[Figure]

| Figures | Parameters | Residual plots |
|---|---|---|
| **Supp 4** | Vilaine DIN | |
| **Supp 4** | Vilaine DIN loads | |
| **Supp 5** | Loire DIN/DIP | |
| **Supp 5** | Vilaine DIN/DIP | |
| **Supp 6** | Vilaine Chl a | |

[Figure]

| Figures | Parameters | Residual plots |
|---------|------------|----------------|
| **Supp 7** | Vilaine Bay DIN/DIP | |
| **Supp 7** | Vilaine Bay DIN/DSi | |
| **Supp 7** | Vilaine Bay DSi/DIP | |

---

## Author Response (AR2)

Dear editor,

We thank you again for giving us the opportunity to revise our manuscript and are grateful to reviewers for their comments and suggestions on the earlier version. Following their suggestions, we have revised the manuscript and made substantial modifications.

In the **introduction**, the objectives became an investigation into long-term evolution in eutrophication parameters in the Loire and Vilaine Rivers and the Vilaine Bay (VB), with a working hypothesis that eutrophication trajectories in coastal waters in recent decades have been influenced by those in rivers. The study also aimed to establish the link between fresh and marine water trajectories and highlight the impact of N versus P reduction strategies in rivers on coastal water quality.

In the **materials and methods**, we provided elements showing the representativeness of the VB monitoring station (Ouest Loscolo). We added the contribution of Loire DIN inputs to VB DIN concentrations estimated by ECO-MARS3D model (Ménesguen et al., 2019), to establish better the links between the Loire and the VB. We also improved the description of the monotonic Mann-Kendall test as a tool to assess the long-term trends and changes in seasonality of DLM.

The **results** have been rewritten and reorganized and the DLM figures modified by transforming the *y-axis* into logarithmic scale so that the reader can access the actual values. We decided to remove some results: diatom/dinoflagellate ratios (Fig. 7g, h).

The **discussion** has been reorganized and extended:
1) Eutrophication trajectories at river basin outlet,
2) Eutrophication trajectories in the VB, including a section on the respective contributions of diatoms and dinoflagellates to eutrophication processes in the VB,
3) River – VB continuum, adding a section dedicated to the potential influence of the processes on nutrient loads within estuaries and dam,
4) Implications for nutrient management.

The comparison between benthic flux measurements in the VB and calculated riverine nutrient loads has been removed (Table 4). The results are accessible in Ratmaya (2018).

Please find enclosed:

1. Responses to referees' comments (in **blue**). Added changes in the manuscript are shown in **red**, in which page and line numbers refer to the revised manuscript: **pages 1-28**

2. Additional changes in the revised manuscript: **pages 29-33**

3. Revised manuscript with MS Word track changes functionality: separate page numbering, **pages 1-47**

We sincerely thank you and look forward to your feedback.

On behalf of the co-authors,

Widya Ratmaya

**Referee 1 – C. Minaudo**

**General comments**

**Referee's comments (RC)** - This study by Ratmaya et al., focuses on eutrophication trajectories over three decades in a large bay located in the French Atlantic coastal zone. It tries to link long term trends and seasonal evolutions in the main bay tributaries with the ones observed in the bay itself, based on a trend+seasonalilty time series decomposition algorithm. This study could be of interest for Biogeosciences readers, but suffers from too many issues such as lack of a clear research question, lack of structure within and between sections, and several technical issues that need to be addressed before it can be considered for publication.

The main issues to me are the following:

- Concentration time series in the Loire River (the main tributary) originate from a station located in a river section under estuarine influence but was considered as representative of the freshwater part.

- Methodology is not clear, especially for the seasonal analysis using the DLM approach. Authors need to define clearly the metrics that were used in this work (e.g. little is said on MK slopes p-values although they appear in Tables)

- Nothing is presented on the impact of estuarine zones on DIN and NIP, disabling the credibility of the interpretations made to explain eutrophication trajectory in the coastal zone.

- If the presence of a dam at the outlet of one of the two tributaries is mentioned, nothing is explained on the potential impacts this should have on the nutrient dynamics discharged into the bay Additionally, this manuscript needs language editing. Many sentences need to be either removed or modified for the sake of clarity. I decided to focus on specific comments on Method and Results sections, because I think interpretation in the Discussion section might change once everything has been addressed properly.

**Author's comments (AC)** - We thank the referee for the detailed and constructive comments. All issues raised are listed and carefully answered point by point below. The previous manuscript was carefully reviewed by an English native. However, the sentences that referee pointed out will be reviewed and modified if necessary.

**Specific comments**

**1.** **RC** - Page 2; Lines 28-29 (2;28-29): this hypothesis has been proven wrong in many studies. I don't think you should present your problematic this way.

**AC** - The hypothesis tested in the present study deals with coastal waters, based on Schindler et al. (2008) and Schindler (2012), who stated that the reduction of P inputs is enough to mitigate eutrophication in lakes and other freshwater ecosystems. Although authors carried out their experiments on lakes, they wondered whether the P-only reduction paradigm could be applied to coastal waters. Schindler (2012) also stated that he was unable to find long-term, ecosystem-scale evidence that controlling N input, either alone or in addition to P resulted in oligotrophication of estuaries. We believe that our dataset provides the opportunity to demonstrate that, conversely, without N input reduction in rivers, some coastal waters under their influence are unlikely to recover from eutrophication.

We have modified the hypothesis as follows (**p.4, l.11-14**):

This long-term ecosystem-scale analysis provided an opportunity to test the hypothesis that eutrophication trajectories in the downstream VB coastal waters during recent decades have been influenced by those in the Loire and Vilaine Rivers. We aim to establish the link between fresh and marine water trajectories and highlight the impact of nutrient reduction strategies in rivers on coastal water quality.

2.    **RC** - Page 3, section 2.2: explain that you extracted the longest records available. The reader doesn't know at this point that multi-decadal data is available.

We have reorganized the **section 2.2** (**p.5-7**) and added the information on the availability of dataset as follows:

Water quality data in rivers (**p.6, l.3-4**): Sainte-Luce-sur-Loire on the Loire and Rieux on the Vilaine provided DIP, DIN and Chl *a*, measured monthly since the 1980s.

River discharge (**p.6, l.8-10**): For the Loire, river discharge measurements at Montjean-sur-Loire were used due to the absence of data at Sainte-Luce-sur-Loire. For the Vilaine, daily discharge data were available at Rieux from the 1980s.

VB dataset (**p.6, l.14-15**): This station is representative of the VB coastal waters (Bizzozero et al., 2018; Ménesguen et al., 2019) and displayed the longest dataset (from 1983 for phytoplankton counts and 1997 for nutrient and Chl *a* concentrations).

The information of acquisition periods, sampling frequencies and methods of analysis is also given in Table S1.

3.    **RC** - Page 3, section 2.2: If Montjean is considered as the last freshwater station on the Loire, why would you use concentrations originating from Ste Luce located in the zone influenced by estuarine salinity? This is a choice that could mislead your interpretations. Also, when computing loads, which site served as the reference? That means did you calculate loads at Montjean or Ste Luce and how did you proceed (e.g. catchment areas ratio?)?

**AC** - Sainte-Luce is the last station for water quality monitoring on the Loire, located upstream of the haline intrusion (Guillaud et al., 2008), therefore it is a freshwater station closer to the river mouth than Montjean. The influence of tidal dynamics at Sainte-Luce was avoided by discarding data collected during high tide. In the database of Loire-Brittany River Basin Authority, Sainte-Luce displays a longer dataset (since 1980s) than Montjean (from 1995). Nutrient concentrations measured at Montjean showed parallel long-term evolutions to those observed at Sainte-Luce (Figure R1).

[Figure]

Figure R1. DIN concentrations in the Loire River at Sainte-Luce (blue) and Montjean (red)

We have added the following information to justify the choice of studied stations:

**p.5, l.18-1** & **p.6, l.1-2**: The Loire-Brittany River Basin Authority (http://osur.eau-loire-bretagne.fr/exportosur/Accueil) furnished dissolved inorganic nutrients and phytoplankton biomass data (dissolved inorganic phosphorus concentrations, DIP; dissolved inorganic nitrogen concentrations, DIN, dissolved silicate concentrations, DSi and chlorophyll a concentrations, Chl *a*) in rivers, at pre-estuarine stations located closest to the river mouth upstream of the haline intrusion (Fig. 1).

**p.6, l.4-5**: For Sainte-Luce-sur-Loire, the influence of tidal dynamics was avoided by discarding data collected during high tide.

Riverine nutrient load calculations were based on nutrient concentrations at Sainte-Luce and river discharge at Montjean, since there is no measurement of discharge at Sainte-Luce. Guillaud et al. (2008) calculated riverine nutrients loads based on the same stations, as a forcing parameter for the ecological ECO-MARS3D model simulating phytoplankton production in the Bay of Biscay (see Ménesguen et al., 2014; Ménesguen and Dussauze, 2015; Ménesguen et al., 2018a, b, 2019). Riverine nutrient loads were calculated as a function of river discharge and nutrient concentrations, not in relation to the catchment area (see below).

For this section, we decide to keep Saint-Luce as reference station for water quality. Exchanging Sainte-Luce dataset for that of Montjean will not affect the overall results.

The following text concerning the riverine nutrient load calculation has been added to the revised manuscript (**p.6, l.12-16**):

In order to calculate riverine nutrient loads, gauging stations located close to the river mouth were selected. River discharge data were extracted from the French hydrologic "Banque Hydro" database (http://www.hydro.eaufrance.fr/). For the Loire, river discharge measurements at Montjean-sur-Loire were used due to the absence of data at Sainte-Luce-sur-Loire. For the Vilaine, daily discharge data were available at Rieux from the 1980s. DIN and DIP loads from rivers were calculated using averaged monthly discharge and individual monthly nutrient concentrations (Romero et al., 2013).

4. **RC** - Page 4, Line 4-5 (4;4-5): you should make sure this assumption on NO3 being >90% TN is correct. For the riverine part, AELB also provides TN concentrations.

**AC** - The referee seems to confuse dissolved inorganic nitrogen (DIN) with TN. In the present study, we considered dissolved inorganic nutrients (as bioavailable forms of nutrient for phytoplankton). Concerning DIN, nitrate was the most dominant form (>90%) of DIN (see Bouraoui and Grizzetti, 2011; Garnier et al., 2018; Ménesguen et al., 2018a, b). Thus, our sentence was correct. However, we can improve the clarity of the sentence.

This information is now found in the **section 2.2**, **p.6, l.3-4** of the revised manuscript.

DIN was defined as the sum of nitrate, nitrite and ammonium, with nitrate as the major component (>90%).

5. **RC** - 4;6: this method for load calculations is subject to large errors, especially on DIP. You should use a discharge weighted method, commonly used by our community, and recommended within OSPAR convention.

   **AC** - The method recommended by OSPAR, discharge weighted concentration (DWC) is commonly used when calculating annual loads, as also stated in Dupas et al. (2018) and in RID document (OSPAR Commission, 2017). This method is not relevant when long-term trend study includes seasonal variation. Moreover, this method has disadvantages. One of which is the application of limit of quantification when data is missing or unavailable. This can overestimate load estimation. Therefore, we prefer to retain our method of load calculation.

6. **RC** - 4;24: residuals as white noise is an hypothesis that is not always met by these algorithms. Please, remove "white noise" in this sentence.

   "white noise" has been removed and the sentence has been modified as follows (**p.8, l.8-9**):

   The model decomposes an observed time-series into component parts, typically trend, seasonal component (i.e., seasonality) and residual.

   The section concerning time series analysis has been substantially modified and now found in the **section 2.3, p.7-10**, in the revised manuscript.

   This section contains four subsections:

   2.3.1 Data pre-processing
   It contains dataset validation and log-transformation prior to time-series analyses.
   2.3.2 Time-series decomposition
   It covers method to analyze time-series data (DLM), approach to take into account missing values, model used for trends and seasonality, assessment of residuals and outliers.
   2.3.3 Trend
   It explains DLM trend plots and the significance test of linear changes in DLM trend components (modified Mann-Kendall test).
   2.3.4 Seasonality
   It explains DLM seasonality plots the significance test of changes in the seasonality (monotonic linear increase or decrease in the value for a given season).

7. RC - We need a metric to assess if your algorithm performs well or note, especially when working at the seasonal scale with variables that don't have stable seasonality patterns (e.g. phytoplankton biomass).

**AC** - In the present study, we use DLM to analyze time series data of water quality parameters linked to eutrophication. The DLM time-series analysis provides figures allowing the visual identification of changes in trends and in seasonality.

We added to the revised manuscript the following subsections to explain DLM trend and seasonality significance assessment respectively:

**2.3.3 Trend (**p.9, l.7-13**)**

The DLM trend plot displayed observed values with a shade of color for each time unit segments: weekly, fortnightly or monthly. The trend was represented by a dark grey line with the shaded area indicating the 90% confidence interval. For the longest common record of all variables, 1997-2013 called the "common period", a monotonic linear trend significance test was performed on DLM trend components using a modified non-parametric Mann-Kendall (MK) test (Yue and Wang, 2004). When monotonic linear trends were significant ($p<0.05$), changes were calculated from differences between the beginning and the end of the common period of the Sen's robust line (Helsel and Hirsch, 2002).

**2.3.4 Seasonality (**p.9, l.14-19** & **p.10, l.1-5**)**

The seasonality plot displayed the DLM seasonal component values. The figure gave a visual access to the inter-annual evolution of the amplitude, corresponding to the difference between the minimum and maximum values of each year. As dependent variables have been log-transformed, the model was multiplicative. Therefore, when seasonal component values equaled to 1 (i.e., horizontal line), fitted values equaled to the trend. The seasonality plot also allowed a visualization of how the values have evolved over the years according to their seasonal position. The significance of changes in the seasonality (monotonic linear increase or decrease in the value for a given season) was assessed for the common period using the modified MK test performed on DLM seasonal components for each season. The seasons were defined as winter (January, February, March), spring (April, May, June), summer (July, August, September), and autumn (October, November, December). The interpretation of the seasonal components per se was not meaningful, therefore changes were not calculated, but when monotonic linear trends were significant (p<0.05), the sign and the percentage of the changes were provided.

Thus, the significance test of linear changes in DLM trends and seasonality components were provided by the modified MK test associated with Sen's robust line.

Table R1 below shows the coefficient of determination for each model parameter, which indicates of the goodness of fit of a model. It is estimated by calculating the square of the sample correlation coefficient between the observed outcomes and the observed predictor values. It can also be viewed as the ratio of the explained variance to the total variance. We also add the estimated significance of trends and seasonality based on the squared correlation coefficient between the calculated trend and deseasonalized data and on the squared correlation coefficient between the calculated seasonal component and detrended data respectively (Minaudo et al., 2015).

Table R1. Coefficient of determination, significance of trends and seasonality estimated for the period of 1997-2013

| Parameters | Overall (%) | Trend (%) | Seasonality (%) |
|---|---|---|---|
| Loire discharge | 94.6 | 52.8 | 94.1 |
| Loire DIP | 51.5 | 16.9 | 45.6 |
| Loire DIP Loads | 71.3 | 29.4 | 67.1 |
| Loire DIN | 87.9 | 38.9 | 87.1 |
| Loire DIN Loads | 92.6 | 60.7 | 91.8 |
| Loire DIN/DIP | 48.1 | 18.6 | 41.6 |
| Loire Chl *a* | 68.8 | 55.6 | 53.3 |
| Vilaine discharge | 96.1 | 65.9 | 95.7 |
| Vilaine DIP | 52.0 | 40.0 | 28.4 |
| Vilaine DIP loads | 71.5 | 48.4 | 60.1 |
| Vilaine DIN | 95.7 | 60.1 | 95.4 |
| Vilaine DIN loads | 95.2 | 62.4 | 94.7 |
| Vilaine Chl *a* | 59.0 | 44.4 | 43.3 |
| Vilaine DIN/DIP | 66.3 | 42.3 | 55.5 |
| VB DIP | 61.0 | 4.9 | 60.5 |
| VB DIN | 85.7 | 33.4 | 85.0 |
| VB DSi | 64.3 | 9.9 | 62.6 |
| VB DIN/DIP | 83.2 | 23.5 | 82.7 |
| VB DIN/DSi | 78.9 | 18.0 | 78.1 |
| VB DSi/DIP | 28.1 | 16.9 | 15.6 |
| VB Chl *a* | 58.8 | 26.7 | 51.0 |
| VB Diatoms | 48.5 | 6.9 | 46.7 |
| VB Dinoflagellates | 43.7 | 2.7 | 43.0 |

8. **RC** - 5;1: why was this log-transformation necessary? It needs justification.

   **AC** - The log transformation was necessary because:

   "many measurements show a more or less skewed distribution. Skewed distributions are particularly common when mean values are low, variances large, and values cannot be negative, as is the case, for example, with species abundance, lengths of latent periods of infectious diseases, and distribution of mineral resources in the Earth's crust. Such skewed distributions often closely fit the log-normal distribution (Aitchison and Brown 1957, Crow and Shimizu 1988, Lee 1992, Johnson et al. 1994, Sachs 1997)."

This subject has been deeply discussed in Limpert et al. (2001).

Log-normal distribution induces a variance to mean relationships, that is, as in our case the mean and the variance vary with time, and thus the homoscedastic hypotheses, i.e., specifically for us, the equality of error terms variance through time, may not be fulfilled. This is why "A variance stabilizing log transformation…" is applied in the first place.

In addition, since all of our variables are positive, treating them without log-transformation may lead confidence intervals to include negative values, which consequently leads to inadequate models.

We have added the following text in the **subsection 2.3.1, p.8, l.1-3**:

Prior to time series decomposition, a variance-stabilizing base e log transformation was applied to all variables, except for phytoplankton counts for which the base was 10, to ensure compliance with the constant variance assumption (i.e. homoscedasticity).

9.  **RC** - 5;7: reading log-transformed units is not convenient for the reader. You can log-transform the axis but still present actual values. Why did you log-transform the data in the first place? It makes the trend observation less clear to the reader.

    We have modified all figures of DLM trend and seasonality (**Figs. 2-7** & **Figs. S2-S7**) by log-transforming the *y*-axis to present the actual values. Please see an example Figure R2.

[Figure]

Figure R2. Trends of Chl *a* concentrations in the Loire

10.  **RC** - The authors decided to use units that are consistent in the manuscript, but not commonly used by researchers on lotic environment. Please, convert all mol/L into mg/L or µg/L

     **AC** - The SI unit of concentration (quantity of substance) is the mole per cubic meter (mol $m^{-3}$), which is commonly used in marine environment research. The use of "mol" is also consistent with the calculation of nutrient molar ratios, which permit us to assess potential nutrient limitation. Therefore, we prefer to keep as it is.

11.  **RC** - 5;8-9: the explanation on trends significativity test is not clear nor properly justified. You need a metric for this. Why not use Sen's Slope significativity test?

     **AC** - Please see the detail for trend significance test in point #7.

12.  **RC** - 5;11: the authors should define clearly which metrics were extracted for the time-series analysis and used for further analysis.

     **AC** - Please see point #7 for trend significance test.

13. **RC** - 5;15: have you conducted MK test on de-seasonalized = observations – seasonal component, or on de-seasonalized = trend component? The latter discards residuals from the analysis and this choice should be justified. Also, I think residuals from your DLM algorithm should be plotted along.

    **AC** - The modified Mann-Kendall test was conducted on de-seasonalized = trend component from DLM (please see point #7 for details).

    We provide residual plots for all parameters treated in this study in Appendix 1 for the consideration and verification of the referee. We did not include residual plots in results, as the number of figures was already high, as noted by the referee in point #21. Furthermore, papers including such figures are very uncommon.

14. **RC** - 5;17: it is not clear how you proceeded to identify seasonal trends. Did you use a seasonal MK test? This needs more details since it is the core of your analysis.

    **AC** - Please see point #7 for detail in seasonal trends.

15. **RC** - Besides, how would you justify analyzing loads evolutions and not only concentrations since you show that Q was stable over time? Removing all the load trajectory description would save space for other elements in your paper, and benefit to the clarity of your messages.

    **AC** - River discharge (Q) appeared stable in spite of oscillations. The modified MK test applied to river discharge trend component from DLM showed a significant decrease between 1997 and 2013 ($p<0.05$, Table 1). Therefore, it was necessary to calculate riverine nutrient loads in order to show that these loads displayed similar trends to those of nutrient concentrations.

16. **RC** - 5;21: What is STATGRAPHIC CENTURION and what are the metrics/analysis conducted with this? Please, add a reference for this.

    **AC** - In the manuscript section 2.4, we mentioned the use of STATGRAPHIC CENTURION software for Spearman Correlation analysis.

    The correlation analysis has been modified to take into account the annual median values of the common period 1997-2013 only. The text has also been modified by adding reference (**p.10, l.6-9**):

    Spearman Correlations were computed for annual median values of the common period in order to analyze relationships between variables, and tested using STATGRAPHIC CENTURION software (Statgraphics Technologies Inc., Version XVII, Released 2014).

17. **RC** - 5;26: how significant is this trend in Q data? A large slope in MK tests doesn't mean that it is statistically significant.

    **AC** - Table 1 shows a $p$-value of 0.014 (for significance level of 0.05) for river discharge, indicating a significant trend. The negative slope indicates a decreasing trend. This decrease in Loire discharge was also observed in previous studies (please see point #15)

18. **RC** - 6;5: this seasonal shift is not observable in Figure 3. Consider adding a Figure to show seasonal variations and evolutions.

    **AC** - The seasonal shift in position of annual DIP minimum from summer to spring is clearly visible by the change in color (Fig. 3b). It started with yellow (summer) around 1999 and changed to green (spring) from 2007 to the end of studied period.

A section dedicated to explain seasonality has been added. Please see point #7.

19. **RC** - In the Result section, it is good to refer to Tables and Figures, but the reader also needs actual values included in the text, otherwise he always has to go back and forth from text to Table/Figure.

**AC** - The values were already in the text accompanying the trend interpretation, except for loads

Actual values have been added into the text of the result section of the revised manuscript (**p.11-14**).

20. **RC** - 6;12: please, be more specific, and always use similar ways of describing the data: first, trends. Second, seasonal variations. It helps increasing the clarity of the manuscript and makes things easier for the reader.

**AC** - We described results as follows: 1) trends accompanied by actual data; 2) seasonality; 3) correlation.

21. **RC** - You have too many additional figures. Please, make a selection of the ones that are really useful to support your ideas.

**AC** - Our manuscript present results of two rivers and a coastal ecosystem. We decided to place the results of one of the two rivers in supplementary materials because they must be accessible for the reader.

22. **RC** - 7;19: add a section for this correlation analysis

The description of results has been modified (please see point #19-20).

23. **RC** - 7;24-29: Do you believe your DLM analysis is suitable for phytoplankton biomass description at the seasonal scale? You need to validate this first, and plots in Figure 7 don't help answer this question if you don't show residuals (you'll see that they don't look like white noise).

**AC** - We presume that these comments deal with phytoplankton abundances, not phytoplankton biomass. We do not understand the second sentence of the referee's comment. It is a time series and for different reasons (e.g.., missing data, irregular sampling frequencies, exceptional abundances), we believe that DLM, which has been used previously in published studies (see Soudant et al., 1997; Scheuerell et al., 2002; Hernández-Fariñas et al., 2014) is an appropriate tool for such data. Let us define white noise as values that are mutually uncorrelated with zero mean and have the same Gaussian probability distribution.

The residuals of diatom abundances are available in Appendix 1. The residuals QQ-plot of diatom abundances is presented below.

[Figure]

Figure R3. The QQ-plot of diatoms abundances:

The Kolmogorov-Smirnov *p*-value is equal to 0.2244. The Stoffer-Toloi (i.e., autocorrelation test) p-value is 0.1723. These results suggest that this is actually white noise.

24. **RC** - Insets in Figure 7 are not explained. It has to be.

    **AC** - The explanation is already in the legend.

    "Figure 7: Long-term trends and seasonality of Chl *a* (a, b), diatom (c, d), dinoflagellate (e, f) and diatom:dinoflagellate ratios (g, h) in VB. Insets show trends with optimal scale. See Fig. 2 for details"

    Results concerning diatom:dinoflagellate ratios have been removed. Thus, Figure 7 of the revised manuscript does not contain any more diatom:dinoflagellate ratios.

25. **RC** - Section 4: this section could be reorganized as follows 1) Nutrients and Chl-a trends at river basin outlets 2) Nutrients and Chl-a trends in the bay 3) River to bay continuum 4) Implications for management

    **AC** - The discussion section of the previous manuscript takes into account an unequivocal link between the bay and rivers, as shown in previous studies (e.g., Guillaud et al., 2008; Ménesguen and Dussauze, 2015; Ménesguen et al., 2018b). Therefore, the paragraphs described the variables (nutrients and phytoplankton) by grouping results from rivers with those from the Vilaine Bay. We understand that it was too hasty (i.e., continuum).

    We have made substantial modification and reorganization of discussion section of the revised manuscript (**p.14-24**) as follows:

    4 Discussion
      4.1 Eutrophication trajectories at the river basin outlet
      4.2 Eutrophication trajectories in the VB
        4.2.1 Increased Chl *a*
        4.2.2 Changes in timing of annual Chl *a* peak
        4.2.3 Role of DSi on seasonal course of diatoms and dinoflagellates
      4.3 Loire/Vilaine - VB continuum
        4.3.1 Rivers as the main external nutrient source to the VB
        4.3.2 Role of estuaries and the Vilaine dam
        4.3.3 Link between eutrophication trajectories in rivers and in the VB

26. **RC** - How does the estuarine zone could interfere in your interpretations? Same question with the presence of a dam at the outlet of the Vilaine river? This needs to be addressed, at list by listing the; different processes that occur. Many has been done on the subject.

We have added the following text explaining the potential influence of processes within estuaries and dam to **section 4.3** (Loire/Vilaine - VB continuum) of the revised manuscript (**p.20, l.9-20** & **p.21, l.1-12**):

4.3.2 Role of estuaries and the Vilaine dam

Biogeochemical processes within estuaries may alter the nutrient transfer from rivers to coastal waters (Statham, 2012). Coupled nitrification-denitrification and ammonification-anammox can be a sink of N in estuaries (Abril et al., 2000). Inorganic nutrients in estuaries can also be removed by phytoplankton uptake, which is nonetheless limited by turbidity (Middelburg and Nieuwenhuize, 2000). Estuaries can also act as a source of nutrients, resulting from mineralization of riverine phytoplankton organic matter (Meybeck et al., 1988; Middelburg et al., 1996). However, for the studied rivers, this process may have diminished with the decreasing trend in riverine Chl a. The desorption of loosely bound P from suspended mineral particles in estuaries can also be a source of DIP (Deborde et al., 2007). Except during flood periods, the suspended particle fluxes from the Loire are generally low (Moatar and Dupont, 2016). In addition to these biogeochemical processes, the increase in population around the Loire estuary (ca. 1% per year, INSEE, 2009) during the last decades could have contributed to the increase in N and P inputs. However, inputs of DIN and DIP from wastewater treatment plants in the Loire and Vilaine estuaries have not increased due to improved treatment techniques (Loire-Brittany River Basin Authority, P. Fera, pers. comm.). The presence of a dam at the river outlet may increase water residence time, thus favoring nutrient uptake by phytoplankton and loss of N via denitrification (Seitzinger et al., 2006). Unfortunately, for these two studied rivers, processes in estuaries and dam are poorly investigated and quantified, which makes it difficult to estimate their influence on nutrient transfer to coastal zone.

Despite influences of estuaries and dam, the increase in DIN:DIP and DSi:DIP ratios in rivers during last two decades, with values already largely above the theoretical value of 16 in the 1990s, has been reflected in the VB coastal waters (Figs. S5, S7). Moreover, significant negative correlations between annual Chl a medians in the VB and in rivers, as well as significant positive correlations between annual medians of DIN and DSi in the VB with those of river discharge suggest that changes in eutrophication parameters in the VB (i.e., phytoplankton biomass) were related to changes in rivers (Ménesguen et al., 2018a, b). Although biogeochemical processes in estuaries and the Vilaine dam may introduce bias in nutrient transfer from rivers to the VB, they are probably not intense enough to decouple the observed trends between rivers and the VB, as suggested by Romero et al. (2016) for the Seine River – Seine Bay continuum.

27. **RC** - 8;21: This can't be said like this. At the outlet of large and intensively managed catchments, nutrients variations are co-controlled by upstream hydrological variations,

delivery to stream modalities (point or diffuse sources?), and by instream retention processes through physical and biogeochemical processes.

AC - The lines that the referee has pointed out dealt with the variation in nutrient transfer from watershed to coastal waters, not the variations in nutrient concentrations in river waters: "The transfer of nutrients from continents to coastal waters is largely driven by freshwater inputs, the dynamics of which depend largely on precipitation in watersheds". Thus, in our opinion, our sentence was correct.

The sentence has been modified a little and can be found now in the subsection **4.3.1, p.19, l.16-18**:

The transfer of nutrients from continents to coastal waters is largely determined by freshwater inputs, the dynamics of which depend largely on precipitation in watersheds.

28. **RC** - 8;22: You should mention the North Atlantic Oscillation to explain the 7 years cycles. See also Dupas et al., 2018 (WRR)

AC - The inter-annual variability (i.e., oscillation of 6-7 years) of river discharges are related to precipitation regimes, which are modulated by climate (i.e., NAO for the North Atlantic region). The relationship between flow regimes, precipitation and NAO was explicitly detailed in Radach and Pätsch (2007). We supposed that we did not need to mention this point, which is not essential to support the main subject of the paper. However, for BGS we can include reference to NAO.

The sentences corresponding to this subject has been modified and can be found in the subsection **4.3.1, p.19, l.18-19** & **p.20, l.1**:

Trends in the Loire and the Vilaine discharges displayed similar oscillations to those of rivers flowing to the North Sea as reported by Radach and Pätsch (2007), suggesting a common hydro-climatic pattern in Western Europe linked to the North Atlantic Oscillation.

29. **RC** - Figure 8 could be a great final figure, but needs to be explained once the processes explaining the different patterns in eutrophication metrics are completely described.

The legend of Figure 8 has been modified as follows:

Graphical representation of the major changes in phytoplankton and nutrient concentrations in rivers (a, b) and the VB coastal waters (c, d) for the period 1997-2005 (top frame) and 2006-2013 (bottom frame). Downward arrows represent long-term trends. Nutrient curves are ranked from the least limiting (bellow) to the most limiting (above) according to Redfield ratios. Nutrient inputs from rivers and sediments are also ranked according to their potential limitation for phytoplankton using Redfield ratios. Benthic nutrient inputs were fitted according to the measurement of benthic fluxes in summer 2015 (Ratmaya, 2018). Shaded areas underline the season of maximum Chl *a*

**Technical comments**

30. **RC** - Page 1, Line 14 (1;14): remove "(i.e., phytoplankton biomass)", as eutrophication expression is not only phytoplankton excessive biomass.

The sentence has been modified as follows (**p.1, l.14-15**):

The evolution of eutrophication parameters (i.e., nutrients and phytoplankton biomass) during recent decades was examined in coastal waters of the Vilaine Bay (VB, France) in relation to those in the Loire and Vilaine Rivers.

**31.** **RC** - 2;5-7: this has to do with different source types and it should be explained. Environmental measures to tackle P were successful because P largely originated from point sources with limited legacy effects in the streams. For N, diffuse sources dominate and there is large legacy effect.

**AC** - The information suggested by the referee has been extensively explained in the cited articles. We mentioned the question of different source types and legacy effects in the discussion section 4.3.

**32.** **RC** - 2;7-11: you should also mention freshwater ponds and lakes were eutrophication is still severe despite large P reductions.

**AC** - We can mention it, but this is not the main subject of the study. We would like to focus on eutrophication in coastal ecosystem, by highlighting the needs to reduce both P and N loads to mitigate eutrophication along the land-sea continuum.

We have modified the introduction section and added the information mentioned by referee (point #31-32) as follows (**p.2, l.16-19** & **p.3, l.1-11**):

Since the beginning of the 1990s, measures to reduce nutrient inputs in European rivers were more effective for P, originating largely from point sources, than for N, coming mainly from diffuse sources (Grizzetti et al., 2012). However, this strong imbalance between N and P input reduction still led to substantial decrease in phytoplankton biomass in many European rivers (Istvánovics and Honti, 2012; Romero et al., 2013). This result is consistent with the idea that P universally limits primary productivity in many freshwater ecosystems (Correll, 1999). Thus, reducing P inputs, and not N, can mitigate eutrophication of freshwater ecosystems (Schindler et al., 2008; Schindler et al., 2016).

Despite significant P input reduction, eutrophication persists in some rivers (Bowes et al., 2012; Jarvie et al., 2013), and particularly in downstream coastal ecosystems, where the primary productivity is often limited by N (Ryther and Dunstan, 1971; Paerl, 2018). As freshwater systems drain into coastal waters (Vannote et al., 1980), the efficient P reduction without simultaneous N abatement may result in more N being transported downstream, where it can exacerbate eutrophication problems in coastal ecosystems, delaying recovery (Paerl et al., 2004), for example the Neuse River Estuaries (Paerl et al., 2004), Belgian coastal waters (Lancelot et al., 2007), and the Seine Bay (Romero et al., 2013). Despite more than 20 years of nutrient reduction implementation in European freshwater ecosystems, including rivers (e.g., Nitrates Directive, 91/676/EEC; Urban Waste Water Treatment Directive, 91/271/EEC), little measurable progress has been observed in many European coastal waters (EEA, 2017; OSPAR, 2017).

**33.** **RC** - 2;16-17: I'd remove the codes for what you called "water masses". Do the authors mean "water body"?

The codes have been removed and the sentences have been modified as follows (**p.3, l.14-18**):

Affected by the Loire and Vilaine river runoff (Guillaud et al., 2008; Gohin, 2012; Ménesguen et al., 2018b), the Vilaine Bay (VB) is one of the European Atlantic coastal ecosystems most sensitive to eutrophication (Chapelle et al., 1994; Ménesguen et al., 2019). The VB coastal waters are classified as a problem area due to elevated phytoplankton biomass, according to the criteria established within OSPAR (OSPAR, 2017) and the European Water Framework Directive (Ménesguen et al., 2018b).

34. **RC** - 2;18: an actual scientific reference would be better.

The sentence has been modified as above (point #33).

35. **RC** - 3;3: "widest" is not correct. You may refer to "largest river basin".

**AC** - The use of the word "widest" refers to the river Loire and not to the river basin. This usage has been confirmed by our English native.

36. **RC** - 3;4: sentence is not clear, please, rephrase it.

The sentence has been rephrased as follows (**p.4, l.5** & **p.5, l.1-2**):

Their catchment areas are dominated by agricultural activity, together sustaining two-thirds of the national livestock and half the cereal production (Bouraoui and Grizzetti, 2008; Aquilina et al., 2012).

37. **RC** - 3;22: sentence is not clear, please, rephrase it.

The sentence will be rephrased in the next version as follows (**p.6, l.12-14**):

Nutrient and Chl *a* concentrations, plus phytoplankton count data in the VB, provided by the French National Observation Network for Phytoplankton and Hydrology in coastal waters (REPHY, 2017), were collected from Ouest Loscolo station (Fig. 1).

38. **RC** - 5;1: the use of ":" separates the sentence in a way that makes it hard to understand. Please, modify.

The ":" has been removed and the sentence has been modified. Please see point #8 for the modification.

39. **RC** - 5;8-9: check for use of different tenses throughout the manuscript.

The sentence has been removed. The time-series analyses section has been modified. Please see point #7 for the modification.

40. **RC** - 5;23: Change this section title to "Discharge and nutrients long term trends in freshwater basin outlets"

The title suggested does not take into account the Chl *a*. We propose a modification to the section title as follows (**p.10, l.11**):

Long term trends in eutrophication parameters in river basin outlet

**Referee 2 – Anonymous referee**

**Referee's comments (RC)** - This paper studies the long term trends in nutrient and phytoplankton dynamics in the Loire and Vilaine rivers, and in the Vilaine Bay (VB). The authors discuss changes in eutrophication of these systems, and relate changes in the VB to those in nutrient inputs from the two rivers. They show that, even though phytoplankton blooms decreased in the riverine systems following reduction in dissolved inorganic P, phytoplankton biomass in the VB has continued to increase. This could be fueled by nitrogen delivery from the rivers (slightly increasing trend for the Loire), together with phosphorus and silica recycling from bottom sediments in the coastal area. This is an interesting discussion point, that totally fits Biogeosciences' scope. This is however only superficially discussed, and the layout of the paper makes it difficult to identify the main conclusions. I also noted important gaps in the methods' description.

The presentation of the river trajectories is extensive, but was already thoroughly discussed in a previous study (Minaudo et al., 2015). Very complete time series of Chla concentrations and abundances of different phytoplankton species in the VB are presented, and could be extremely valuable to examine changes in community structure. However, these are not discussed in depth. Moreover, more elements should be provided to the reader to justify that the data presented here is enough to support the conclusions of the study. In fact, interpretations of the dynamics in the VB are derived from observations at a single point, at which the influence of the Loire river is not obvious and not discussed.

I believe these major shortcomings should be addressed before this work can be published.

**General comments**

1. More information on the influence of the Loire river on the VB dynamics is needed. In fact, nutrients need to travel more than 120km from the Loire river monitoring station (Saint Luce sur Loire) to the Bay, through the Loire estuary and along the coast. Do coastal currents carry most of the Loire river's exports to the VB? How can processing in the estuary and along the coast impact loads reaching the VB?

2. Methods on the Dynamic Linear Models (DLM) and Mann-Kendall (MK) test analysis are not detailed enough. I am also not convinced that the MK test provides any more information than the DLM analysis. To my understanding, numerical estimates on trends and seasonal variations can also be extracted from the latter. Using these two methods to come up with the same interpretations waters down important messages in the results and discussion sections.

3. Authors refer several times throughout the manuscript to "management scenarios focused solely on P reduction" or on "P alone". However, this is not totally accurate for the study area, and should be moderated. Even though ecosystems responded quicker to P reduction strategies (e.g. for point sources) than to policies on agricultural fertilization, those already exist (e.g. EU Nitrates Directive).

4. In general, statements are sometimes vague or not totally accurate. The structure of the results and discussion sections makes it difficult for the reader to identify the main conclusions of the study.

These points, together with more minor concerns, are more detailed hereafter, in the specific comments.

**Author's comments (AC)** - We thank the referee for the detailed analysis and constructive comments. For this referee, the link between the Loire River and the VB was not well enough established in spite of numerous works cited in the previous version. We will add the information

on the continuum to the revised version. In the case of the Loire/Vilaine Rivers – the VB continuum, we recognize that there was no formal decision to reduce P-only. However, the small decrease of DIN concentrations in the Vilaine and their increase in the Loire especially in summer during recent decades provide a scenario that allows testing the P-only paradigm.

All issues raised by the referee are carefully answered point by point below.

**Specific comments/scientific questions**

1.  **RC** - L7-8, P2. "This result is consistent with the idea that reducing P alone, and not N, can mitigate eutrophication of freshwater systems (Schindler et al., 2008)": This paper from Schindler et al. does not show this; they study the effect of reducing N only. Moreover, this is not a scientific consensus (e.g. Pearl et al., 2016, Environ. Sci. Technol. 50, pp 10805–10813). This sentence should be moderated.

    **AC** - It is true that these authors also studied the reduction of N in their lake, but it was to support the hypothesis that the reduction of P can be enough for lake restoration. At the end of their summary, the authors also stated that to reduce eutrophication, the focus of management must be on decreasing inputs of P. In a more recent article, Schindler et al. (2016) clearly argued for a reduction of P alone to control eutrophication in lakes and other freshwater ecosystems, even though they recognize that anthropogenic nitrogen emissions can also affect human health and ecosystems (i.e., Box 2).

    These lines of introduction section have been modified in the revised manuscript. Please see point #32 of the referee #1 comments for detailed modification.

2.  **RC** - L14-15, P2. "Nutrient inputs ...control phytoplankton production in coastal waters of the northern Bay of Biscay": Riverine inputs constitute the major nutrient source, but don't necessarily control phytoplankton dynamics. Guillaud et al. (2018) show that sediments have a high influence on Chla levels as well (light limitation in high flow periods/winter).

    **AC** - It is true that there are also environmental conditions allowing nutrients to be consumed by primary producers, such as water residence time, light availability, etc. The primary production in coastal waters off the Loire and Vilaine River is limited by light availability due to insufficient irradiance during winter and suspended sediment flux from rivers and resuspension during the period of high hydrodynamic activity (Guillaud et al., 2008). However, these authors also showed that, except during periods of light limitation (November - February), phytoplankton blooms in this area respond to the variation in river discharge.

    These lines of introduction have been removed and modified as follows (**p.3, l.12-18**):

    The Loire River, alongside the Vilaine River, are among these major European rivers whose phytoplankton biomass and P concentrations have decreased since the early 1990s, but with minor, if any, simultaneous diminution in N concentrations (Romero et al., 2013; Minaudo et al., 2015). Affected by the Loire and Vilaine river runoff (Guillaud et al., 2008; Gohin, 2012; Ménesguen et al., 2018b), the Vilaine Bay (VB) is one of the European Atlantic coastal ecosystems most sensitive to eutrophication (Chapelle et al., 1994; Ménesguen et al., 2019). The VB coastal waters are classified as a problem area due to elevated phytoplankton biomass, according to the criteria established within OSPAR (OSPAR, 2017) and the European Water Framework Directive (Ménesguen et al., 2018b).

    We have also added sentences explaining the influence of turbidity on phytoplankton bloom in the Material & Methods **section 2.1, p.5, l.13-16:**

    During periods of prevailing winds, particularly from south-west and west, the water column of the VB is subjected to vertical mixing, which can lead sometimes to sediment resuspension

and high turbidity (Goubert et al., 2010). Except during winter and period of high hydrodynamic activity, phytoplankton production in the VB is not limited by light (Guillaud et al., 2008).

..and in the discussion **subsection 4.2.1, p.17, l.20** & **p.18, l.1:**

In the VB, except during winter and high hydrodynamic activity periods, phytoplankton production is limited by nutrients (Guillaud et al., 2008).

3. **RC** - L22-24, P2. Consider adding references to support this.

These lines have been modified as follows and references have been added (**p.3, l.18-19** & **p.4, l.1-3**)

However, there is little information on how eutrophication parameters have evolved in the VB over the past 20 years in the light of eutrophication mitigation in the Loire and Vilaine Rivers. An approach taking into account seasonal variations is required as phytoplankton in many coastal ecosystems, such as the coastal waters off the Loire and Vilaine Rivers, is often limited by P in spring and by N in summer (Lunven et al., 2005; Loyer et al., 2006).

4. **RC** - L9, P3. "The VB...is located under direct influence of these two rivers": This is not really clear from Fig. 1. See general comment 1.

**AC** - The link between the Loire inputs and dynamics of the coastal waters of the Northern Bay of Biscay, including the Vilaine Bay, has been established using ecological model ECO-MARS3D (see Ménesguen et al., 2018b). These authors showed in their figure 6 the influence area of several large French Atlantic river plumes during three different flow regimes. It also showed that the VB coastal waters are always affected by the Loire river plume whatever the regime scenario. This can justify the link between the Loire River and the VB (i.e., continuum).

The following text has been added to the revised manuscript to explain the contribution of the Loire and Vilaine Rivers to the VB fertilization (**section 2.1, p.5, l. 5-9**):

The Loire river plume tends to spread north-westward with a dilution of 20 to 100-fold by the time it reaches the VB (Ménesguen and Dussauze, 2015; Ménesguen et al., 2018b). The ECO-MARS3D model estimates that the Loire constitutes >60% of VB DIN concentrations during flood regimes and from 20 to 40% during low discharge periods (Gohin, 2012; M. Plus, Ifremer Brest, pers. comm.). The Vilaine river plume tends to spread throughout the bay before moving westward (Chapelle et al., 1994).

5. **RC** - L4-5, P4. The link between the first two sentences of this paragraph is not clear.

These lines have been modified as follows (**p.6, l.2-3**):

DIN was defined as the sum of nitrate, nitrite and ammonium, with nitrate as the major component (>90%).

6. **RC** - L25, P4 – L6, P5. This paragraph would benefit from more explanations on the DLM method. When you say "look like interpolation", do you mean it is equivalent to interpolation? If yes, which kind of interpolation?

**AC** - The sequential DLM approach is provided by the Kalman filter, by identifying the missing values and replacing them with normal random variables. This approach may be viewed as one that uses a prior for the parameter which replaces the missing values. This is another way to say "absence of data leads to no change in distributions for model parameters".

The section concerning time series analysis has been substantially modified and now found in the **section 2.3, p.7-10** of the revised manuscript.

The sentence corresponding to this subject has been replaced by the following (**p.8, l.9-12**):

The DLM approach is particularly suitable for environmental data series characterized by outliers, irregular sampling frequency and missing data. The latter are taken into account by the Kalman filter (Kalman, 1960), using a prior which replaces the missing value, i.e., no information leads to no change in distributions for model parameters (West and Harrison, 1997).

**RC** - Why do you choose to fit second order polynomial functions for the trends, and bimodal trigonometric functions for the seasonality? Is it based on any preliminary analysis of the data?

**AC** - We choose a second order polynomial model because looking at the log-transformed time series it appeared to us that a first order (i.e., adapting trend up to linear) was too restrictive and a third order (i.e., adapting trend up to cubic) was not necessary, leading to an over fitted model.

Yes it is based on preliminary analysis of the data. In the VB area, the annual patterns of phytoplankton variability have a six months periodicity. This bimodal pattern is characterized by two peaks per year, such as spring and autumn or summer and winter blooms. In order to allow our model to adapt to such periodicity we have to include a two harmonics seasonal component.

The following sentences have been added to justify the choice of the model (**p.8, l.14-18**):

The model used was a second order polynomial trend, which allows modelling up to quadratic trend. This was chosen because linear trend (i.e., first order polynomial) was too restrictive and cubic trend (i.e., third order polynomial) might lead to an over fitted model. For the seasonal component, the model used was trigonometric with two harmonics, which allows modelling up to bimodal pattern. This bimodal pattern is characterized by two peaks per year, such as spring and autumn or summer and winter blooms. This model specification was used for all parameters.

**RC** - What does "time units" refer to? Is it the frequency at which the trends/seasonal variations are estimated?

**AC** - Time unit is the smallest time interval between sampling dates within a period of analysis. In our case, the period is one year. Time units are weekly, fortnightly or monthly depending on the data. Seasonal variations are estimated for each time unit.

More information about time units has been added to the revised manuscript.

**p.9, l.1-2**: The time unit, defined as the smallest time interval between sampling dates within a period of analysis (i.e., one year), was weekly, fortnightly or monthly according to sampling frequencies of variables (see Table S1).

**p.9, l.8-9**: The DLM trend plot displayed observed values with a shade of color for each time unit segments: weekly, fortnightly or monthly.

**RC** - Why are those plotted with (two different types of) log scales? It makes it more difficult to link the figures with the values provided in text.

**AC** - Y-axis of all graphics has been modified to show original units using a log-scaled y-axis.

All DLM trend and seasonality figures have been modified (**Figs. 2-7** & **Figs. S2-S7**) by log-transforming the *y*-axis to present the actual values. Please see point #9 of the referee #1 comments for an example.

7. **RC** - L14-19, P5. What extra information does the MK test provide? Trend values can already be extracted from the DLM analysis. Is the method applied to the trend/seasonality functions from the DLM analysis, or to the raw data? Are uncertainties accounted for?

    **AC** - The modified MK test was used as a formal trend significance test. The test was applied respectively to trend and seasonality components from DLM, not to the raw data.

    Yes, uncertainties are taken into account by the DLM.

    We have modified the section of time-series analyses and added more detail of the use the modified MK test in the revised manuscript. Please see point #7 of the referee #1 comments.

8. **RC** - L19, P7-L7, P8. Results on Chla concentrations and phytoplankton species in the VB are not thoroughly presented here. It seems from the seasonality plot that, in the timeframe of the study, Chla has always peaked in spring and summer, and that since 2006 the summer peak has reached similar concentrations to the spring one. It's also interesting to note that there seems to be a succession of 3 algae blooms: a diatom bloom in spring, a dinoflagellate one in early summer, when DSi is depleted, and another diatom one in late summer.

    **AC** - In this study, we used only the total counts of diatoms and dinoflagellates, to account for the role DSi. These two groups represent >85% of total micro-phytoplankton counts and thus the biomass (**section 2.2, p.7, l.3-7**).

    This change in Chl *a* seasonality was mainly due to the increase in summer diatom abundances and the decrease in spring ones, as suggested by their seasonality.

    The seasonal pattern of phytoplankton blooms was characterized by a diatom bloom in spring corresponding to high river flows and another one in late summer. Dinoflagellates tend to increase in summer but their abundances remain largely lower than those of diatoms, except during discolored water events (Souchu et al., 2013; Sourisseau et al., 2016). The collapse of spring diatom bloom is due more to DIP depletion than DSi (please see point #11).

9. **RC** - L21-22. P8. Why would trends in discharge in the studied rivers depend on variations in the precipitation in river basins flowing to the North Sea?

    The sentences have been modified and now found in the **4.3.1, p.19, l.18-19** & **p.20, l.1**:

    Trends in the Loire and the Vilaine discharges displayed similar oscillations to those of rivers flowing to the North Sea as reported by Radach and Pätsch (2007), suggesting a common hydro-climatic pattern in Western Europe linked to the North Atlantic Oscillation.

10. **RC** - L30. P8-L4. P9. This paragraph would be more convincing if estimates of the loads from the different sources were provided. Is the Loire "probably" the major nutrient source, or has it been shown that it actually is? How much water/nutrients are retained in the Arzal dam, and how does it influence the loads reaching the VB? Are the discharge and loads from the Vilaine really negligible in summer, even though it flows directly to the Bay, while the Loire river plume has to travel 120km?

    **AC** - According to the modelling study, the Loire is actually the major nutrient source (please see point #4).

    Concerning the Vilaine, during the period of low water discharge (10 – 100 m$^3$ s$^{-1}$), the dam is closed at high tide. The small releases due to the lock functions of the dam (shipping, fishway and salt-water pump) represent half of the "natural" discharge (Traini et al., 2015). The dam is closed below 10 m$^3$ s$^{-1}$. The summer Vilaine discharge measured at Rieux displayed strong variations, ranging between <1 to 100 m$^3$ s$^{-1}$, with >95% of values below 60 m$^3$ s$^{-1}$. Therefore, we consider that half of water discharge during the low water period was retained by the dam.

Unfortunately, we do not have any measurement of nutrient concentrations at the dam outlet nor inside the dam, which could be used to estimate the nutrient retention. The presence of a dam at the river outlet may increase water residence time, thus favor nutrient uptake by phytoplankton and loss of N via denitrification (Seitzinger et al., 2006). The presence of Arzal dam may thus attenuate nutrient transport from the Vilaine River to the VB. The use of loads calculated from Rieux station has likely overestimated nutrient loads from the Vilaine River.

We have modified the sentences mentioned by the referee and can be found in the revised manuscript in the **subsection 4.3.1, p.20, l.4-8**:

However, with a tenfold higher discharge than the Vilaine, the Loire remains the main source of freshwater for the northern Bay of Biscay, with a major role in the eutrophication of coastal waters in south Brittany, including the VB (Guillaud et al., 2008; Ménesguen et al., 2018a, 2019). Aside from flood periods, the closure of the Arzal dam during the low-water periods (Traini et al., 2015), makes nutrient inputs into the VB by the Vilaine negligible in summer, compared to those from the Loire.

11. **RC** - L9. P9-L20. P9. The phytoplankton succession is not thoroughly discussed here. See Specific comment 8. Even though they are decreasing, spring diatom abundances are still superior to summer ones. It is mentioned that temperature changes can induce shifts in species' succession. Is it the case here?

**AC** - Concerning phytoplankton succession, please see point #8. This is true that the spring diatom abundances remain higher than summer ones. The point that we would like to highlight here is the increase in summer diatom abundances, accompanying the increase in summer Chl *a* as the indicator of the VB degradation. An increase in winter sea surface temperature has been reported in the continental shelf off the Loire and Vilaine Rivers (Désaunay et al., 2006). Thus, the change in the timing of annual maxima observed for phytoplankton biomass could not be attributed to the increase in temperature and was better explained by changes in nutrient loads. Moreover, changes in phytoplankton community structure at species level, in relation to changes in environmental parameters (e.g., temperature), will be examined in other study.

The possible influence of temperature on phytoplankton biomass is discussed in the **subsection 4.2.1, p.17, l.9-20** & **p.18, l.1-3** of the revised manuscript (see text below)

It would also be interesting to discuss the relationship between phytoplankton successions and variations in DSi, for example.

In the coastal waters off the Loire and Vilaine Rivers, including the VB, the phytoplankton bloom is generally limited by DIP in spring and by DIN in summer (Loyer et al., 2006; Guillaud et al., 2008). This pattern has been verified by bioassay (M. Retho, Ifremer 2015, unpublished data). On the basis of DIP and DSi concentrations, and DSi:DIP ratios, the diatoms are rarely limited by DSi. The decrease in DIP riverine loads has increased the DSi:DIP ratios in the VB during the past decades (Fig. S7) and reinforced therefore the DIP limitation in the VB, as suggested by Billen et al. (2007) for the Seine River – the Seine Bay continuum.

The relationship between DSi and the seasonal course of diatoms and dinoflagellates is discussed in the **subsection 4.2.3 p.18, l.15-19** & **p.19, l.1-7** of the revised manuscript (see text below)

The discussion section has been substantially modified, reorganized and extended (**p.14-24**). Also see point #25 of the referee #1 comments. The eutrophication trajectories in the VB can be found in the revised manuscript in the section 4.2 (**p.17-19**).

4.2 Eutrophication trajectories in the VB

[revised manuscript text omitted]

12. **RC** - L20. P10. Does Table S2 show values for the Bay of Biscay or for the Ouest Loscolo station only?

    **AC** - No, it shows global annual median values for the Bay of Biscay, as detailed in the table legend.

13. **RC** - L21-23. P10. Precise that these correlations are at the annual scale. Seasonal variations of DIN and DSi do not seem correlated.

    **AC** - The precision of correlation analysis has been given in the end of the section 2.4. We explained that we used annual median values to compute Spearman's Rank correlation analysis.

    As mentioned in Table 3, annual medians of DIN and DSi in the VB were correlated.

    The sentence has been modified and replaced in the **subsection 4.3.2, p.21, l.7-10**.

    Moreover, significant negative correlations between annual Chl a medians in the VB and in rivers, as well as significant positive correlations between annual medians of DIN and DSi in the VB with those of river discharge suggest that changes in eutrophication parameters in the VB (i.e., phytoplankton biomass) were related to changes in rivers (Ménesguen et al., 2018a, b).

    The correlation analysis has been modified to take into account only the annual median values of the common period 1997-2013 (please see also point #16 of the referee #1 comments). Thus the Table 3 and its legend have also changed.

    Table 3: Spearman's rank correlations between annual median values of river discharge, nutrient concentrations and phytoplankton biomass in the Loire, Vilaine and the VB for the common period 1997-2013 (n = 17). Asterisks designate significant correlations (***p<0.001, **p<0.01, *p<0.05)

14. **RC** - L5-16. P11. Please provide some numbers to support your conclusions.

These lines have been modified and replaced in the **subsection 4.3.4, p.22, l.7-17**.

4.3.4 Role of internal nutrient loads

In shallow ecosystems, internal nutrient recycling can regulate phytoplankton production and potentially exacerbate eutrophication (Paerl et al., 2016), as observed both in lakes (Jeppesen et al., 2005) and coastal ecosystems (Pitkänen et al., 2001). Compared to freshwater, the fragility of marine ecosystems is related to salinity (Blomqvist et al., 2004). The presence of sulfate a major element of salinity) decreases the efficiency of sediments to retain DIP (Caraco et al., 1990; Lehtoranta et al., 2009) and favors the recycling of DIP over DIN, the latter being potentially eliminated through denitrification (Conley, 2000; Conley et al., 2009). In the VB, measurements of benthic nutrient fluxes confirm that sediments represent a substantial DIP and DSi source compared to riverine inputs (Ratmaya, 2018), allowing summer phytoplankton production to benefit from surplus DIN inputs from the Loire. Consequently, the increase in summer diatom abundances in the VB was mainly due to increased summer DIN loads from the Loire, sustained by internal sources of DIP and DSi coming from sediments.

15. **RC** - L9-12. P12. An opening on eutrophication and its mitigation would fit better. regarding the introduction.

**AC** - The conclusion section will be modified following modification made in discussion section.

The conclusions and perspectives section has been modified as follows (**p.24, l.12-18** & **p. 25, l.1-10**):

Parallel investigation of eutrophication parameters in the Loire and Vilaine Rivers, and coastal waters under their influence revealed several striking patterns and relationships, of which the most apparent was upstream recoveries from eutrophication accompanied by increased eutrophication downstream (Fig. 8). During the last two decades, Loire-Vilaine coastal waters have experienced a diminution in DIP inputs, whereas DIN continued to increase in the Loire during summer. While the decreasing trends in DIP were accompanied by declining phytoplankton biomass in rivers, the seasonal cycle of phytoplankton has been changed in downstream VB, with an increase in biomass, a shift in its annual peak from spring to summer, and a modification in the seasonal course of diatoms and dinoflagellates. Moreover, the concept of diatom replacement by dinoflagellates during the eutrophication process does not seem to be applicable to all shallow coastal ecosystems.

These results open up a whole field of investigation into the effects of changes in the phytoplankton dynamics on food webs, which is of major importance to this flatfish nursery and commercial shellfish area (Désaunay et al., 2006). Further studies are necessary to investigate the modifications in the phytoplankton community, especially the phenology of the different species, as well as the possible consequence on food webs. Finally, the internal loads of nutrients from sediments are suspected of counteracting the reduction of external nutrients, thus delaying the restauration progress. During the eutrophication process, sediments may also play an important role in the balance between diatoms and others classes of phytoplankton. Taking into account these internal processes in modelling studies (i.e., ECO-MARS3D, Ménesguen et al., 2018a, b, 2019), will better simulate nutrient load scenarios in shallow coastal bays (work in progress).

16. Table S1. When different measurement methods were used for a same variable. consider indicating which time period corresponds to which method.

**Wording**

**RC** - Throughout the text: "Vilaine Bay/VB" -> "the Vilaine Bay/VB"

It has been corrected throughout the text.

**RC** - L15. P1. "in relation to those in their..." -> "in relation to changes in its"?

The sentence has been modified as follows (**p.1, l.14-15**):

The evolution of eutrophication parameters (i.e., nutrients and phytoplankton biomass) during recent decades was examined in coastal waters of the Vilaine Bay (VB, France) in relation to changes in the Loire and Vilaine Rivers.

**RC** - L4. P2. "myriad responses" -> "myriad of responses"

**AC** - The use of "myriad responses" was based on Cloern (2001) and it has been validated by our English native.

**RC** - L15-18. P4. "The removed...general trend observed": Please reformulate.

The sentence has been modified as follows (**p.7, l.15-18**):

The removed DIP datasets represented 29% and 31% of the total number of data, corresponding respectively to the period 1980-1989 in the Loire, and 1980-1989 and 2009-2011 in the Vilaine.

**RC** - L10&12. P5. "position of" -> "timing of"

It has been corrected throughout the text. This section has been modified in the revised manuscript. Please see point #7 of the referee #1 comments.

**RC** - L14. P9 & L8. P12. "course" -> "succession

**AC** - "succession" is not the right term in our case (please see point #8). Thus we would keep using "course".

**Appendix 1.** Residual plots for all parameters used in the present study

| Figures | Parameters | Residual plots |
|---|---|---|
| **2** | Loire discharge |
[Figure]
 |
| **3** | Loire DIP | |
| **3** | Loire DIP Loads | |
| **4** | Loire DIN | |
| **4** | Loire DIN Loads | |

| Figures | Parameters | Residual plots |
|---|---|---|
| **5** | Loire Chl *a* |
[Figure]
 |
| **6** | Vilaine Bay DIP | |
| **6** | Vilaine Bay DIN | |
| **6** | Vilaine Bay DSi | |
| **7** | Vilaine Bay Chl *a* | |
| **7** | Vilaine Bay Diatom | |

[Figure]

| Figures | Parameters | Residual plots |
|---|---|---|
| **7** | Vilaine Bay Dinoflagellate | LL=, -97.16, AIC=, 202.32, Séquence(s)=, 12-1, 10-1, 9-3, 8-1, 7-5 |
| **Supp 2** | Vilaine discharge | LL=, -62.86, AIC=, 133.72, Séquence(s)=, |
| **Supp 3** | Vilaine DIP | LL=, -28, AIC=, 64, Séquence(s)=, 16-1, 10-1, 9-1, 8-1, 7-3 |
| **Supp 3** | Vilaine DIP loads | LL=, -340.7, AIC=, 689.4, Séquence(s), 7-4 |
| **Supp 4** | Vilaine DIN | LL=, 337.08, AIC=, -666.16, Séquence(s)=, |
| **Supp 4** | Vilaine DIN loads | LL=, -244.72, AIC=, 507.44, Séquence(s)=, 7-1 |

| Figures | Parameters | Residual plots |
|---------|-----------|----------------|
| **Supp 5** | Loire DIN/DIP | |
| **Supp 5** | Vilaine DIN/DIP | |
| **Supp 6** | Vilaine Chl *a* | |
| **Supp 7** | Vilaine Bay DIN/DIP | |
| **Supp 7** | Vilaine Bay DIN/DSi | |
| **Supp 7** | Vilaine Bay DSi/DIP | |

[Figure]

**Additional changes made in the manuscript**

1. **Abstract**: We have added text concerning potential influence of internal nutrient loads on eutrophication management strategies (**p.2, I.7-9**).

   Internal benthic DIP and DSi recycling appears to have contributed to the worsening of summer VB water quality, augmenting the effects of anthropogenic DIN inputs. For this coastal ecosystem, nutrient management strategies should consider the internal nutrient loads in counteracting decreased external inputs.

2. **Introduction**: An introduction on the possible effect of increasing N and P inputs, but not Si to coastal waters, on the modification in the phytoplankton community, with a decrease in diatoms for the benefit of non-siliceous algae (**p.4, I.4-8**).

[revised manuscript text omitted]
 | 0.11 | 0.17 | -0.14 | -0.48 | -0.61* | -0.34 | -0.58* | -0.50* | 0.25 | -0.45 | 0.33 | 1.00 |

---

## Author Response (AR3)

Dear editor,

We would first like to thank the Associate Editor and reviewer (Dr. C. Minaudo) for accepting the revised manuscript.

Please find enclosed:

1.  Responses to referees' technical corrections. Added changes in the manuscript are shown in **red**, in which page and line numbers refer to the revised manuscript: **pages 1-3**,

2.  Revised manuscript with MS Word track changes functionality: separate page numbering, **pages 1-48**. The revised manuscript with accepted changes is attached in the separate file.

As agreed, we also added new co-author (Dr. Martin Plus, [martin.plus@ifremer.fr](mailto:martin.plus@ifremer.fr)) and changed the order as follows:

Widya Ratmaya, Dominique Soudant, Jordy Salmon-Monviola, **Martin Plus**, Nathalie Cochennec-Laureau, Evelyne Goubert, Françoise Andrieux-Loyer, Laurent Barillé, Philippe Souchu.

We sincerely thank you for your consideration.

On behalf of the co-authors,

Widya Ratmaya

**Reviewer 1. C. Minaudo**

**Technical correction**

1.  Page 2 line 2: change "The long-term trajectories of this case study provide a more evidence..." to "The long-term trajectories of this case study evidence..."

    Done

2.  Page 2 lines 5-6: "...especially during the period of N limitation". Please, be more specific on the period you mention, or change the sentence to "...especially when N is the limiting factor".

    Done

3.  Page 2 line 9: "...should consider the internal nutrient loads in counteracting decreased external inputs" is unclear. I suggest this sentence: "...should consider the role played by internal nutrient loads to tackle eutrophication processes"

    Done

4.  Page 4, line 17: "The Loire is the longest and widest river in France (1,012 km)...". Authors either have to also specify river width, or should remove "and widest".

    Widest was removed

5.  Page 5 line 1: please, delete "together"

    Done

6.  Page 5 line 4: change "in the northern Bay of Biscay, including VB" into "in the northern Bay of Biscay, which includes VB"

    Done

7.  Page 5 line 9: "The Vilaine river plume tends to spread throughout the bay before moving westward (Chapelle et al., 1994)." If there is no estimates of the contribution of the Vilaine to the VB, this has to be specified to be consistent with the text just above on the Loire River contribution.

    We added the information of the Vilaine contribution to VB DIN, also both river contribution to VB DIP concentrations. Text was modified as follows p.5, l.6-12:

    The model ECO-MARS3D estimates that the Loire constitutes, on average, 30% of VB DIN concentrations during floods (min-max 2-49%) and 18% during periods of low water (min-max 6-29%). The contributions of the Loire to VB DIP can reach 36% during flood periods (average 12%) but remains <5% during periods of low water. The Vilaine river plume tends to spread throughout the bay before moving westward (Chapelle et al., 1994).

The Vilaine contributions to the DIN concentrations in the VB, estimated by the model, are slightly lower than those of the Loire whatever the period, on average 23% and 17% for flood and low water periods respectively. The Vilaine contributes, on average, to 5% and 8% of VB DIP concentrations during flood and low water periods respectively.

8.  Page 6 lines 7-11: "For the Loire, river discharge measurements at Montjean-sur-Loire were used due to the absence of data at Sainte-Luce-sur-Loire. For the Vilaine, daily discharge data were available at Rieux from the 1980s. DIN and DIP loads from rivers were calculated using averaged monthly discharge and individual monthly nutrient concentrations (Romero et al., 2013)."

    It will be good to mention the distance between stations, or catchment areas differences. Also, I suggest transposing discharge values from one station to another using a catchment area ratio Q_stationB = Q_stationA * area_catchmentB / area_catchmentA, or at least show that difference in drainage areas are small

    We added the information in drainage area as follows (p.6, l.12-14):

    The difference in drainage areas between Montjean (109 930 km²) and Sainte-Luce (111 570 km²) is negligible (ratio of 1.01) and would not have significantly changed calculations of nutrient inputs.

9.  Page 7 lines 17-18: "DSi in rivers was not analyzed for trends because of the short data period". Please, specify this period. Considered changing this sentence into "Trend analysis was not conducted on DSi in rivers because data covered the period XXXX-XXXX only"

    The period was added, and the sentence was modified according to the referee suggestion

10. Page 8 line 12: change "readers are referred to..." to "readers may refer to..."

    Done

11. Page 8 lines 16-17: consider changing "which allows modelling up to bimodal pattern" into "which considers potential bimodal pattern"

    Done

12. Page 9 lines 9-10: consider changing "For the longest common record of all variables, 1997-2013 called the "common period", a monotonic..." into "For the period 1997-2013, common to all variables ("common period" hereafter in the text), a monotonic..."

    Done

13. Page 11 line 9: change "Trends of DIP and DIP loads..." into "Trends of DIP concentration and load..."

Done

14. Page 15 line 8: add reference to Dupas et al., Multidecadal trajectory of riverine nitrogen and phosphorus dynamics in rural catchments - Water Resources Research, 2018 https://doi.org/10.1029/2018WR022905

   Done

15. Legend figure 8: watch for typo: "below" instead of "bellow"

   Done

[revised manuscript text omitted]
^{-1}$) $p$ | Change (%) | DIP ($\mu mol\ L^{-1}$) $p$ | Change (%) | DIP loads ($mol\ s^{-1}$) $p$ | Change (%) | DIN ($\mu mol\ L^{-1}$) $p$ | Change (%) | DIN loads ($mol\ s^{-1}$) $p$ | Change (%) | DSi ($\mu mol\ L^{-1}$) $p$ | Change (%) | Chl $a$ ($\mu g\ L^{-1}$) $p$ | Change (%) | Diatoms (Cells $L^{-1}$) $p$ | Change (%) | Dinoflagellates (Cells $L^{-1}$) $p$ | Change (%) |
|---|---|---|---|---|---|---|---|---|---|---|---|---|---|---|---|---|---|---|
| Loire | 0.01 | − 94 (16%) | <0.001 | − 0.85 (47%) | <0.001 | − 0.60 (52%) | 0.63 | NS | 0.42 | NS | | | <0.001 | − 54 (93%) | | | | |
| Vilaine | 0.02 | − 8.7 (23%) | <0.001 | − 1.9 (75%) | <0.001 | − 0.09 (88%) | <0.001 | − 71 (21%) | <0.001 | − 4.6 (38%) | | | <0.001 | − 12 (76%) | | | | |
| VB | | | <0.001 | − 0.05 (13%) | | | 0.01 | + 3.2 (40%) | | | <0.001 | + 3.6 (34%) | <0.001 | + 2.1 (126%) | <0.001 | + 90*10³ (227%) | <0.001 | + 8*10³ (108%) |

**Table 2: Statistical results of modified Mann-Kendal test performed on DLM seasonal components of eutrophication parameters in rivers and in the VB for the common period 1997–2013. If the test was significant at *p*<0.05, percentages of changes relative to the initial values of the Sen's robust line were calculated. Increasing or decreasing trends are indicated by + and − signs respectively. NS = non-significant**

| Site/ Season | Discharge ($m^3 s^{-1}$) | | DIP ($\mu mol\ L^{-1}$) | | DIP loads ($mol\ s^{-1}$) | | DIN ($\mu mol\ L^{-1}$) | | DIN loads ($mol\ s^{-1}$) | | DSi ($\mu mol\ L^{-1}$) | | Chl *a* ($\mu g\ L^{-1}$) | | Diatoms (Cells $L^{-1}$) | | Dinoflagellates (Cells $L^{-1}$) | |
|---|---|---|---|---|---|---|---|---|---|---|---|---|---|---|---|---|---|---|
| | *p* | % | *p* | % | *p* | % | *p* | % | *p* | % | *p* | % | *p* | % | *p* | % | *p* | % |
| **Loire** | | | | | | | | | | | | | | | | | | |
| Winter | 0.63 | NS | 0.04 | − 23% | <0.01 | − 41% | 0.02 | − 24% | <0.01 | − 40% | | | <0.001 | + 190% | | | | |
| Spring | 0.50 | NS | <0.001 | − 28% | 0.02 | − 33% | 0.21 | NS | 0.49 | NS | | | <0.001 | + 283% | | | | |
| Summer | 0.60 | NS | <0.001 | + 33% | <0.001 | + 59% | <0.01 | + 55% | 0.01 | + 69% | | | 0.09 | NS | | | | |
| Autumn | 0.98 | NS | <0.01 | + 35% | 0.26 | NS | 0.29 | NS | 0.92 | NS | | | <0.001 | − 82% | | | | |
| **Vilaine** | | | | | | | | | | | | | | | | | | |
| Winter | 0.23 | NS | 0.02 | − 17% | 0.07 | NS | 0.90 | NS | 0.11 | NS | | | <0.01 | + 97% | | | | |
| Spring | 0.93 | NS | 0.06 | NS | 0.07 | NS | 0.99 | NS | 0.56 | NS | | | <0.001 | + 63% | | | | |
| Summer | 0.26 | NS | <0.001 | + 9.4% | 0.09 | NS | 0.29 | NS | 0.28 | NS | | | <0.001 | − 41% | | | | |
| Autumn | 0.97 | NS | 0.51 | NS | 0.40 | NS | 0.66 | NS | 0.69 | NS | | | 0.01 | − 44% | | | | |
| **VB** | | | | | | | | | | | | | | | | | | |
| Winter | | | 0.73 | NS | | | 0.03 | + 32% | | | 0.329 | NS | 0.11 | NS | 0.85 | NS | 0.05 | NS |
| Spring | | | <0.001 | − 30% | | | 0.10 | NS | | | 0.086 | NS | <0.001 | − 36% | 0.93 | NS | 0.83 | NS |
| Summer | | | <0.001 | + 80% | | | 0.17 | NS | | | 0.085 | NS | 0.19 | NS | <0.001 | + 43% | <0.001 | − 23% |
| Autumn | | | 0.94 | NS | | | 0.76 | NS | | | 0.647 | NS | 0.37 | NS | 0.27 | NS | 0.87 | NS |

**Table 3: Spearman's rank correlations between annual median values of river discharge, nutrient concentrations and phytoplankton biomass in the Loire, Vilaine and the VB for the common period 1997-2013 ($n = 17$). Asterisks designate significant correlations (***$p<0.001$, **$p<0.01$, *$p<0.05$)**

| | Loire discharge | Vilaine discharge | DIN Loire | DIP Loire | Chl *a* Loire | DIN Vilaine | DIP Vilaine | Chl *a* Vilaine | DIN VB | DIP VB | DSi VB | Chl *a* VB |
|---|---|---|---|---|---|---|---|---|---|---|---|---|
| Loire discharge | 1.00 | | | | | | | | | | | |
| Vilaine discharge | 0.88*** | 1.00 | | | | | | | | | | |
| DIN Loire | 0.52* | 0.39 | 1.00 | | | | | | | | | |
| DIP Loire | 0.51* | 0.43 | 0.44 | 1.00 | | | | | | | | |
| Chl *a* Loire | -0.08 | -0.06 | 0.25 | 0.35 | 1.00 | | | | | | | |
| DIN Vilaine | 0.33 | 0.47 | 0.02 | 0.55* | 0.59* | 1.00 | | | | | | |
| DIP Vilaine | 0.16 | 0.24 | 0.23 | 0.77** | 0.65* | 0.54 | 1.00 | | | | | |
| Chl *a* Vilaine | -0.21 | -0.28 | 0.31 | 0.20 | 0.64** | 0.04 | 0.35 | 1.00 | | | | |
| DIN VB | 0.78** | 0.74** | 0.36 | 0.35 | -0.10 | 0.29 | -0.01 | -0.20 | 1.00 | | | |
| DIP VB | 0.13 | -0.09 | 0.07 | 0.38 | 0.05 | 0.11 | 0.29 | 0.19 | -0.12 | 1.00 | | |
| DSi VB | 0.55* | 0.41 | 0.35 | 0.08 | -0.48 | -0.17 | -0.51 | -0.31 | 0.63* | -0.02 | 1.00 | |
| Chl *a* VB | 0.11 | 0.17 | -0.14 | -0.48 | -0.61* | -0.34 | -0.58* | -0.50* | 0.25 | -0.45 | 0.33 | 1.00 |